# AYA VISION: ADVANCING THE FRONTIER OF MULTILINGUAL MULTIMODALITY

## ABSTRACT

Building multimodal language models is fundamentally challenging: requiring alignment of vision and language modalities, curating high-quality instruction data, and preserving existing text-only capabilities once vision is introduced. These difficulties are further magnified in multilingual settings, where the need for multimodal data in different languages exacerbates existing data scarcity, machine translation often distorts meaning, and catastrophic forgetting is more pronounced. To address these issues, we propose: (1) a synthetic annotation framework that curates high-quality, diverse multilingual multimodal instruction data across many languages; (2) a cross-modal model merging technique that mitigates catastrophic forgetting, effectively preserving text-only capabilities while simultaneously enhancing multimodal generative performance. Together, these contributions yield **Aya Vision**, a family of open-weights multilingual multimodal models (8B and 32B) that achieve **leading performance across both multimodal and text-only tasks**, outperforming significantly larger models. Our work provides guidance and reusable components for scalable multilingual data curation, robust multimodal training, and advancing meaningful evaluation in multilingual multimodal AI.

## 1 INTRODUCTION

Multimodal large language models (MLLMs) (55; 54; 20; 96; 45; 14; 7; 98) have achieved significant advancements in joint reasoning across modalities but predominantly remain limited to English. This language barrier limits global accessibility and reduces their practical impact.

Expanding MLLMs to multilingual settings brings several key challenges. First, there is a serious lack of high-quality multimodal datasets covering diverse languages. Despite recent progress in multilingual language modeling (101; 19; 16), multimodal resources are typically limited to short, simplistic, and task-specific image-text pairs (27; 103; 84), which

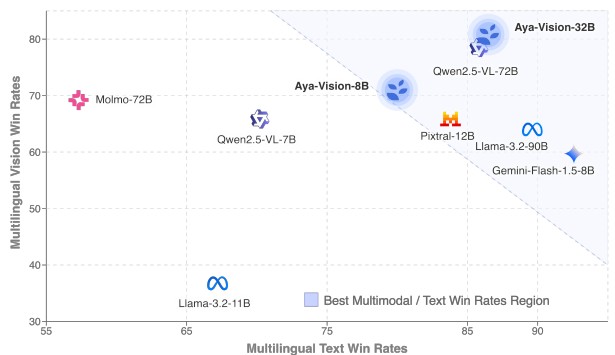

Figure 1: **Aya Vision sets a new standard for multilingual performance across modalities in 23 languages.** Aya-Vision-8B delivers best-in-class multimodal performance without sacrificing text capabilities, while Aya-Vision-32B outperforms all baselines, including much larger models, achieving an optimal trade-off between efficiency and cross-modal strength.

do not reflect the complexity of real-world conversational scenarios. Machine translation is commonly used to address this gap, but often introduces linguistic artifacts like "translationese", as well as cultural biases and misalignments (102; 83; 32; 66; 91; 82; 105; 73). Creating accurate, diverse and context-aware multilingual multimodal instruction data remains an open and essential problem.

Another issue is the known trade-off between adding visual capabilities and preserving strong text-only performance. Incorporating vision often leads to catastrophic forgetting, where previously learned language abilities degrade (6; 20; 28; 72). This effect worsens as models scale to more

languages. Evaluating progress is also challenging due to the limited scope of existing tools. Most benchmarks rely on constrained, multiple-choice formats (12; 81; 112), which do not capture the open-ended interactions of real-world use. The few existing benchmarks that support more complex, generative tasks (58; 3) are currently English-only, leaving multilingual multimodal evaluation largely unexplored.

In this work, we tackle these challenges jointly. To address data scarcity, we replace naive translation pipelines with a hybrid approach that combines a specialized translation model with a larger LLM to detect and correct systematic translationese artifacts. We call this method **context-aware rephrasing**, which enables the creation of higher-quality, human-preferred multilingual multimodal instruction data. To mitigate catastrophic forgetting, we propose a **novel cross-modal merging strategy** (§ 3) that fuses capabilities across models, enabling preservation and "on-the-fly" extension of skills across modalities. We view this as a powerful paradigm for efficiently adapting models to new tasks. Our merging strategy improves performance by 50.2% on text-only tasks and 20.5% on multimodal tasks relative to the unmerged checkpoint, leveraging the compositionality between tasks and modalities.

The result of our work is **Aya Vision**, a family of multilingual multimodal models in 8B and 32B sizes, designed for fluent, instruction-following generation across 23 languages. Aya-Vision-8B outperforms Qwen-2.5-VL-7B, Llama-3.2-11B-Vision, Pixtral-12B, and Gemini-Flash-1.5-8B, achieving up to a 79% win rate across multimodal tasks. Aya-Vision-32B surpasses models more than twice its size, including Llama-3.2-90B-Vision, Molmo-72B, and Qwen-2.5-VL-72B, with win rates up to 72.4%.

Our key contributions are:

1. **A family of state-of-the-art multilingual multimodal LLMs (Aya-Vision-8B/32B):** Trained to generate fluent, conversational outputs in 23 languages spoken by half the world's population. Aya Vision models are optimized for multilingual and multimodal instruction-following, and achieve strong human preference [1].

2. **A multilingual multimodal synthetic annotation framework:** We introduce a pipeline combining synthetic data distillation, automatic translation, and context-aware rephrasing, which significantly expands the length and diversity of image-text pairs (average tokens increase from 27.2 to 140.8; lexical diversity from 11.0 to 61.2), and improves translation quality by 11.24%.

3. **Cross-modal model merging for capability preservation and enhancement:** Our method merges pretrained models to counteract catastrophic forgetting. It restores lost text capabilities (up to +50.2% text win rate) and improves vision-language understanding (+20.5% win rate), *without additional training*.

4. **New benchmark for multilingual multimodal evaluation:** We release *AyaVisionBench*[1], covering 23 languages and 9 vision-language tasks, and *m-WildVision*[1], a high-quality translation of WildVision (58). Together, they offer a meaningful and challenging testbed for multilingual multimodal models.

## 2 A COMPREHENSIVE MULTILINGUAL MULTIMODAL DATA FRAMEWORK

We introduce a robust multimodal synthetic re-annotation pipeline for constructing high-quality multilingual instruction dataset. As shown in Figure 2, our pipeline consists of three key stages: (1) *distillation-based recaptioning*, (2) *dataset filtering*, and (3) *translation with multilingual rephrasing*. This process significantly improves linguistic diversity, naturalness, and coverage across 23 languages.

**Data Collection.** We begin by curating a diverse English multimodal instruction-tuning dataset. Our collection builds on open-source resources, most notably *Cauldron* (46), which aggregates 50 vision-language datasets (∼30M), and *PixMo*(20), covering 7 multimodal tasks (∼6M). Additional sources such as *SlideVQA* (93), *PDFVQA* (21), and *ScreenQA* (34), with overall coverage of visual

---

[1]We will release both models and benchmarks here: https://huggingface.co/collections/xxx

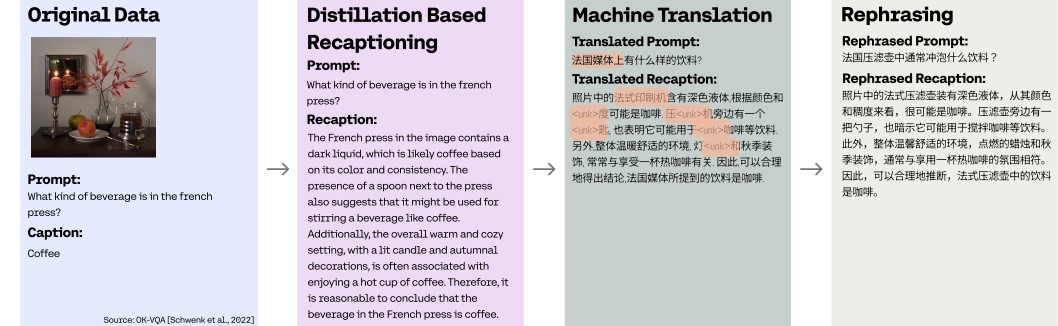

Figure 2: **Our synthetic annotation pipeline produces diverse, high-quality multimodal responses.** It includes three stages: (1) recaptioning, (2) translation, and (3) LLM-based rephrasing. Rephrasing corrects common translation errors – e.g., unknown tokens ("consistency") or lexical ambiguities ("French press" → "French media") – improving fluency and semantic accuracy.

question answering (VQA), captioning, document understanding, chart and figure analysis, table reasoning, logical problem-solving, textbook QA, image comparison, and screenshot-to-code. To ensure task balance and promote generalization, we regulate the sample count across categories. The resulting dataset comprises approximately 2.29M examples. Table 3 in Appendix D presents the task-wise distribution. This curated English dataset serves as the basis for further downstream recaptioning and multilingual synthesis pipeline.

**Distillation-based Recaptioning.** Our goal is to alter the data distribution to better reflect real-world usage. To this end, we generate synthetic alternatives to the original completions across the ∼2.3M examples we collected. The original data primarily sourced from open-source, academic image captioning corpora like MS-COCO (51), Visual Genome (43), Open Images (44), and exhibits limited linguistic variety and stylistic repetition. Captions are typically short (avg. 14.2 words), simple, and lack the conversational tone expected from state-of-the-art generative models.

We address these limitations through a recaptioning pipeline that rewrites captions using task-specific prompt templates to guide our open-weight multimodal teacher model. Prompts are carefully designed to retain consistent with ground-truth answers while enhancing fluency and informativeness. For example, prompts for reasoning tasks elicit step-by-step outputs, while captioning tasks encourage longer, more vivid descriptions. Prompt design is essential to recaptioning effectiveness (30; 23); Examples are shown in Appendix K.

This process bridges the gap between narrowly scoped training data and the diverse language expected in modern multimodal systems. After recaptioning, the average word count increases from 14.2 to 100.1, token count from 27.2 to 140.8, and lexical diversity (measured by MTLD (87)) improves from 11.0 to 61.2, approaching the variability found in fluent human writing (64; 70). These more expressive annotations improve generalization and robustness in downstream tasks; Recaptioned examples can be found in Appendix L.

**Verifying and Filtering Recaptioned Instruction Data.** While recaptioning enhances data diversity and fluency, it can introduce hallucinations or factual errors ungrounded in the image (79; 53; 50; 29). Training on such data may amplify a models tendency to hallucinate or produce inaccurate outputs. To mitigate this, we implement a two-stage filtering pipeline to improve the reliability of the recaptioned dataset. Unlike single-pass filters like CLIP score-based filtering (25) or reward-based hallucination mitigation (8; 104), our method adds a second semantic safeguard to detect fluent but incorrect generations.

**Stage 1: Keyword-based filtering.** We begin with keyword detection to identify common failure modes in recaptioned outputs, such as refusals to respond or repeated prompt phrases. A curated list of keywords is used to automatically identify these issues. Flagged samples are either regenerated or discarded if problems persist. While effective for surface-level errors, keyword matching struggles with subtler issues, especially in tasks requiring deterministic or subjective answers like QA or math reasoning. In such cases, the teacher model may ignore ground truth or hallucinate details, leading to flawed outputs.

**Stage 2: LLM-based semantic filtering.** To address more nuanced errors, we apply a second-stage filtering using `command-r-plus-08-2024`[2] for semantic verification (see Appendix M for prompt and filtered examples). The original and rephrased captions are presented to the model, which acts as a semantic judge to assess whether the answer to the original remains valid in the rephrased version. This ensures that recaptions do not alter the intended meaning or contradict the ground truth. All corrupted samples identified are discarded. The overall error rate is 3.2% with more errors in complex tasks – 4.6% in reasoning versus 2.5% in VQA tasks – aligning with trends observed in prior work (111; 107; 92). Combined with keyword filtering, this semantic check yields a cleaner, more reliable dataset for visual instruction tuning.

**Hybrid Translation Pipeline for Multilingual Instruction Data.** Unlike prior work that relies solely on proprietary LLMs (112; 59) or highlights cross-lingual gaps without addressing mitigation strategies (33), we propose a two-stage **hybrid approach** to multilingual translation. Although GPT models perform well in high-resource languages, they often struggle in low-resource settings. Meanwhile, high-quality, in-language datasets remain scarce and are mostly reserved for evaluation (91; 80; 1; 82). Translating instruction data has proven effective for enhancing cross-lingual generalization (75; 19; 22; 101). However, machine translation can introduce issues like unnatural phrasing or semantic drift (11; 102; 91). To balance coverage and quality, we first use the NLLB-3.3B model[3] (17) to translate our English dataset into 22 languages (Appendix C). Then, we apply post-editing using `command-r-plus-08-2024`[2], which uses the machine output as in-context input to improve fluency and fix common errors while preserving semantics (120; 76). Prompt templates and examples are provided in detail in Appendix N.

To ensure training efficiency and avoid overfitting, we translate only subsets of the English data per language, reducing duplication and repeated exposure. Partial translation has been shown to maintain strong generalization while reducing data volume (26; 85; 66; 67; 5). Translation quality is assessed with the reference-free metric **COMET**[4] (78; 77). Average scores improve from **0.75** (NLLB) to **0.83** after post-editing, indicating a significant gain in fluency and adequacy. Language-specific improvements are in Table 7 (Appendix O).

## 3 OPTIMIZING ACROSS LANGUAGES AND MODALITIES WITH CROSS-MODAL MERGING

Achieving optimal performance in multilingual multimodal LLMs requires careful balancing of the fine-tuning data across languages, modalities, and tasks (55; 46; 99; 18). Skewed language distributions reduce generalization, and real-world applications demand that models support both text-only and multimodal use cases. A key challenge is preserving the strong text-only capabilities of the base LLM while adding robust multimodal abilities.

Simply adding text-only data during multimodal fine-tuning (20; 112) often fails to preserve text performance (Figure 3) and can lead to overfitting, while reusing previously seen text offers minimal benefit and may degrade multimodal capabilities (60). We address this using two complementary strategies.

**1. Weighted sampling of diverse data sources:**, We design a *balanced fine-tuning mix* by sampling from three data sources: (i) upsampled, synthetically re-annotated English data (3.5M seen samples from 2.29M original) to ensure coverage of diverse tasks and high-quality examples; (ii) uniformly sampled multilingual data (3.4M out of 5M), covering 22 non-English languages while preserving task balance; and (iii) downsampled high-quality orig-

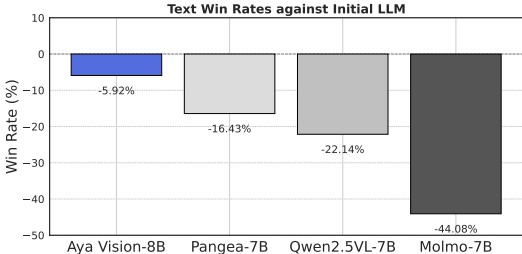

Figure 3: **Degradation in text-only win-rates after multimodal training.** Each model is compared to their initial LLM on m-ArenaHard (19). Including a percentage of text-only data in the final multimodal training mix is insufficient to retain open-ended generative performance.

[2]https://huggingface.co/CohereLabs/c4ai-command-r-plus-08-2024
[3]https://huggingface.co/facebook/nllb-200-3.3B
[4]https://huggingface.co/Unbabel/wmt23-cometkiwi-da-xxl

inal datasets (3.7M from 6M) to support evaluation-specific formats (e.g., short-form VQA) without overpenalizing free-form generation. The final training set comprises 2.75M sequence-packed samples: 66% synthetically re-annotated data (35% multilingual), and 34% high-quality original datasets (see details in Figure 10 and Figure 8). Contrary to prior work (112; 20), we do not include any text-only data during training.

**2. Cross-model model merging:** To recover text-only performance without sacrificing vision capabilities, we introduce a training-free method: *cross-modal model merging*. Concretely, we posit that since the multimodal model is initialized from the final preference-tuned LLM checkpoint, sharing a part of the optimization trajectory (37; 24; 36) makes the multimodal LLM and the backbone LLM amenable to merging. Thus, rather than adding more text data, we linearly interpolate the weights of the preference-tuned text-only LLM and the multimodal model, preserving visual modules for restoring text quality:

$$W_{\text{merged}} = \alpha \cdot W_{\text{mm-LLM}} + (1 - \alpha) \cdot W_{\text{text-LLM}}$$

This approach effectively balances capabilities across modalities and improves text-only performance *a posteriori*, with no additional training (§7).

## 4 ARCHITECTURE AND TRAINING DETAILS

**Architecture.** Aya Vision follows the common late-fusion architecture for vision-language models (55; 54; 46; 65; 14; 20), comprising three main components: (1) a vision encoder that produces image patch embeddings (74; 115; 14; 100), (2) a vision-language connector that maps these embeddings into the language models input space, and (3) a large language model. Further architectural details are provided in Appendix F.

**Multimodal Training.** Aya Vision is trained in two stages: during *vision-language alignment*, we freeze both the vision encoder and language model, and train only the connector to map image features into the LLM input space. This stage uses LLaVA-Pretrain[5] (English-only), with 14% of the data drawn from our multilingual pipeline to improve cross-lingual grounding. In the subsequent *supervised fine-tuning (SFT)* stage, we unfreeze the connector and language model (keeping the vision encoder frozen), and experiment with both full and LoRA-based tuning (35). We apply sequence packing (up to 8192 tokens) to improve training efficiency. Dataset composition is shown in Figure 10, with further discussion in §3. Hyperparameters are listed in Table 5.

## 5 EVALUATION

**Baselines.** We compare Aya Vision models against a range of state-of-the-art multimodal LLMs, both open- and closed-weight, to evaluate multilingual, multimodal, and text-only capabilities. We select models based on architecture, model size, base model family, and language coverage. The selected models cover a range of sizes (7B to 90B), base models (Llama-3.2, Qwen-2.5, Molmo), and language coverage (including both English and multilingual models). Our evaluation includes open-weight models (Pixtral (3), Molmo (20), Qwen-2.5-VL (7) and Pangea (112)) as well as the closed-weight (Gemini-Flash-1.5 (96)). For model families, Qwen, Molmo, and Llama, we report results across multiple sizes ranging from 7B to 90B.

**Multilingual Multimodal Evaluation.** While recent efforts have explored multilingual evaluation for multimodal LLMs (12; 81; 94; 112), existing benchmarks still fall short of enabling robust, real-world evaluation. Most focus on static, single-turn tasks with predefined answers, failing to capture the nuanced, open-ended, and dynamic nature of real-world user interactions. To address this, we introduce: **AyaVisionBench**, a benchmark designed to evaluate multilingual multimodal models on generation quality across 23 languages, with a focus on relevance, fluency, and engagement. It emphasizes open-ended instruction following and cross-modal reasoning. Construction details are in Appendix E.1.

To complement AyaVisionBench, we release **m-WildVision**, a multilingual extension of WildVision-Bench (58) across 23 languages, with translated prompts designed to evaluate open-ended multimodal generation across diverse linguistic contexts. We also include **xChatBench**

---

[5]https://huggingface.co/datasets/liuhaotian/LLaVA-CC3M-Pretrain-595K

(112), which enables fine-grained, score-based evaluation across 7 languages and multiple interaction types. Evaluation protocols for all three benchmarks are detailed in Appendix E.1.1. In addition to the preference-based open-ended evaluation, we evaluate Aya Vision on structured multimodal benchmarks that require constrained outputs (e.g., multiple choice or short-form answers) for automatic scoring. Specifically, we use **xMMMU** (112), **MaXM** (12), **CVQA** (81), **MTVQA** (94) and **Kaleidoscope** (82). These benchmarks cover a range of languages and tasks, evaluating multimodal understanding, reasoning, and knowledge. Language coverage is listed in Table 4, with additional details in Appendix E.

**Multilingual Text-Only Evaluations.** As shown in Figure 3, vision-language models often suffer degradation in text-only performance. To assess this, we evaluate Aya Vision and baselines on multilingual text benchmarks as a final component of our evaluation suite. We evaluate models using two complementary approaches: open-ended evaluation and task-specific benchmarks. For **open-ended evaluation**, we use m-ArenaHard (49; 19) to assess models' performance in free-form text generation across 23 languages. Following (19), we adopt gpt-4o-2024-11-20 as the LLM judge. For **task-specific benchmarks**, we evaluate models on MGSM (88), Global MMLU-Lite (90), and FLORES (31), which cover mathematical reasoning, multilingual understanding, and machine translation, respectively. For FLORES, we evaluate translation from English to the target language (En→X), as it presents a greater challenge and better reflects multilingual capabilities. We also include IFEval (117), an English-only benchmark, to assess instruction-following skills that may influence both text-only and multimodal tasks. Each benchmark covers a distinct set of languages, with metrics summarized in Table 4; further details are provided in Appendix E.

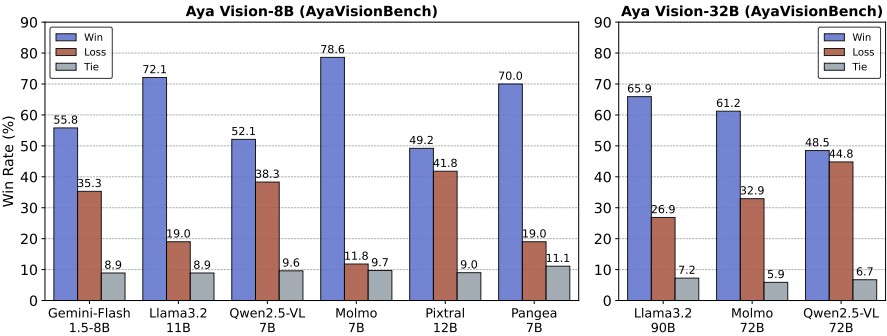

Figure 4: **Aya-Vision-8B and Aya-Vision-32B achieve strong performance on preference evaluation.** Pairwise win rates on AyaVisionBench, averaged across 23 languages. Aya-Vision-8B is compared against Gemini-Flash-8B, Llama-3.2-11B-Vision, Qwen-2.5-VL-7B, Pixtral-12B, and Pangea-7B. Aya-Vision-32B is compared against Llama-3.2-91B-Vision, Qwen-2.5-VL-72B, Molmo-72B. Language-specific breakdowns are provided in Tables 9 and 12 in the Appendix R.

| Models / Evaluations | MaxM | xMMMU | CVQA | MTVQA | Kaleidoscope | xChat | avg |
|---|---|---|---|---|---|---|---|
| Pangea-7B | 51.27 | 44.00 | 60.53 | 18.32 | 29.46 | 32.21 | 39.30 |
| Molmo-7B-D | 44.16 | 37.87 | 58.53 | 16.89 | 36.42 | 23.36 | 36.21 |
| Llama-3.2-11B-Vision | 39.30 | 42.73 | 58.92 | 16.40 | 36.50 | 28.59 | 37.07 |
| Pixtral-12B | 44.43 | 42.27 | 63.54 | 19.81 | 36.08 | **64.50** | 45.11 |
| Qwen-2.5-VL-7B | 52.65 | **46.77** | **73.22** | **29.57** | **39.64** | 58.14 | **50.00** |
| Aya-Vision-8B | **58.21** | 39.94 | 61.86 | 19.33 | 38.62 | 58.64 | 46.16 |
| Molmo-72B | 55.62 | 51.53 | 72.77 | 18.66 | 50.34 | 45.43 | 49.06 |
| Llama-3.2-90B-Vision | **64.17** | 52.40 | 81.88 | 27.44 | 48.41 | 51.12 | 54.24 |
| Qwen-2.5-VL-72B | 56.42 | **61.74** | **82.10** | **31.92** | **55.02** | **71.13** | **59.72** |
| Aya-Vision-32B | 62.28 | 45.11 | 74.06 | 23.46 | 41.73 | 70.07 | 52.81 |

Table 1: **Evaluation on multilingual multimodal benchmarks for Aya-Vision-8B and Aya-Vision-32B, alongside baselines**. For each benchmark, we report results on languages included in Aya-Vision's 23-language set. The full results for all languages are provided in the Appendix R.

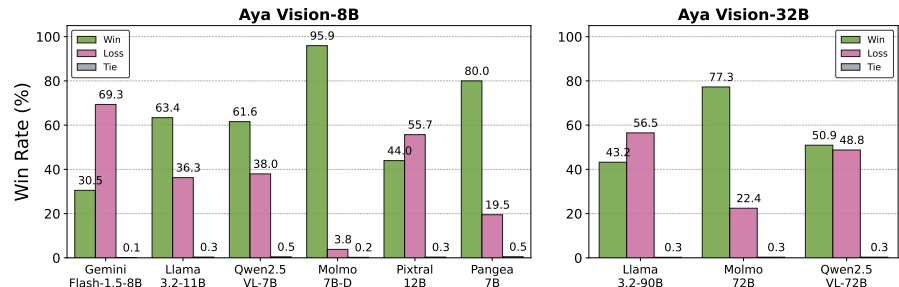

Figure 5: **Aya-Vision models rank among the top performers in text-only preference evaluation, outperforming much larger models.** Pairwise win rates for Aya-Vision-8B (left) and Aya-Vision-32B (right) on m-ArenaHard (19), averaged over 23 languages. Language-specific breakdowns are provided in Tables 8 and 11 in the Appendix R.

## 6  RESULTS AND DISCUSSION

**Aya-Vision-8B achieves best-in-class performance in preference evaluation.** Figure 4 and Figure 12 in the Appendix E.4 show pairwise win rates on AyaVisionBench and m-WildVision, averaged over 23 languages, comparing Aya-Vision-8B with state-of-the-art multimodal LLMs. Aya-Vision-8B consistently outperforms all baselines, with win rates ranging from 49.6% to 80.3%. Performance is slightly higher on m-WildVision, by an average of 6%, likely due to the more challenging nature of AyaVisionBench, as indicated by higher tie rates. Aya-Vision-8B surpasses both Qwen-2.5-VL-7B and Pixtral-

| Models | GMMLU | MGSM | FLORES | IFEval | avg |
|---|---|---|---|---|---|
| Pangea-7B | 49.35 | 50.51 | 28.04 | 23.99 | 37.97 |
| Molmo-7B-D | 39.63 | 49.94 | 15.74 | 56.10 | 40.35 |
| Llama-3.2-11B | 60.75 | 72.84 | _31.84_ | **83.43** | _62.22_ |
| Pixtral-12B | **66.09** | **77.62** | 29.29 | 65.59 | 59.65 |
| Qwen-2.5-VL-7B | _64.82_ | 60.90 | 27.98 | 72.46 | 56.54 |
| Aya-Vision-8B | 62.52 | _76.42_ | **35.90** | _82.78_ | **64.41** |
| Molmo-72B | 71.02 | _86.00_ | 32.52 | 78.10 | 66.91 |
| Llama-3.2-90B | _77.46_ | 66.67 | **38.25** | _88.14_ | _67.63_ |
| Qwen-2.5-VL-72B | **81.49** | **89.61** | 35.71 | **89.74** | **74.14** |
| Aya-Vision-32B | 63.58 | 79.46 | _37.79_ | 78.50 | 64.83 |

Table 2: **Evaluation on multilingual text-only academic benchmarks for Aya-Vision-8B and Aya-Vision-32B together with the baselines**. For each benchmark, we include languages that are in the list of Aya Vision's 23 languages. The results for all languages are provided in the Appendix R.

12B by 54.8% win rate averaged across the two datasets, despite Pixtral-12B being a larger model. It also outperforms the strong proprietary model Gemini-Flash1.5-8B, averaging a 60.3% win rate, and achieves a dominant 71.7% win rate over Pangea-7B, which is trained with a predominantly multilingual dataset.

**Aya Vision outperforms far larger models.** Figure 4 and Figure 12 in the Appendix E.4 show pairwise win rates for Aya-Vision-32B on AyaVisionBench and m-WildVision, averaged across 23 languages. Aya-Vision-32B consistently outperforms models more than twice its size – such as Molmo-72B, Qwen-2.5-VL-72B, and Llama-3.2-90B-Vision – with win rates ranging from 48.5% to 73%. Notably, it surpasses Llama-3.2-90B-Vision by 65.9% on AyaVisionBench and 73% on m-WildVision. Its closest competitor, Qwen-2.5-VL-72B, is outperformed by 50.8% on average across both benchmarks.

**Aya-Vision models achieve competitive performance on academic benchmarks.** Although optimized for open-ended generation, Aya-Vision models perform strongly on multiple-choice and short-form academic benchmarks, which often fail to fully capture the generative capabilities of modern MLLMs. Results are shown in Table 1. On MaxM, a short-form VQA benchmark, Aya-Vision-8B outperforms all models in its parameter class, including larger ones like Pixtral-12B and Llama-3.2-11B-Vision. On Kaleidoscope, it performs competitively with Qwen-2.5-VL-7B and surpasses all other baselines. Aya-Vision-32B also delivers strong results, outperforming Molmo-72B on all benchmarks except xMMMU, and closely matching Llama-3.2-90B-Vision on average despite being nearly 3× smaller.

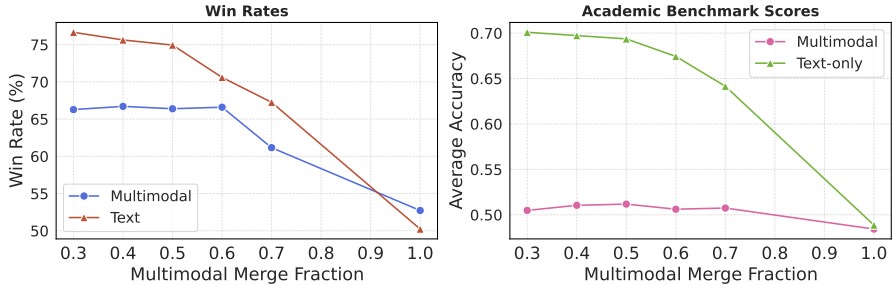

Figure 7: **Impact of cross-modal merging across various merge ratios.** Win rates are computed against Pangea-7B on AyaVisionBench (multimodal) and m-ArenaHard (text-only) across 7 languages. The multimodal academic score is the average of CVQA and xMMMU, while the text-only academic score averages IFEval, MGSM, and MMMLU (subset).

**Aya Vision models punch above their size in text-only preference evaluation.** A key concern with multimodal models is that adding vision capabilities may compromise text performance. To evaluate this trade-off, we assess text-only results on the m-ArenaHard dataset using pairwise win rates averaged across 23 languages, as shown in Figure 5. At the 8B scale, Aya-Vision-8B strikes a strong balance between performance and efficiency, outperforming all open models in its class and rivaling proprietary ones. It achieves a win rate of 63.4%, surpassing the larger Llama-3.2-11B-Vision and remains competitive with Pixtral-12B, which achieves a slightly higher win rate of 56.0%. Aya-Vision-32B is even more efficient. It outperforms significantly larger models such as Molmo-72B with a win rate of 77.3% and Qwen-2.5-VL-72B with 50.9%. Despite being nearly three times smaller, it closely matches Llama-3.2-90B-Vision, which reaches 43.2%. These results demonstrate Aya-Vision's ability to deliver strong text performance at a fraction of the size, while maintaining multimodal capabilities, as shown in Figures 4 and 12 in the Appendix E.4.

To further understand text performance preservation, Figure 3 compares win rates on m-ArenaHard for Aya-Vision-8B, Pangea-7B, Qwen-2.5-VL-7B, and Molmo-7B relative to their base LLMs. Aya-Vision-8B shows minimal degradation, with only a 5.9% drop, demonstrating that cross-modal merging effectively retains text quality.

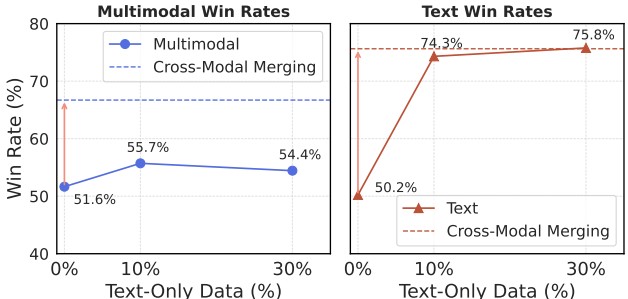

Figure 6: **Modal merging enables efficient cross-modal transfer.** Multimodal and text-only win rates on AyaVisionBench and m-ArenaHard against Pangea-7B. We vary the text-only mixture during SFT and compare it to cross-modal merging (dashed line).

## 7 KEY ABLATIONS

To isolate the impact of key design choices, we conduct controlled ablations at the 8B scale, varying only one factor at a time: (1) cross-modal model merging, (2) adding text-only data, (3) proportion of multilingual data during SFT. All other settings remain fixed. We evaluate each variant using multimodal and text win rates on AyaVisionBench and the m-ArenaHard subset[6], comparing them against Pangea-7B. Additionally we report average metrics on academic vision (CVQA, xMMMU) and text benchmarks (IFEval, MMMLU subset, MGSM). Additional ablation studies covering (4) the vision encoder, and (5) full fine-tuning versus low-rank adaptation, presented in Appendix H.

**Model merging improves multilingual performance across tasks and modalities; and is more effective than adding *seen* text data for cross-modal transfer.** We systematically evaluate our cross-modal model merging strategy by ablating the interpolation weight $\alpha$ between the fine-tuned multimodal LLM and its original text-only counterpart. An $\alpha$ of 0 corresponds to the text-only model, while $\alpha = 1$ is the fully multimodal one.

As shown in Figure 7 (left), merging not only preserves text-only multilingual performance but also unexpectedly boosts multilingual vision win rates as text-only contributions increase – up to

---

[6]English, French, Hindi, Arabic, Turkish, Japanese, Chinese

an optimal point. Text metrics improve steadily with higher text-LLM weighting, while vision performance plateaus. Based on these trends, we select $\alpha = 0.4$ as the optimal balance for both our 8B and 32B models.

We also compare merging to the conventional approach of adding seen text-only data during SFT in proportions of 0%, 10%, and 30%. Figure 6 shows that while more text data improves text win rates (from 50.2% to 74.8%), it does not translate to stronger multimodal performance. In fact, increasing text data from 10% to 30% slightly reduces multimodal win rates, likely due to more capacity being allocated to text modeling. These results confirm that model merging is a effective and efficient method for cross-modal knowledge transfer.

**Balanced multilingual data leverages cross-lingual transfer from English for best performance across modalities and languages.** To measure the impact of the ratio of multilingual data in the training mixture, we train 3 variants with varying proportions of multilingual multimodal data – 17.5%, 35%, and 67%, which is uniformly distributed across 22 languages (except English).

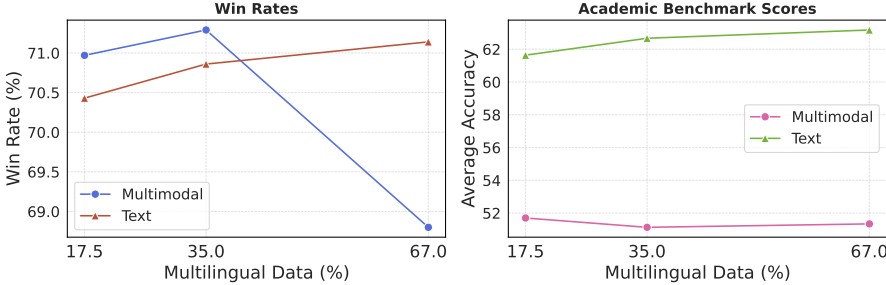

Figure 8: **A balanced data mixture is essential for multilingual multimodal performance.** Multimodal and text win-rates are calculated against Pangea-7B on AyaVisionBench and m-ArenaHard respectively over 7 languages. Multimodal academic benchmark is an average of CVQA and xM-MMU; Text-Only academic benchmarks are averaged over IFEval, MGSM and MMMLU (subset).

As shown in Figure 8, we find that increasing the ratio of multilingual multimodal data from 35% to 67% leads to degradation in the quality of generations – reducing the win-rates from 71.4% to 68.7%, and also hurts multimodal academic benchmarks, emphasizing the importance of the balance between English and multilingual data. Given the scarcity of high-quality multilingual multimodal data, upsampling this bucket requires repeating the data multiple times, limiting its benefit in multilingual multimodal performance. Additionally, a sufficient percentage of the more diverse English data is crucial for cross-lingual transfer.

**Both data improvements and cross-modal merging are essential to Aya Vision's performance.** Compared to a model trained purely on open-source task-specific data, each of our contributions significantly improves performance where our novel data framework leads to a 17% gain in win rate, underscoring the importance of fluent, detailed, and diverse completions. Next, our cross-modal merging enables an extra gain of 11.9% multimodal win rates beyond its significant impact on text-performance, achieving a total increase to nearly 30%.

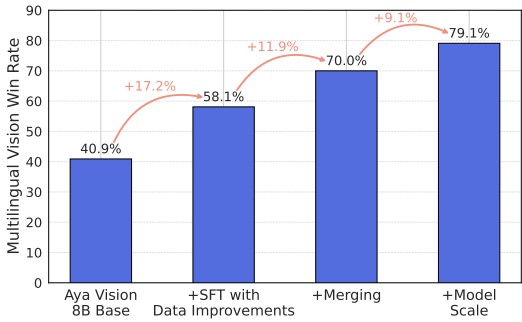

Figure 9: **Impact of various interventions.** Step-by-step improvements in Aya Vision 8B's pairwise win-rates against Pangea-7B.

## 8 CONCLUSION

In this work, we introduced Aya Vision, a family of multilingual vision-language models (8B and 32B) designed to improve multimodal understanding across 23 languages. Addressing key challenges in this space, we propose a scalable synthetic annotation framework to overcome multilingual data scarcity, and a training-free model merging approach to preserve text-only performance during multimodal training. Our models outperform existing open-weight baselines and are supported by AyaVisionBench, a benchmark tailored for evaluating generative multilingual multimodal systems.

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

## A   LIMITATIONS

Given the scarcity of high-quality multilingual data, in our multilingual data ablations, we sample the text-only data from the same corpus used for post-training the LLM using the Aya Expanse recipe (19); prior to the multimodal training. This leads to a portion of the data repeated across training stages which could potentially lead to over-fitting.

We use VLM-as-a-judge models for win-rates evaluations as a proxy for human preferences. While using large language models for win-rates evaluations is a standard practice (19; 101), for generations which are quite close, the judge preference might deviate from human preferences. We attempt to provide a comprehensive set of guidelines to the judge as shown in Appendix Q to ensure close adherence to human preferences.

## B   RELATED WORK

**Multilingual Multimodal Instruction Data.**   To overcome the scarcity of multilingual multimodal instruction datasets, several recent efforts have relied heavily on translating English-centric datasets using large language models (LLMs). Approaches such as PANGEA (112) and PALO (59) expand language coverage by translating large-scale instruction-following datasets or aligning multilingual captions. While effective in bootstrapping resources, these methods are constrained by limited linguistic diversity and suffer from "translationese" – artifacts of literal or non-fluent translations produced by automated systems. Furthermore, such datasets often exhibit rigid task formats and lack the conversational naturalness crucial for high-quality interaction in multilingual multimodal settings.

**Visual Instruction Tuning**  Visual instruction tuning (55; 13; 54; 14; 3; 106; 20; 7) combines a pretrained vision encoder (74; 115; 14; 100) with an offtheshelf large language model via a dedicated visionlanguage connector. This process extends the LLMs text capabilities into the visual domain while retaining its desirable attributes– such as in-context learning, reasoning, and instruction following. As a result, visual instruction tuning has emerged as a highly effective method to achieve state-of-the-art performance on a wide range of tasks – even outperforming certain proprietary models.

**Multilingual Multimodal Models**  Initial works on multilingual multimodal models (68; 38; 114) focused on learning robust, universal representations for retrieval tasks across modalities. However, these models require further downstream training to be used as generative models. On the other hand, (26; 13; 112) perform large-scale multilingual multi-task fine-tuning to enable multilingual understanding and generation. However, they focus only on vision-language academic benchmarks which are reference based – focusing on exact matches rather than free-form holistic evaluations of the generations.

**Multilingual Multimodal Evaluations**  Multilingual multimodal evaluation benchmarks have traditionally focused on visual question answering (VQA) tasks, where the model-generated response must exactly match a human-provided reference answer (12; 81; 94). This approach often penalizes responses that are semantically correct but differ syntactically from the reference (3). To address these limitations, recent work (112; 59) has proposed multilingual multimodal chat benchmarks. Instead of relying solely on exact matches, these benchmarks evaluate free-form responses by employing a Vision-Language model as an adjudicator–either by scoring responses against a detailed rubric or by selecting the superior generation from a pair of outputs.

**Multimodal Merging**  Recent work by (118) introduces REMEDY, a method for merging VLM weights – including the connector layer – after low-rank fine-tuning on various VLM tasks. However, REMEDY does not address the merging of weights that have been trained for different modalities. In a closely related concurrent work, (48) merges a text-only reward model with a vision-language model with the goal to specifically transfer the reward modeling capabilities from the text-based reward model to build a multimodal reward model.

| Task | Orig. | Multi. | Synth. | Total | Per(%) |
|---|---|---|---|---|---|
| General VQA | 269.0k | 311.2k | 168.2k | 748.4k | 27.2 |
| Captioning | - | 74.6k | 109.0k | 183.6k | 6.7 |
| OCR | 231.8k | 60.7k | 188.8k | 481.3k | 17.5 |
| Figures/Charts | 290.0k | 31.3k | 159.6k | 480.9k | 17.5 |
| Table Compr. | 77.5k | 260.7k | 56.5k | 394.7k | 14.4 |
| Reason./Logic/Math | - | 136.4k | 60.9k | 197.2k | 7.2 |
| Multi Image | 39.6k | 78.0k | 97.3k | 214.8k | 7.8 |
| Textbook/Academic | 19.1k | - | 12.8k | 31.9k | 1.2 |
| Screenshot Code | 9.5k | 5.2k | - | 14.7k | 0.5 |
| **Total** | **936.3k** | **958.1k** | **853.0k** | **2.75M** | **100%** |

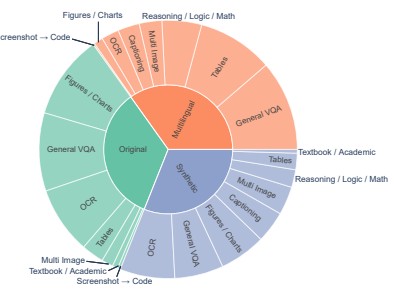

Figure 10: **Overview of our multilingual multimodal SFT mixture from various task categories.**
Left: Number of samples across data sources and tasks categories used in training. Right: Visual
breakdown of dataset source distributions.

## C  LANGUAGE COVERAGE

Arabic, Chinese, Czech, Dutch, English, French, German, Greek, Hebrew, Hindi, Indonesian,
Italian, Japanese, Korean, Persian, Polish, Portuguese, Romanian, Russian, Spanish, Turkish,
Ukrainian, Vietnamese

## D  DATA COLLECTION

Our curated English dataset contains approximately 2.29 million examples, spanning a wide range
of multimodal tasks. The task-wise breakdown, including both absolute counts and relative proportions, is summarized in Table 3.

Table 3: Task-wise distribution in our curated dataset, showing the proportion and the number of
samples in the ∼2.29M collection.

| Task | VQA | Capt. | OCR/ Doc | Chart/ Fig | Table Compr. | Logic. Reasoning | 2 Image Diff. | Textbook | SS to Code |
|---|---|---|---|---|---|---|---|---|---|
| **Total Samples** | 560K | 220K | 490K | 289K | 222K | 252K | 239K | 20K | 9.5K |
| **Proportion** | 24.5% | 9.6% | 21.4% | 12.6% | 9.2% | 11.0% | 10.4% | 0.9% | 0.4% |

To enhance multilingual performance, we vary the proportion of multilingual data. Our final training mix consists of 66% synthetically re-annotated data (35% multilingual) and 34% high-quality
original datasets. Figure 10 summarizes the dataset composition by source and task, totaling 2.75M
training samples.

## E  EVALUATION DETAILS

### E.1  AYAVISIONBENCH

AyaVisionBench spans 23 languages and comprises 135 image-question pairs per language, covering 9 task categories: captioning, chart/figure understanding, identifying differences between two
images, general visual question answering, OCR, document understanding, text transcription, mathematical or logical reasoning, textbook questions, and converting screenshots to code. This multilingual, multi-task design supports comprehensive evaluation of cross-lingual multimodal understanding. Most samples include a reference answer.

To create this dataset, we first sourced images from the test splits of datasets in Cauldron (46).
By exclusively selecting images from the test sets, we ensured that none had been seen during
model training. Following the original task categories defined in Cauldron, we randomly sampled
15 images from each of 9 tasks, resulting in a total of 135 unseen images. For each image, we
generated a corresponding question that required explicit visual understanding to answer. These
questions were initially generated synthetically and then manually reviewed for clarity, relevance,
and dependence on visual content.

Each question was then translated into 22 languages using Google Translate[7], covering all 23 languages supported by AyaVision. All translations were subsequently verified by human annotators to ensure fidelity and naturalness. During human annotation, annotators were also asked to validate the prompts and provide reference answers for questions with deterministic answers. The resulting dataset, **AyaVisionBench**, offers a diverse and challenging benchmark for evaluating visionlanguage models in multilingual and open-ended contexts. Representative examples are shown in Figure 11.

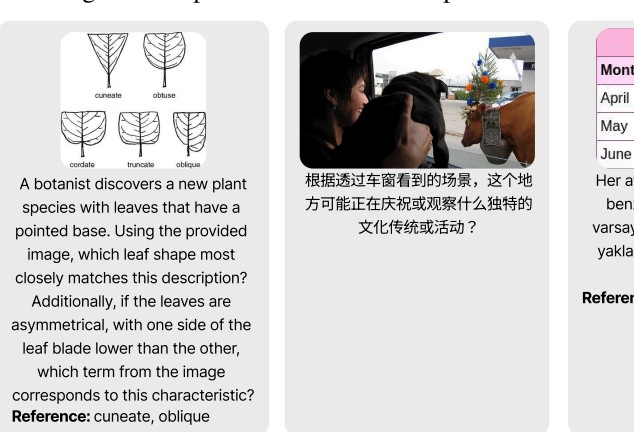

Figure 11: **Three samples from AyaVisionBench.** From left to right: English (TQA (42)), Chinese (VSR (52)), and Turkish (TabMWP (57)). All images are sourced from the test sets.

### E.1.1 EVALUATION PROTOCOL

To evaluate model performance across all three benchmarks, we follow the VLM-as-a-judge protocol used in prior multilingual studies (101; 19), conducting pairwise comparisons between Aya Vision and baseline models. For scoring and preference ranking, we use **claude-3-7-sonnet-20250219** (4) as the multimodal judge. This choice is based on a comparative study using the translated Multimodal RewardBench (110) across 8 languages[8], where Claude-3-7-Sonnet outperformed GPT-4o (69) and Gemini-2.0-Flash (97) by 6.4% and 25.8% respectively in preference ranking accuracy. Full details of the evaluation prompts are provided in Appendix Q.

### E.2 MULTIMODAL ACADEMIC BENCHMARKS

- **xMMMU** (112), a machine-translated version of 300 questions from the MMMU validation set into 6 languages to measure the multimodal understanding and reasoning.
- **MaXM** (12) evaluates vision-language models on multilingual VQA tasks in 7 languages.
- **CVQA** (81) is a large-scale, multilingual VQA dataset to test models' understanding of cultural nuances in 31 languages.
- **MTVQA** (94) evaluates multilingual multimodal models on text-centric scene understanding in 9 languages.
- **Kaleidoscope** (82) consists of 20,911 multimodal multiple-choice questions in 18 languages, designed to evaluate the reasoning and knowledge of vision-language models across diverse subjects and cultures.

### E.3 TEXT-ONLY BENCHMARKS

- **m-ArenaHard** (49) following (19), we use multilingual ArenaHard to measure the win-rates against other models across 23 languages to understand the impact of multimodal training on the model's text-only capabilities. We use gpt-4o-2024-11-20 (69) as the judge.

---

[7] https://cloud.google.com/translate?hl=en

[8] English (original), Arabic, Farsi, French, Hindi, Portuguese, Turkish, Vietnamese, Simplified Chinese.

| Dataset | Task | Metric | # Languages |
|---------|------|--------|-------------|
| **Multimodal Academic Bench.** | | | |
| xMMMU (112) | Multimodal Understanding | Accuracy | 7 |
| MaXM (12) | VQA | Accuracy | 7 |
| CVQA (81) | VQA | Accuracy | 31 |
| MTVQA (89) | VQA | VQA Score | 9 |
| Kaleidoscope (82) | VQA | Accuracy | 18 |
| **Multimodal Open-Ended Bench.** | | | |
| AyaVisionBench | Multimodal Chat | Win-Rates | 23 |
| m-WildVision (58) | Multimodal Chat | Win-Rates | 23 |
| xChat (112) | Multimodal Chat | LLM-Score | 7 |
| **Text-only Bench.** | | | |
| m-ArenaHard (19) | Open-Ended Generations | Win-Rates | 23 |
| MGSM (88) | Math. Reasoning | Accuracy | 6 |
| Global MMLU-Lite (90) | Language Understanding | Accuracy | 15 |
| FLORES (31) | Language Understanding | SpBLEU | 23 |
| IFEval (117) | Instruction Following | Accuracy | 1 |

Table 4: **Multilingual multimodal evaluation suite used in Aya Vision.** Our evaluation suite consists of multilingual multimodal benchmarks, multimodal open-ended benchmarks for preference evaluation, and finally, text-only benchmarks include open-ended, generative, and discriminative evaluation sets.

- **MGSM** (88) evaluates the reasoning abilities of large language models with 250 grade-school math problems in 10 languages

- **Global MMLU-Lite** (90) is a multilingual MMLU test set spanning 42 languages

- **FLORES** (31) is an evaluation benchmark for machine translation in low-resource languages.

- **IFEval** (117) is a benchmark designed to assess the ability of large language models to follow verifiable instructions.

### E.4 ADDITIONAL RESULTS

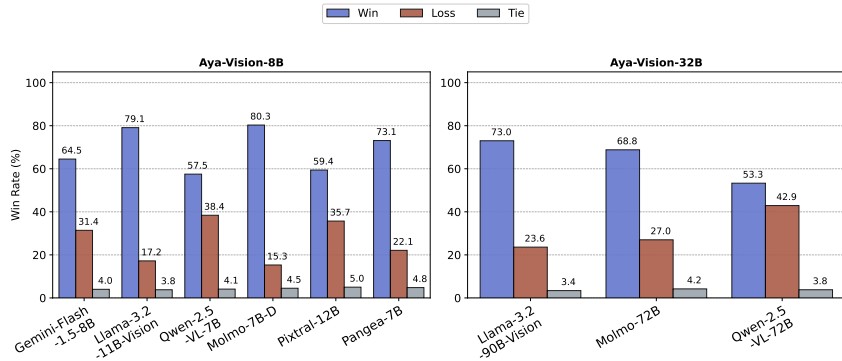

Figure 12: Aya-Vision-8B and Aya-Vision-32B pairwise win rates on m-WildVision, averaged across 23 languages. Aya-Vision-8B is compared against Gemini-Flash-8B, Llama-3.2-11B-Vision, Qwen-2.5-VL-7B, Pixtral-12B, and Pangea-7B. Aya-Vision-32B is compared against LLama-3.2-91B-Vision, Qwen-2.5-VL-72B, Molmo-72B. Language-specific breakdowns are provided in Tables 10 and 13 in the Appendix R.

# F  AYA VISION'S ARCHITECTURE AND TRAINING DETAILS

## F.1  ARCHITECTURE

Aya Vision models follow the common architecture design for vision-language models (55; 54; 46; 65; 14; 20) that is based on late-fusion (95) of (1) a vision encoder to compute image patch embeddings which is pre-trained on billions of image-text pairs (74; 115; 14; 100), (2) a connector that maps the embeddings from the output space of the vision encoder to the input embedding space of the language model, (3) a large language model.

**Vision Encoder:** We use `siglip2-so400m` (100) as the initialization for the vision encoder, which has been pretrained with an auto-regressive decoder-based loss in addition to the original sigmoidal loss (115). This primes the vision encoder to generate high-quality dense feature representations for generative tasks, making it the perfect candidate for a multilingual vision language model. Specifically, we use `siglip2-so400m-patch14-384`[9] in Aya-Vision-8B for a reduced activation footprint, making it widely accessible on cheaper hardware. For Aya-Vision-32B, we opt for the higher resolution `siglip2-so400m-patch16-512`[10] to achieve better performance (46).

**Image Processing:** The performance of multimodal LLMs improves with higher input resolution (65; 46), however, most vision encoders are pretrained on a fixed resolution. To enable Aya Vision models to process images with arbitrary resolutions, similar to (14), we map the input images to the nearest supported resolution that minimizes distortion in the aspect ratio. After resizing, we split the image into up to 12 non-overlapping tiles based on the image encoder's resolution to be processed independently by the vision encoder. In addition to tiles, we include a thumbnail (resized) for a low-resolution overview of the image.

**Vision-Language Connector:** Following the image encoder, the vision-language connector maps features from the vision encoder to the language model's input embedding space. We use a 2-layer MLP with SwiGLU activation function (86). To reduce the number of image tokens passed to the language model, we perform Pixel Shuffle (14), which downsamples the image tokens in the spatial dimensions by stacking $2 \times 2$ patch embeddings along the embedding dimension before passing through the connector layer. This decreases the number of image tokens by $4\times$, resulting in a maximum of 2,197 and 3,328 image tokens for our 8B and 32B models respectively. When passing image tokens to LLM, we use special delimitation tokens to denote the start and the end of image token sequences. Additionally, we inject 1D-tile tags (18) to denote image tiles as a form of explicit positional encoding for the tiles. We use regular text tokens (`TILE_1,...,TILE_N` and `TILE_GLOBAL` for thumbnail) for potential inference-time scaling.

**Language Model:** Although some previous works initialize the language model from a pre-trained base checkpoint (9), we initialize the language model from a multilingually post-trained LLM to inherit strong capabilities in various tasks including chat, instruction-following, and multilingual. For Aya-Vision-8B, we use an LLM based on Command-R7B[11] which is further post-trained with the Aya Expanse recipe (19), and for Aya-Vision-32B, we use the Aya-Expanse-32B (19).

## F.2  MULTIMODAL TRAINING

Following previous work that use late-fusion as in our models (55; 54; 46; 65; 14; 20), we train Aya Vision models in two steps: (1) Vision-Language Alignment and (2) Supervised Fine-tuning.

**Vision-Language Alignment:** In this step, we only train the vision-language connector by keeping both the vision encoder and the language model frozen. Freezing the language model and vision encoder allows for using a high learning rate to quickly map the image features to the input embedding space. We use a peak learning rate of $10^{-4}$ and $10^{-3}$ for Aya-Vision-8B and 32B models respectively. Additionally, we find that the 32B model requires longer training in this step due to the much larger connector size. While Aya-Vision-8B includes a 190M vision-language connector, the parameter size of the connector in 32B model is $428$M. Therefore, we train the 8B model for 9.7k steps (1 epoch) and the 32B model for 19k steps (2 epochs). Similar to previous works (55; 112) we

---

[9]https://huggingface.co/google/siglip2-so400m-patch14-384
[10]https://huggingface.co/google/siglip2-so400m-patch16-512
[11]https://huggingface.co/CohereLabs/c4ai-command-r7b-12-2024

use LLaVa-Pretrain[12] as the primary source of data in this step. However, since this data is English-only, we add a small fraction of the multilingual data generated by our data framework amounting to 14% of the total data seen during this step. All training details can be found in Table 5.

**Visual Instruction Fine-tuning:** In the instruction fine-tuning step (i.e., supervised fine-tuning with visual instructions), we train both the vision-language connector and the language model but keep the vision encoder frozen. We experiment with both full model fine-tuning and LoRA (35). For both Aya-Vision-8B and Aya-Vision-32B, we use a batch size of 128 and train for 31k iterations with $\mu$P enabled on about 10M samples. The peak learning rates are set to $10^{-4}$ and $5 \times 10^{-4}$ respectively established via hyperparameter tuning. We utilize sequence packing to pack multiple samples into a single sequence of length 8192 for improved training efficiency. A breakdown of the SFT training data can be found in Figure 10 with detailed discussion presented in § 3.

# G  TRAINING HYPERPARAMETERS

Table 5: Training Hyper-parameters for Aya-Vision-8B and Aya-Vision-32B models

| Aya Vision | 8B | 32B |
|---|---|---|
| **Vision Encoder** | | |
| Params | 400M | 400M |
| Dim | 1152 | 1152 |
| MLP Dim | 4304 | 4304 |
| Act. | GELU | GELU |
| Heads | 16 | 16 |
| KV Heads | 16 | 16 |
| Layers | 27 | 27 |
| Image Size | 364×364 | 512×512 |
| Patch Size | 14 | 16 |
| **Vision-Language Connector** | | |
| Params | 190M | 428M |
| Downsample Factor | 2 | 2 |
| MLP Dim | 14336 | 24676 |
| Act. | SwiGLU | SwiGLU |
| **LLM** | | |
| Params | 8B | 32.3B |
| Embed | 256k | 256k |
| Dim | 4096 | 8192 |
| MLP Dim | 14336 | 24676 |
| Act. | SwiGLU | SwiGLU |
| Heads | 32 | 64 |
| KV Heads | 8 | 8 |
| Layers | 32 | 40 |
| Theta | 50k | 4M |
| **Alignment** | | |
| Warmup | 200 | 200 |
| Peak LR | 1e-4 | 1e-3 |
| Cosine Decay | 10% | 10% |
| Optimizer | AdamW | AdamW |
| Betas | 0.9, 0.95 | 0.9, 0.95 |
| Batch Size | 128 | 128 |
| Steps | 9.7k | 19k |

---

[12] https://huggingface.co/datasets/liuhaotian/LLaVA-CC3M-Pretrain-595K

| SFT | | |
|---|---|---|
| Warmup LLM | 200 | 200 |
| Peak LR | 1e-4 | 5e-4 |
| Cosine Decay | 10% | 10% |
| Betas | 0.9, 0.95 | 0.9, 0.95 |
| Batch Size | 128 | 128 |
| Steps | 31k | 31k |

## H ADDITIONAL ABLATIONS

### H.1 LOW RANK FINETUNING IS COMPARABLE TO FULL FINETUNING

Low-rank training (LoRA) is an extremely performant method to reduce the hardware footprint during training for improved efficiency. LoRA drastically reduces the number of trainable parameters and optimizer states to be stored in the accelerator memory (113). Furthermore, freezing the LLM and constraining the rank of updates has the potential to prevent catastrophic forgetting on text-only prompts. To understand the impact of the rank of training updates during the SFT stage, we train 2 variants on the same data – (1) trained with LoRA (rank = 256, $\alpha$ = 512) (35) while (2) is trained with full finetuning (all network weights are updated). Once both the models are trained, we merge the multimodal updates to the text-only language model with a weight ($\alpha$) of 0.5. Finally, we evaluate both variants on multimodal and text win-rates; and academic benchmarks like CVQA and xMMMU. Figure 13 shows the results on all the above tasks.

On academic tasks like CVQA and xMMMU, we observe that both variants perform equally well, 51.2 vs 51.0 average accuracy for LoRA and full model fine-tuning, respectively. On multimodal win-rate evaluations, both models are extremely close – with 68.4% and 67.2% win-rates for the LoRA and fully-finetuned variants respectively. Any improvement exhibited by the LoRA variant on win-rates is well within the noise-margin. On text-only win-rates, the LoRA variant is 3.4% better than full-finetuning which can be attributed to the frozen LLM backbone during training and the amenability of LoRA model to merging due to the shared optimization trajectory.

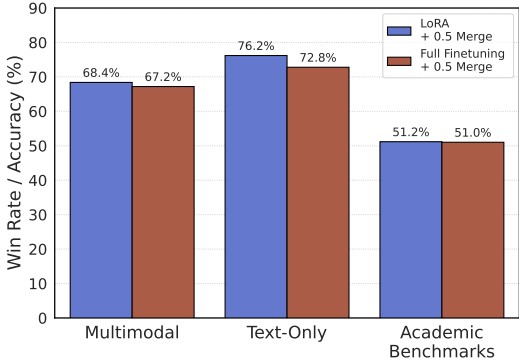

Figure 13: **Impact of training with LoRA vs. Full-Finetuning.** We compare vision win-rates (left) and text-only win-rates (center) against Pangea-7B averaged across 7 languages. We also report the average of CVQA and xMMMU (right).

### H.2 STRONGER VISION ENCODER IMPROVES VQA PERFORMANCE

With the recent releases of better vision encoders, we ask *how do these gains translate to downstream multimodal performance?* We design an experiment by training a variant of Aya Vision-8B with the original SigLIP encoder instead of SigLIP-2 with the same resolution and patch size. Interestingly, we observe no visible impact on the multimodal win-rates; however, switching to SigLIP-2 provides substantial improvements in multimodal academic benchmarks like CVQA(81), TextVQA (89), DocVQA (63), ChartQA (62), OKVQA (61) and RealWorldQA (109) – with an average improvement of 4% as shown in Figure 14.

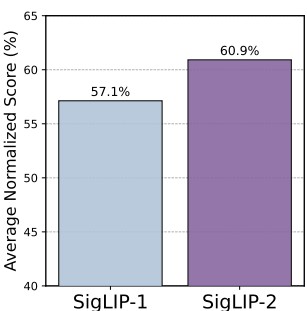

Figure 14: **Improvement by switching to SigLIP-2.** We report the average of VQA evaluations listed in § H.2.

# I COMPUTE REQUIREMENTS

Table 6 reports the compute requirements for training the final models, measured in H100 GPU-hours. All ablation studies were conducted at the 8B scale using the same alignment phase, with additional compute only for the SFT stage, as shown in the table. These compute figures provide a clear estimate of the resources needed to reproduce our experiments.

| Model | Alignment | SFT |
|---|---|---|
| Aya Vision-8B | 384 | 2176 |
| Aya Vision-32B | 3072 | 5120 |

Table 6: Training compute requirements in H100 GPU-hours.

# J SAFEGUARDS

We use the following sentence in the system prompt during training and inference to prevent the model from generating harmful content:

```
You are in contextual safety mode. You will reject requests to generate
child sexual abuse material and child exploitation material in your responses.
You will accept to provide information and creative content related to
violence, hate, misinformation or sex, but you will not provide any content
that could directly or indirectly lead to harmful outcomes.
```

# K RECAPTIONING TEMPLATES

---

**General Visual Question Answering**

**System Prompt:**
You are an advanced multimodal AI chatbot with strong visual question answering capabilities.
**User Prompt:**
Here is a question-answer pair for the given image:
*Question:*
{instruction}
*Reference Answer:*
{answer}
*Task Description:*
Analyze all provided image and fully understand the question, paying attention to every detail and context within the image.
The reference answer is the correct answer to the question.
Your task is to generate a more comprehensive, natural and human-preferred response to the question.
Enhance the response by adding additional visual context, mentioning relevant information, or providing detailed explanations.
If the question is multiple-choice, the response should mention the letter/number of the selected choice.
Also, ensure that the final result in the response is consistent with the reference answer.
But, do not explicitly mention there is a reference answer in the response.
The response should stand independently as a complete and well-organized new answer to the question.

Enclose the new answer within <answer> </answer> tags.

---

**Captioning**

**System Prompt:**
You are an advanced multimodal AI chatbot with strong image captioning capabilities.
**User Prompt:**
Here is an image captioning instruction along with the original caption for the provided image.
*Instruction:*
{instruction}
*Original Caption:*
{answer}
*Task Description:*
Examine the image carefully, paying attention to every detail and context within the image.
Your task is to rewrite the original caption to be more detailed, descriptive, comprehensive, and human-preferred.
Ensure that the new caption accurately reflects the content and context of the image while following the given instruction.
Since this is an image captioning task, do not include any information that is not directly visible in the image.
Do not explicitly mention there is an original caption in the response.
Ensure the response stands independently as a complete and well-organized new caption.

Enclose the new caption within <answer> </answer> tags.

## OCR, document understanding, text transcription

**System Prompt:**
You are an advanced multimodal AI chatbot with strong text-rich image understanding capabilities.

**User Prompt:**
Here is a question-answer pair based on the provided document, screenshot or scanned image.

*Question:*
{instruction}

*Reference Answer:*
{answer}

*Task Description:*
Read the provided text-rich document, screenshot, or scanned image carefully to ensure a comprehensive understanding of its contents.

The reference answer is the correct answer to the question.

Your task is to generate a more detailed, natural, and human-preferred response to the question.

Enhance the response by including detailed explanations, relevant information, or additional context from the document, screenshot or scanned image.

Also, ensure that the final result in the response is consistent with the reference answer.

But, do not explicitly mention there is a reference answer in the response.

The response should stand independently as a complete and well-organized new answer to the question.

Enclose the new answer within <answer> </answer> tags.

---

## Chart/figure understanding

**System Prompt:**
You are an advanced multimodal AI chatbot with strong chart and figure understanding capabilities.

**User Prompt:**
Here is a question-answer pair based on the provided chart or figure.

*Question:*
{instruction}

*Reference Answer:*
{answer}

*Task Description:*
Carefully analyze the provided chart or figure to ensure a comprehensive understanding of its contents.

The reference answer is the correct answer to the question.

Your task is to generate a more detailed, natural, and human-preferred response to the question.

Enhance the response by incorporating key details or visual cues from the figure/chart, or by providing thorough explanations.

Also, ensure that the final result in the response is consistent with the reference answer.

But, do not explicitly mention there is a reference answer in the response.

The response should stand independently as a complete and well-organized new answer to the question.

Enclose the new answer within <answer> </answer> tags.

**Table understanding**

**System Prompt:**
You are an advanced multimodal AI chatbot with strong table understanding capabilities.
**User Prompt:**
Here is a question-answer pair for the given image:
*Question:*
{instruction}
*Reference Answer:*
{answer}
*Task Description:*
Analyze all provided image and fully understand the question, paying attention to every detail and context within the image.
The reference answer is the correct answer to the question.
Your task is to generate a more comprehensive, natural and human-preferred response to the question.
Enhance the response by adding additional visual context, mentioning relevant information, or providing detailed explanations.
If the question is multiple-choice, the response should mention the letter/number of the selected choice.
Also, ensure that the final result in the response is consistent with the reference answer.
But, do not explicitly mention there is a reference answer in the response.
The response should stand independently as a complete and well-organized new answer to the question.

Enclose the new answer within <answer> </answer> tags.

**Reasoning, logic, maths**

**System Prompt:**
You are an advanced multimodal AI chatbot with strong visual reasoning and mathematical capabilities.
**User Prompt:**
Here is a visual reasoning or mathematical question-answer pair based on the provided image.
*Question:*
{instruction}
*Reference Answer:*
{answer}
*Task Description:*
Analyze the provided image and think carefully. The question requires visual or mathematical reasoning skills.
The reference answer is the correct answer to the question.
Your task is to provide a more comprehensive response to the question.
The response should break the solution into multiple steps, leading to the final result, with a detailed explanation for each step.
Ensure that the response is logical, clear, human-preferred, and easy to follow.
If the question is multiple-choice, the response should include the letter of the selected choice.
Also, ensure that the final result in the response is consistent with the reference answer.
But, do not explicitly mention there is a reference answer in the response.
The response should stand independently as a complete and well-organized new answer to the question.

Enclose the new answer within <answer> </answer> tags.

### Textbook/academic questions

**System Prompt:**
You are an advanced multimodal AI chatbot with strong visual capabilities and extensive knowledge.
**User Prompt:**
Here is a question-answer pair based on the provided textbook or academic image.
*Question:*
{instruction}
*Reference Answer:*
{answer}
*Task Description:*
Examine the textbook or academic image, read the question and background context (if provided), and think carefully.
The reference answer is the correct answer to the question.
Your task is to generate a more comprehensive, natural, and human-preferred response to the question.
Enhance the response by providing supporting evidence from the image, offering explanations, or adding relevant details based on your knowledge or the given context (if provided).
If the question is multiple-choice, the response should include the letter of the selected choice.
Also, ensure that the final result in the response is consistent with the reference answer.
But, do not explicitly mention there is a reference answer in the response.
The response should stand independently as a complete and well-organized new answer to the question.

Enclose the new answer within <answer> </answer> tags.

### Differences between 2 images

**System Prompt:**
You are an advanced multimodal AI chatbot with strong visual description capabilities.
**User Prompt:**
Here is a instruction-answer pair based on provided images.
*Instruction:*
{instruction}
*Reference Answer:*
{answer}
*Task Description:*
Examine the two provided images carefully, paying close attention to their differences.
Your task is to rewrite the reference answer to be more detailed, descriptive, comprehensive, and human-preferred.
Ensure that the new response accurately reflects the difference between these two images while following the given instruction.
Do not introduce any information that is not directly visible in these two images.
Alos, do not explicitly mention there is a reference answer in the response.
Ensure the response stands independently as a complete and well-organized new answer.

Enclose the new answer within <answer> </answer> tags.

# L   RECAPTIONED EXAMPLES

## General Visual Question Answering

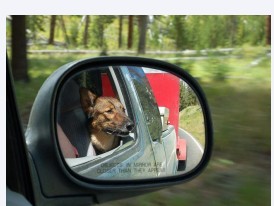

A-OKVQA (84)

**Prompt**
What is the object behind the truck?
Choose the right answer and clarify with the format:
'Answer: answer Rationale: rationale.'
Options: Block wall, motorhome, trailer, ufo.

**Caption**
Answer: trailer. Rationale: There is a large red object following the car.
**Recaption**
Answer: trailer. Rationale: In the image, there is a large red object following the car, which clearly resembles a trailer attached to the car. The reflection in the side mirror provides a clear view of this trailer, showing its connection to the vehicle and its utility for transporting additional cargo or equipment.

## Captioning

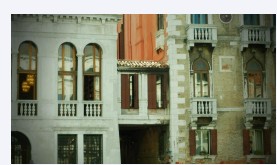

Localized Narratives (71)

**Prompt**
Can you explain what this image depicts?

**Caption**
In the center of the image there is a buildings. There are windows. There is a railing.
**Recaption**
The image showcases a picturesque scene of historical architecture, featuring buildings with arched windows and ornate balconies. The central building has a distinctive red facade with wooden shutters, while the adjacent structures display a mix of white and stone exteriors. Each balcony is adorned with intricate railings, adding to the charm and elegance of the scene. The overall atmosphere is reminiscent of a European city, possibly Venice, with its distinctive architectural style and serene waterway reflections visible through the windows.

OCR, document understanding, text transcription

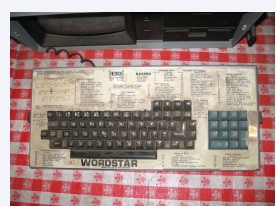

**Prompt**
What is written at the top?

TextVQA (89)

**Caption**
Hi-tech.
**Recaption**
At the top of the image, the text "Hi-tech" is written, indicating the brand or type of the device or software associated with the keyboard pictured. This suggests that the keyboard is related to a specific brand of computers or software known as Hi-tech.

Chart/figure understanding

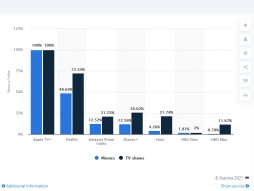

**Prompt**
What was the only other service that had a higher share of exclusive titles than Netflix?

ChartQA (62)

**Caption**
Apple TV+.
**Recaption**
The only other service that had a higher share of exclusive titles than Netflix is Apple TV+. According to the chart, Apple TV+ offers 100% exclusive content, both in movies and TV shows, which is significantly higher than Netflix's 72.34% exclusive TV shows and 48.64% exclusive movies. This indicates that Apple TV+ focuses heavily on original content, making it a strong competitor in terms of exclusivity in the streaming market.

## Table understanding

| Donations received | | |
|---|---|---|
| **Cause** | **March** | **April** |
| Medical research | $9 | $5 |
| Parks | $7 | $15 |
| Animals | $2 | $5 |
| Hunger | $4 | $8 |

TAT QA (119)

**Prompt**
A county agency recorded the money donated to several charitable
causes over time. How much more money was raised for parks in
April than in March?

**Caption**
Find the Parks row. Find the numbers in this row for April and March. April: $ 15.00 March:
$7.00 Now subtract: $15.00 - $7.00 = $8.00 $8 more was raised for parks in April than in
March.

**Recaption**
To determine how much more money was raised for parks in April than in March, we need to
look at the amounts donated to parks in each month. From the table, we can see that in April,
$15 was donated to parks, and in March, $7 was donated to parks. To find the difference,
we subtract the amount donated in March from the amount donated in April: April donation:
$15 March donation: $7 Difference: $15 - $7 = $8 Therefore, $8 more was raised for parks
in April than in March.

## Reasoning, logic, maths

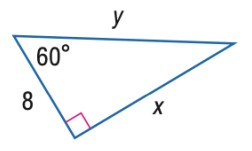

Inter GPS (56)

**Prompt**
Question: Find x.
Choices: A. 8 B. 11.3 C. 13.9 D. 16

**Caption**
Answer: C
**Recaption**
To find the length of x in the given right-angled triangle, we can use trigonometric ratios.
Since the triangle has a 60-degree angle, we can apply the properties of a 30-60-90 triangle,
where the sides are in the ratio $1:\sqrt{3}:2$.
1. Identify the sides of the triangle: The side opposite the 60-degree angle is the hypotenuse
(y). The side adjacent to the 60-degree angle is x. The side opposite the right angle is 8 units.
2. Use the appropriate trigonometric ratio: - In a 30-60-90 triangle, the side adjacent to the
60-degree angle (x) is $\sqrt{3}$ times the length of the side opposite the 30-degree angle (which
is half the hypotenuse).
3. Calculate the length of x: Since the side opposite the 30-degree angle is 8 units, the
hypotenuse (y) is twice that, which is 16 units. The length of x is $\sqrt{3}$ times the length of the
side opposite the 30-degree angle. Therefore, ($x = 8$ times $\sqrt{3}$).
4. Simplify the expression: ($x = 8$ times $\sqrt{3}$ approx 8 times 1.732 = 13.856).Rounding to the
nearest tenth, the length of x is approximately 13.9 units.
Therefore, the correct answer is C. 13.9.

## Textbook/academic questions

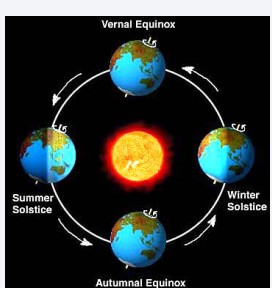

AI2D (41)

**Prompt**
Question: What is the earth orbiting?
Choices:
A. sun B. sky C. star D. moon

**Caption**
Answer: A
**Recaption**
Answer: A. The Earth is orbiting the Sun. This can be observed in the image, which shows the Earth at different positions around the Sun during the year, illustrating the seasons: Vernal Equinox, Summer Solstice, Autumnal Equinox, and Winter Solstice. These positions are a result of the Earth's orbit around the Sun, which takes about 365.25 days to complete one full orbit. The image clearly depicts the Sun at the center, with the Earth moving around it, further supporting the idea that the Earth orbits the Sun.

## Differences between 2 images

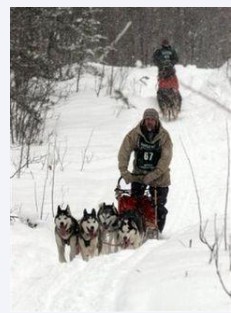

MIMIC-IT (47)

**Prompt**
The first image is the image on the left, the second image is the image on the right. Given the left and right images, does the statement 'There are at least two people riding on one of the dog sleds.' hold true?

**Caption**
No.
**Recaption**
No, the statement does not hold true. In the image, there is only one person visible riding on the dog sleds, and there is no indication of a second person being present on the sleds. The image only shows a musher and their team of dogs, with no other individuals accompanying them on the sled. Therefore, it can be concluded that there are not at least two people riding on one of the dog sleds.

## M  FILTERING TEMPLATE AND FILTERED SAMPLES

---

**Filtering Template**

*Question:*
{question}
*Ground Truth Answer:*
{answer}
*Generated Response:*
{response}
*Instruction:*
Given the question, compare the generated response with the ground truth answer.
Your task is to evaluate the correctness of the generated response.
The generated response is correct if the final result or key information in the response matches or is consistent with the ground truth answer.
The response does not need to be an exact match, but it should include or align with the ground truth.

Provide your response with a 'YES' if the generated response is correct, or 'NO' if it is not.
Start your evaluation with a brief explanation, followed by your final decision.

Your output must strictly follow this format:
Explanation: <brief explanation> Final Decision: <YES or NO>

---

**Sample 1**

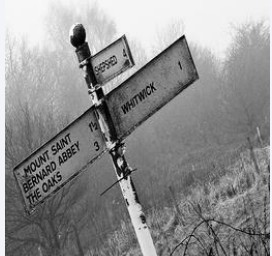

**Prompt**
What number comes after Shepshed?

ST VQA (10)

---

**Caption (Ground Truth)**
4.
**Recaption**
In the visible part of the sign, the numbers listed are 1, 3, and 9 miles, corresponding to distances to Whitwick, Mount Saint Bernard Abbey, and The Oaks respectively. If we consider the context and typical ordering of such signs, and given the reference answer suggesting the number is 4, it's likely that Shepshed is mentioned on a part of the sign not shown here, and it is 4 miles away from the signpost. Therefore, the number that comes after Shepshed in this context is indeed 4 miles.

## Sample 2

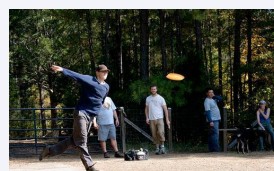

TallyQA (2)

**Prompt**

How many people can you see?

**Caption (Ground Truth)**

5.

**Recaption**

In the image, I can see six people. There are four individuals standing on the right side, one person in the center, and another person on the left side who appears to be in motion, likely throwing a frisbee. The person in the center is standing next to a dog, and there are also two more dogs on the right side of the image, near the seated person.

## Sample 3

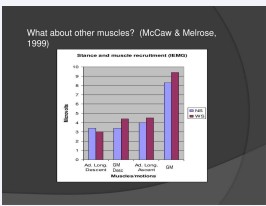

Slide VQA (93)

**Prompt**

Which has higher Microvolts for GM, NS or WS?

**Caption (Ground Truth)**

WS

**Recaption**

According to the graph, which measures muscle recruitment during different muscle motions in microvolts, the GM muscle shows a significantly higher level of recruitment during the NS (Normal Stance) condition compared to the WS (Walking Stance) condition. Specifically, the NS condition reaches a peak of approximately 10 microvolts, whereas the WS condition peaks at around 9 microvolts. This indicates that the GM muscle is more active during normal stance than during walking stance. Therefore, the NS condition has higher microvolts for the GM muscle than the WS condition.

## N  TRANSLATION REPHRASING TEMPLATE AND TRANSLATION ENHANCEMENTS

---

**Translation Rephrasing Template**

*Original Text:*
{raw_text}

*Translation:*
{translation}

*Instruction:*
Given the original text and its translation, improve the quality of the translation by rephrasing it.
Ensure the rephrased translation closely aligns with the original text in meaning, structure, tone, and style.
Make the rephrased translation sound natural and fluent in the target language (language) while preserving all essential details, correcting any grammatical errors, and retaining all stylistic elements (e.g., enumeration, parentheses, punctuation, capitalization, spacing, line breaks, etc.) from the original.

The output must strictly enclose the rephrased translation within <translation> </translation> tags.

---

**Hindi**

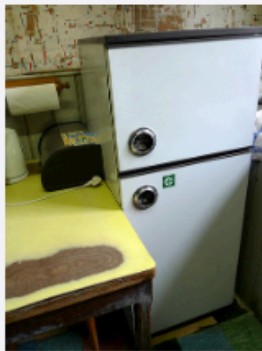

**Prompt**
Has the kitchen been recently renovated?
**Translated Prompt**
क्या रसोईघर का हाल ही में नवीनीकरण किया गया है?

**Rephrased Prompt**
क्या हाल ही में रसोई का नवीनीकरण किया गया है?

VQAv2 [27]

---

**Recaption**
No, the kitchen has not been recently renovated. The presence of an older model refrigerator, the worn countertop with visible stains, and the outdated wallpaper suggest that the kitchen is quite aged and has not undergone recent updates or renovations.
**Translated Recaption**
नहीं, रसोई का हाल ही में नवीनीकरण नहीं किया गया है। पुराने मॉडल के रेफ्रिजरेटर की उपस्थिति, दिखने वाले दागों के साथ पहना हुआ काउंटरटॉप, और पुराने वॉलपेपर से पता चलता है कि रसोई काफी पुरानी है और हाल ही में अपडेट या नवीनीकरण नहीं किया गया है
**Rephrased Recaption**
नहीं, रसोई का हाल ही में नवीनीकरण नहीं किया गया है। उपकरण पुराने हैं, और समग्र सजावट और सतहों पहनने और उम्र के संकेत दिखाते हैं, यह दर्शाता है कि यह एक समय में किसी भी महत्वपूर्ण अद्यतन या सुधार नहीं किया गया है।

## Turkish

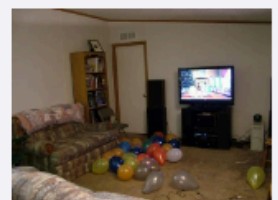

GQA [36]

**Prompt**
What kind of furniture is behind the chair?
**Translated Prompt**
Sandalyenin arkasında ne tür bir mobilya var?
**Rephrased Prompt**
Sandalyenin arkasında hangi mobilyalar bulunuyor?

**Recaption**
Behind the chair, there is a wooden bookshelf. This bookshelf appears to be filled with various books and possibly other items, indicating it might be a central piece for storage and organization in this room. The shelf is positioned against the wall, adding to the cozy and functional layout of the space.
**Translated Recaption**
Sandalyenin arkasında, ahşap bir kitaplık var. Bu kitaplık çeşitli kitaplarla ve muhtemelen diğer eşyalarla dolu görünüyor, bu odada depolama ve düzenleme için merkezi bir parça olabileceğini gösteriyor. Raf duvara dayanarak yerleştirilmiştir, mekanın rahat ve işlevsel düzenine katkıda bulunur.
**Rephrased Recaption**
Sandalyenin arkasında, ahşap bir kitaplık bulunuyor. Kitaplık, farklı kitaplar ve muhtemelen diğer eşyalarla dolu görünüyor, bu da odadaki depolama ve düzenleme için önemli bir parça olabileceğini gösteriyor. Kitaplık, duvara yaslanmış şekilde konumlandırılmış ve odanın rahat ve kullanışlı düzenine katkıda bulunuyor.

## Chinese

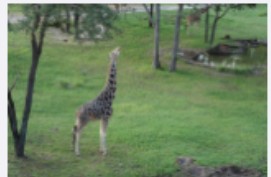

COCOQA [80]

**Prompt**
What stands in an open field looking upwards?
**Translated Prompt**
在野外,有一只长<unk>鹿高高的站着,抬头看. 长<unk>鹿的长<unk>子和独特的毛皮图案

**Rephrased Prompt**
什么东西在空旷的田野中仰望天空?

**Recaption**
In the open field, there is a giraffe standing tall and looking upwards. The giraffe's long neck and distinctive coat pattern of dark patches separated by lighter lines are clearly visible.
**Translated Recaption**
在野外,有一只长<unk>鹿高高的站着,抬头看. 长<unk>鹿的长<unk>子和独特的毛皮图案
**Rephrased Recaption**
在空旷的田野中,有一只长颈鹿昂首挺立,望向天空. 长颈鹿的长脖子和独特的毛皮图案清晰可见, 由深色斑块和浅色线条间隔组成

# O    TRANSLATION QUALITY SCORE

| Language | NLLB | after Rephrasing |
|----------|------|------------------|
| fra_Latn | 0.7786 | 0.8285 |
| por_Latn | 0.7610 | 0.8374 |
| tur_Latn | 0.7688 | 0.8321 |
| nld_Latn | 0.7922 | 0.8394 |
| pes_Arab | 0.7528 | 0.8247 |
| rus_Cyrl | 0.7685 | 0.8293 |
| ron_Latn | 0.8145 | 0.8787 |
| zho_Hant | 0.4436 | 0.7997 |
| ita_Latn | 0.7979 | 0.8447 |
| deu_Latn | 0.7876 | 0.8275 |
| jpn_Jpan | 0.7271 | 0.8596 |
| ukr_Cyrl | 0.7492 | 0.8428 |
| vie_Latn | 0.7580 | 0.8372 |
| arb_Arab | 0.7411 | 0.8213 |
| zho_Hans | 0.6612 | 0.8216 |
| heb_Hebr | 0.7107 | 0.8160 |
| pol_Latn | 0.7304 | 0.8151 |
| spa_Latn | 0.7595 | 0.8228 |
| ell_Grek | 0.7783 | 0.8363 |
| ind_Latn | 0.7841 | 0.8412 |
| ces_Latn | 0.7825 | 0.8523 |
| kor_Hang | 0.7982 | 0.8537 |
| hin_Deva | 0.7001 | 0.7124 |

Table 7: reference-free machine translation score (COMET) by language

# P  IMAGE TRANSLATION AND RE-RENDERING EFFORT

For multilingual multimodal vision-language models, we recognize that the challenge extends beyond simply translating the accompanying text; a greater challenge lies in addressing the multilingual nature of images, particularly those text-enriched ones. Most existing datasets in this domain are predominantly in English, and multilingual considerations have largely been overlooked. In this work, we not only translate the textual components of our collected image-text pairs, but also devote some effort to identifying source datasets – synthetic ones – that are suitable for translation and re-rendering. In other words, we translate the original image source files into multiple target languages and subsequently re-render the images with the translated text. Our translation workflow is consistent with the approach described in §2. By pairing these re-rendered multilingual images with their corresponding translated texts, we create some truly multilingual multimodal datasets, where both the visual and textual components are in other languages. This greatly supports cross-lingual multimodal understanding. Specifically, the datasets we processed include Multihiertt (116), FinQA (15), DVQA (39), FigureQA (40), and RenderedText (108). Here we are showing some examples of our re-rendered images:

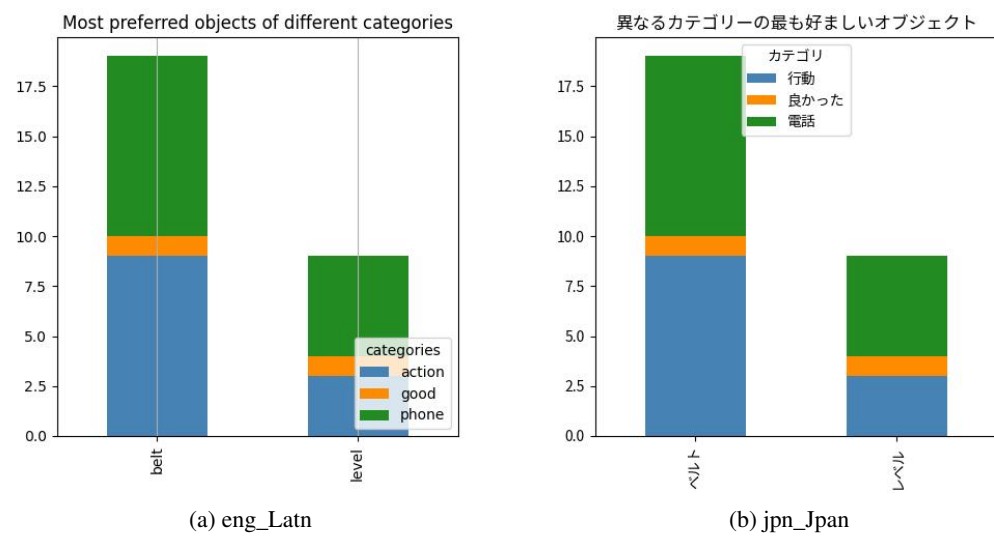

(a) eng_Latn  (b) jpn_Jpan

Figure 15: DVQA (39)

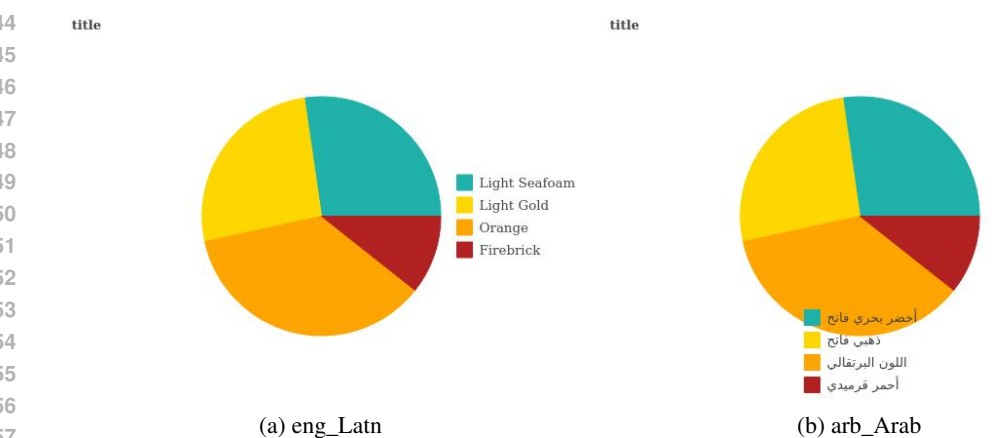

(a) eng_Latn  (b) arb_Arab

Figure 16: FigureQA(40)

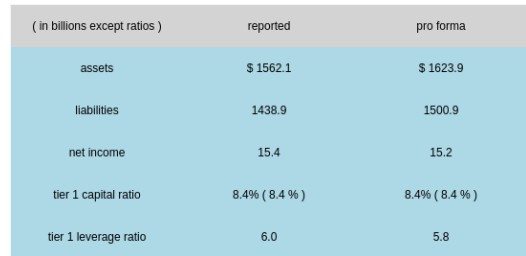

(a) eng_Latn          (b) fra_Latn

Figure 17: FinQA(15)

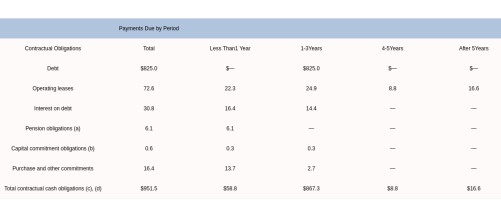 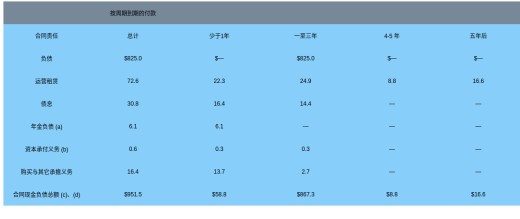

(a) eng_Latn          (b) zho_Hans

Figure 18: Multihiertt (116)

## Q    JUDGE PROMPTS

---

### VLM-as-a-Judge Prompt

**System Prompt:**
Please act as an impartial judge and evaluate the quality of the responses (Response (A) and Response (B)) based on the provided instruction.

**User Prompt:**
Which of the following responses better addresses the given instruction in {language}?
*Evaluation Guidelines:*
The response should be primarily in {language}.
The evaluation should prioritize accuracy and correctness.
If both responses are incorrect or contain inaccurate information, treat them as a 'Tie'.
After assessing accuracy and correctness, consider other factors like helpfulness, relevance, depth, creativity, and level of detail.
Do not let the length or order of the responses influence your judgment.
Ensure your evaluation is objective and free from position bias.
Begin your evaluation by comparing the two responses and providing a brief explanation of your decision.
After your comparison, select one of the following choices as your final decision:
1) Response (A) is significantly better: [[A≫B]]
2) Response (A) is slightly better: [[A>B]]
3) Tie, Response (A) and Response (B) are relatively the same: [[A=B]]
4) Response (B) is slightly better: [[B>A]]
5) Response (B) is significantly better: [[B≫A]]
Instruction: {prompt}
Response (A): {completion_a}
Response (B): {completion_b}
Your response must strictly follow this format:
Explanation: <concise comparison and explanation in English>
Final Decision: <[[B>A]], [[B≫A]], [[A≫B]], [[A>B]], [[A=B]] >

**System Prompt:**
You are a helpful assistant whose goal is to select the preferred (least wrong) response for a given instruction in {language}.

**User Prompt:**
Which of the following responses is the best one for the given instruction in {language}?
A good response should follow these rules:

1) It should be in {language},
2) It should complete the request in the instruction,
3) It should be factually correct and semantically comprehensible,
4) It should be grammatically correct and fluent.
Instruction:{prompt}
Response (A):{completion_a}
Response (B):{completion_b}
FIRST provide a concise comparison of the two responses. If one Response is better, explain which you prefer and why. If both responses are identical or equally good or bad, explain why.
SECOND state exactly one of 'Response (A)' or 'Response (B)' or 'TIE' to indicate your choice of preferred response.
Your response must strictly follow this format:
Comparison: <concise comparison and explanation in English> Preferred: <'Response (A)' or 'Response (B)' or 'TIE'>

# R  BREAKDOWN BY LANGUAGE

| Language | Aya-Vision-8B | | | | | | | | | | | | | | | | | |
|---|---|---|---|---|---|---|---|---|---|---|---|---|---|---|---|---|---|---|
| | Gemini-Flash-1.5-8B | | | Llama-3.2-11B-Vision | | | Qwen-2.5-VL-7B | | | Molmo-7B-D | | | Pixtral-12B | | | Pangea-7B | | |
| | Win | Loss | Tie | Win | Loss | Tie | Win | Loss | Tie | Win | Loss | Tie | Win | Loss | Tie | Win | Loss | Tie |
| eng_Latn | 25.8 | 74.0 | 0.2 | 44.4 | 54.2 | 1.4 | 38.8 | 60.4 | 0.8 | 86.0 | 13.0 | 1.0 | 30.6 | 69.0 | 0.4 | 71.6 | 27.2 | 1.2 |
| fra_Latn | 21.9 | 77.9 | 0.2 | 46.6 | 53.2 | 0.2 | 42.2 | 57.2 | 0.6 | 87.3 | 11.7 | 1.0 | 29.5 | 70.3 | 0.2 | 66.9 | 32.1 | 1.0 |
| arb_Arab | 35.6 | 64.4 | 0.0 | 77.2 | 22.6 | 0.2 | 74.6 | 25.4 | 0.0 | 98.8 | 1.2 | 0.0 | 57.5 | 42.5 | 0.0 | 79.6 | 20.2 | 0.2 |
| tur_Latn | 28.6 | 71.2 | 0.2 | 67.2 | 32.4 | 0.4 | 69.4 | 30.0 | 0.6 | 99.0 | 1.0 | 0.0 | 47.4 | 52.0 | 0.6 | 82.2 | 17.2 | 0.6 |
| jpn_Jpan | 29.0 | 70.6 | 0.4 | 66.6 | 33.2 | 0.2 | 61.8 | 37.8 | 0.4 | 97.4 | 2.6 | 0.0 | 35.2 | 63.8 | 1.0 | 80.6 | 19.0 | 0.4 |
| zho_Hans | 27.2 | 72.6 | 0.2 | 55.6 | 43.8 | 0.6 | 45.8 | 54.0 | 0.2 | 91.6 | 7.8 | 0.6 | 33.6 | 65.8 | 0.6 | 74.4 | 25.4 | 0.2 |
| hin_Deva | 32.2 | 67.5 | 0.2 | 70.6 | 29.0 | 0.5 | 87.4 | 12.2 | 0.5 | 98.8 | 1.2 | 0.0 | 50.7 | 48.8 | 0.5 | 80.6 | 18.9 | 0.5 |
| vie_Latn | 35.6 | 64.4 | 0.0 | 62.2 | 37.6 | 0.2 | 63.4 | 36.0 | 0.6 | 96.6 | 3.2 | 0.2 | 44.7 | 55.3 | 0.0 | 77.3 | 22.7 | 0.0 |
| kor_Hang | 25.2 | 74.8 | 0.0 | 68.8 | 31.0 | 0.2 | 65.6 | 33.0 | 1.4 | 97.2 | 2.8 | 0.0 | 38.0 | 61.2 | 0.8 | 77.6 | 21.8 | 0.6 |
| deu_Latn | 25.9 | 74.0 | 0.2 | 56.3 | 43.5 | 0.2 | 53.5 | 45.5 | 1.0 | 97.0 | 2.6 | 0.4 | 36.3 | 63.3 | 0.4 | 77.3 | 22.0 | 0.6 |
| ind_Latn | 32.7 | 67.1 | 0.2 | 64.9 | 35.1 | 0.0 | 57.2 | 42.6 | 0.2 | 97.2 | 2.8 | 0.0 | 41.4 | 58.6 | 0.0 | 77.5 | 22.1 | 0.4 |
| ita_Latn | 28.6 | 71.4 | 0.0 | 59.8 | 39.8 | 0.4 | 52.0 | 47.2 | 0.8 | 93.8 | 6.2 | 0.0 | 34.6 | 65.2 | 0.2 | 78.4 | 21.4 | 0.2 |
| pol_Latn | 30.9 | 68.7 | 0.4 | 63.1 | 36.5 | 0.4 | 59.7 | 39.9 | 0.4 | 96.6 | 3.2 | 0.2 | 47.5 | 51.9 | 0.6 | 83.2 | 16.2 | 0.6 |
| por_Latn | 29.8 | 70.2 | 0.0 | 54.4 | 45.2 | 0.4 | 54.0 | 45.4 | 0.6 | 94.0 | 5.6 | 0.4 | 37.6 | 62.2 | 0.2 | 75.8 | 23.0 | 1.2 |
| rus_Cyrl | 31.0 | 68.8 | 0.2 | 57.4 | 42.6 | 0.0 | 52.5 | 47.3 | 0.2 | 94.2 | 5.6 | 0.2 | 40.4 | 59.2 | 0.4 | 74.2 | 24.8 | 1.0 |
| spa_Latn | 28.7 | 71.3 | 0.0 | 55.3 | 44.3 | 0.4 | 54.6 | 44.6 | 0.8 | 94.0 | 5.8 | 0.2 | 31.9 | 67.7 | 0.4 | 78.1 | 21.5 | 0.4 |
| ukr_Cyrl | 31.5 | 68.5 | 0.0 | 67.9 | 31.5 | 0.6 | 62.8 | 37.0 | 0.2 | 99.0 | 1.0 | 0.0 | 56.4 | 43.2 | 0.4 | 85.7 | 14.3 | 0.0 |
| ces_Latn | 32.8 | 67.0 | 0.2 | 66.6 | 33.0 | 0.4 | 62.8 | 36.8 | 0.4 | 98.0 | 2.0 | 0.0 | 55.6 | 44.0 | 0.4 | 86.6 | 13.0 | 0.4 |
| nld_Latn | 29.8 | 70.0 | 0.2 | 58.1 | 41.2 | 0.6 | 51.7 | 48.3 | 0.0 | 96.0 | 4.0 | 0.0 | 37.8 | 62.2 | 0.0 | 83.3 | 16.3 | 0.4 |
| ell_Grek | 37.4 | 62.4 | 0.2 | 73.6 | 25.8 | 0.6 | 85.8 | 14.0 | 0.2 | 99.4 | 0.4 | 0.2 | 57.8 | 41.8 | 0.4 | 95.0 | 4.6 | 0.4 |
| heb_Hebr | 34.7 | 65.3 | 0.0 | 86.6 | 13.4 | 0.0 | 86.2 | 13.8 | 0.0 | 99.0 | 1.0 | 0.0 | 65.1 | 34.7 | 0.2 | 82.2 | 17.2 | 0.6 |
| pes_Arab | 35.1 | 64.9 | 0.0 | 71.3 | 28.7 | 0.0 | 71.5 | 28.1 | 0.4 | 98.8 | 0.8 | 0.4 | 54.4 | 45.6 | 0.0 | 93.6 | 6.2 | 0.2 |
| ron_Latn | 32.0 | 68.0 | 0.0 | 63.2 | 36.6 | 0.2 | 63.2 | 36.4 | 0.4 | 97.0 | 2.6 | 0.4 | 47.0 | 52.8 | 0.2 | 78.4 | 21.0 | 0.6 |
| **avg** | 30.5 | 69.3 | 0.1 | 63.4 | 36.3 | 0.4 | 61.6 | 37.9 | 0.5 | 95.9 | 3.8 | 0.2 | 44.0 | 55.7 | 0.3 | 80.0 | 19.5 | 0.5 |

Table 8: Win/Loss/Tie rates by Language for Aya-Vision-8B on m-ArenaHard

| | Aya-Vision-8B | | | | | | | | | | | | | | | | | |
| | Gemini-Flash-1.5-8B | | | Llama-3.2-11B-Vision | | | Qwen-2.5-VL-7B | | | Molmo-7B-D | | | Pixtral-12B | | | Pangea-7B | | |
| Language | Win | Loss | Tie | Win | Loss | Tie | Win | Loss | Tie | Win | Loss | Tie | Win | Loss | Tie | Win | Loss | Tie |
|---|---|---|---|---|---|---|---|---|---|---|---|---|---|---|---|---|---|---|
| eng_Latn | 27.6 | 56.7 | 15.7 | 50.8 | 30.6 | 18.7 | 31.3 | 48.5 | 20.1 | 48.3 | 33.0 | 18.6 | 33.6 | 56.7 | 9.7 | 56.0 | 26.9 | 17.2 |
| fra_Latn | 61.2 | 31.3 | 7.5 | 69.4 | 19.4 | 11.2 | 49.2 | 40.3 | 10.4 | 67.8 | 23.7 | 8.5 | 38.1 | 51.5 | 10.4 | 70.9 | 17.9 | 11.2 |
| arb_Arab | 70.9 | 19.4 | 9.7 | 79.8 | 9.0 | 11.2 | 61.9 | 30.6 | 7.5 | 83.9 | 7.6 | 8.5 | 58.2 | 36.6 | 5.2 | 66.4 | 20.9 | 12.7 |
| tur_Latn | 53.4 | 38.4 | 8.3 | 75.9 | 18.1 | 6.0 | 56.4 | 38.4 | 5.3 | 85.5 | 4.3 | 10.3 | 52.6 | 42.1 | 5.3 | 69.9 | 16.5 | 13.5 |
| jpn_Jpan | 47.0 | 44.0 | 9.0 | 67.2 | 21.6 | 11.2 | 45.5 | 49.2 | 5.2 | 72.9 | 13.6 | 13.6 | 42.5 | 47.0 | 10.4 | 65.7 | 18.7 | 15.7 |
| zho_Hans | 52.2 | 35.1 | 12.7 | 66.4 | 19.4 | 14.2 | 35.8 | 55.2 | 9.0 | 79.7 | 10.2 | 10.2 | 40.3 | 44.8 | 14.9 | 59.7 | 23.1 | 17.2 |
| hin_Deva | 58.2 | 35.1 | 6.7 | 79.8 | 14.2 | 6.0 | 69.4 | 21.6 | 9.0 | 85.6 | 6.8 | 7.6 | 45.5 | 50.0 | 4.5 | 68.7 | 21.6 | 9.7 |
| vie_Latn | 56.0 | 36.6 | 7.5 | 65.7 | 23.9 | 10.4 | 58.2 | 35.1 | 6.7 | 79.7 | 13.6 | 6.8 | 48.5 | 46.3 | 5.2 | 72.4 | 20.9 | 6.7 |
| kor_Hang | 56.0 | 32.8 | 11.2 | 73.9 | 18.7 | 7.5 | 54.5 | 32.1 | 13.4 | 79.7 | 8.5 | 11.9 | 42.5 | 47.0 | 10.4 | 76.1 | 14.2 | 9.7 |
| deu_Latn | 48.1 | 42.1 | 9.8 | 66.2 | 24.1 | 9.8 | 42.9 | 47.4 | 9.8 | 77.8 | 12.0 | 10.3 | 33.8 | 58.6 | 7.5 | 69.2 | 21.1 | 9.8 |
| spa_Latn | 53.7 | 37.3 | 9.0 | 70.2 | 19.4 | 10.4 | 37.3 | 50.0 | 12.7 | 65.2 | 20.3 | 14.4 | 37.3 | 50.0 | 12.7 | 64.9 | 23.9 | 11.2 |
| ind_Latn | 58.2 | 31.3 | 10.4 | 74.6 | 18.7 | 6.7 | 59.7 | 35.1 | 5.2 | 78.8 | 16.1 | 5.1 | 59.7 | 35.1 | 5.2 | 65.7 | 25.4 | 9.0 |
| ita_Latn | 61.2 | 29.9 | 9.0 | 71.6 | 18.7 | 9.7 | 47.0 | 39.5 | 13.4 | 72.9 | 15.2 | 11.9 | 47.0 | 39.5 | 13.4 | 66.4 | 23.1 | 10.4 |
| pol_Latn | 58.2 | 36.6 | 5.2 | 74.6 | 20.1 | 5.2 | 47.8 | 44.8 | 7.5 | 87.3 | 4.2 | 8.5 | 47.8 | 44.8 | 7.5 | 72.4 | 16.4 | 11.2 |
| por_Latn | 55.2 | 33.6 | 11.2 | 70.9 | 22.4 | 6.7 | 49.2 | 38.1 | 12.7 | 66.1 | 21.2 | 12.7 | 49.2 | 38.1 | 12.7 | 73.1 | 15.7 | 11.2 |
| rus_Cyrl | 50.0 | 43.3 | 6.7 | 63.4 | 25.4 | 11.2 | 41.8 | 50.0 | 8.2 | 70.3 | 16.9 | 12.7 | 41.8 | 50.0 | 8.2 | 67.9 | 18.7 | 13.4 |
| ukr_Cyrl | 57.5 | 32.1 | 10.4 | 73.9 | 17.9 | 8.2 | 55.2 | 35.8 | 9.0 | 83.9 | 8.5 | 7.6 | 55.2 | 35.8 | 9.0 | 74.6 | 16.4 | 9.0 |
| ces_Latn | 51.5 | 41.0 | 7.5 | 78.4 | 17.2 | 4.5 | 51.5 | 41.0 | 7.5 | 88.1 | 6.8 | 5.1 | 51.5 | 41.0 | 7.5 | 76.1 | 12.7 | 11.2 |
| nld_Latn | 53.0 | 35.8 | 11.2 | 67.9 | 20.9 | 11.2 | 55.2 | 32.1 | 12.7 | 79.7 | 12.7 | 7.6 | 55.2 | 32.1 | 12.7 | 69.4 | 18.7 | 11.9 |
| ell_Grek | 64.9 | 30.6 | 4.5 | 83.6 | 11.9 | 4.5 | 67.2 | 25.4 | 7.5 | 94.9 | 2.5 | 2.5 | 67.2 | 25.4 | 7.5 | 83.6 | 8.2 | 8.2 |
| heb_Hebr | 67.2 | 28.4 | 4.5 | 87.3 | 8.2 | 4.5 | 73.9 | 18.7 | 7.5 | 90.7 | 1.7 | 7.6 | 73.9 | 18.7 | 7.5 | 75.4 | 17.9 | 6.7 |
| pes_Arab | 67.9 | 23.9 | 8.2 | 75.4 | 17.2 | 7.5 | 61.9 | 26.9 | 11.2 | 84.8 | 5.9 | 9.3 | 61.9 | 26.9 | 11.2 | 82.8 | 9.7 | 7.5 |
| ron_Latn | 59.0 | 32.1 | 9.0 | 73.1 | 21.6 | 5.2 | 58.2 | 31.3 | 10.4 | 83.0 | 8.5 | 8.5 | 58.2 | 31.3 | 10.4 | 68.7 | 20.9 | 10.4 |
| **avg** | 56.0 | 35.1 | 8.9 | 72.1 | 19.1 | 8.8 | 52.7 | 37.7 | 9.6 | 78.5 | 11.9 | 9.6 | 49.6 | 41.3 | 9.1 | 70.3 | 18.7 | 11.0 |

Table 9: Win/Loss/Tie rates by Language for Aya-Vision-8B on AyaVisionBench

| | Aya-Vision-8B | | | | | | | | | | | | | | | | | |
| | Gemini-Flash-1.5-8B | | | Llama-3.2-11B-Vision | | | Qwen-2.5-VL-7B | | | Molmo-7B-D | | | Pixtral-12B | | | Pangea-7B | | |
| Language | Win | Loss | Tie | Win | Loss | Tie | Win | Loss | Tie | Win | Loss | Tie | Win | Loss | Tie | Win | Loss | Tie |
|---|---|---|---|---|---|---|---|---|---|---|---|---|---|---|---|---|---|---|
| eng_Latn | 42.2 | 53.4 | 4.4 | 59.8 | 37.4 | 2.8 | 37.4 | 58.4 | 4.2 | 59.0 | 35.0 | 6.0 | 46.2 | 49.0 | 4.8 | 59.0 | 35.0 | 6.0 |
| fra_Latn | 61.2 | 36.6 | 3.6 | 74.4 | 22.0 | 3.6 | 49.2 | 49.4 | 3.4 | 69.8 | 26.2 | 4.0 | 49.8 | 45.2 | 5.0 | 70.9 | 17.9 | 11.2 |
| arb_Arab | 70.9 | 19.4 | 9.7 | 84.8 | 13.0 | 2.2 | 61.9 | 30.6 | 7.5 | 72.0 | 22.6 | 5.4 | 67.8 | 29.2 | 3.0 | 72.0 | 22.6 | 5.4 |
| tur_Latn | 63.6 | 32.4 | 4.0 | 83.0 | 14.4 | 2.6 | 56.4 | 38.4 | 5.3 | 85.5 | 4.3 | 10.3 | 52.6 | 42.1 | 5.3 | 69.9 | 16.5 | 13.5 |
| jpn_Jpan | 63.2 | 33.2 | 3.6 | 81.7 | 13.5 | 4.8 | 47.1 | 48.3 | 4.6 | 73.2 | 20.9 | 5.8 | 53.7 | 41.3 | 5.0 | 73.2 | 20.9 | 5.8 |
| zho_Hans | 65.6 | 29.8 | 4.6 | 77.2 | 18.0 | 4.8 | 46.6 | 49.6 | 3.8 | 79.7 | 28.4 | 5.2 | 51.4 | 44.6 | 4.0 | 66.4 | 28.4 | 5.2 |
| hin_Deva | 69.7 | 26.8 | 3.4 | 83.2 | 15.0 | 1.8 | 78.3 | 18.5 | 3.2 | 85.6 | 6.8 | 7.6 | 45.5 | 50.0 | 4.5 | 68.7 | 21.6 | 9.7 |
| vie_Latn | 70.5 | 26.1 | 3.4 | 78.0 | 19.4 | 2.6 | 59.3 | 37.7 | 3.0 | 79.7 | 13.6 | 6.8 | 48.5 | 46.3 | 5.2 | 78.2 | 17.2 | 4.6 |
| kor_Hang | 66.0 | 29.6 | 4.4 | 86.2 | 10.4 | 3.4 | 54.5 | 32.1 | 13.4 | 79.7 | 8.5 | 11.9 | 42.5 | 47.0 | 10.4 | 76.1 | 14.2 | 9.7 |
| deu_Latn | 57.8 | 39.6 | 2.6 | 75.0 | 20.6 | 4.4 | 42.9 | 47.4 | 9.8 | 77.8 | 12.0 | 10.3 | 33.8 | 58.7 | 7.5 | 69.2 | 21.1 | 9.8 |
| spa_Latn | 53.7 | 37.3 | 9.0 | 71.1 | 25.1 | 3.8 | 37.3 | 50.0 | 12.7 | 65.3 | 20.3 | 14.4 | 37.3 | 50.0 | 12.7 | 64.9 | 23.9 | 11.2 |
| ind_Latn | 58.2 | 31.3 | 10.5 | 78.2 | 17.6 | 4.2 | 59.0 | 35.8 | 5.2 | 89.4 | 7.2 | 3.4 | 56.6 | 35.2 | 8.2 | 65.8 | 27.2 | 7.0 |
| ita_Latn | 62.0 | 33.2 | 4.8 | 73.8 | 22.2 | 4.0 | 49.4 | 45.8 | 4.8 | 84.8 | 10.8 | 4.4 | 53.4 | 41.4 | 5.2 | 71.4 | 23.2 | 5.4 |
| pol_Latn | 62.7 | 32.5 | 4.8 | 80.2 | 16.2 | 3.6 | 56.5 | 40.1 | 3.4 | 90.0 | 5.4 | 4.6 | 63.1 | 34.1 | 2.8 | 77.8 | 18.6 | 3.6 |
| por_Latn | 62.0 | 31.0 | 7.0 | 74.2 | 21.6 | 4.2 | 48.4 | 45.4 | 6.2 | 66.1 | 21.2 | 12.7 | 50.6 | 41.8 | 7.6 | 66.8 | 25.6 | 7.6 |
| rus_Cyrl | 65.0 | 32.8 | 2.2 | 81.9 | 14.3 | 3.8 | 56.1 | 41.3 | 2.6 | 85.9 | 8.7 | 5.4 | 56.3 | 40.2 | 3.4 | 70.8 | 23.9 | 5.2 |
| ukr_Cyrl | 62.5 | 34.3 | 3.2 | 82.4 | 13.2 | 4.4 | 58.3 | 37.1 | 4.6 | 92.6 | 4.6 | 2.8 | 69.9 | 25.9 | 4.2 | 80.2 | 16.2 | 3.6 |
| ces_Latn | 63.4 | 30.0 | 6.6 | 79.2 | 15.0 | 5.8 | 60.0 | 36.4 | 3.6 | 88.0 | 6.8 | 5.4 | 63.8 | 30.8 | 5.4 | 80.4 | 14.6 | 5.0 |
| nld_Latn | 63.0 | 33.6 | 3.4 | 77.8 | 17.6 | 4.6 | 52.8 | 43.0 | 4.2 | 91.0 | 6.0 | 3.0 | 57.0 | 37.8 | 5.2 | 76.8 | 18.8 | 4.4 |
| ell_Grek | 75.2 | 22.0 | 2.8 | 84.4 | 12.6 | 3.0 | 73.8 | 23.2 | 3.0 | 95.2 | 3.2 | 1.6 | 75.0 | 20.8 | 4.2 | 90.0 | 7.4 | 2.6 |
| heb_Hebr | 70.0 | 26.0 | 4.0 | 85.2 | 11.2 | 3.6 | 77.8 | 18.8 | 3.4 | 92.0 | 4.6 | 3.4 | 70.4 | 25.0 | 4.6 | 73.2 | 22.6 | 4.2 |
| pes_Arab | 76.8 | 19.8 | 3.4 | 88.2 | 9.4 | 2.4 | 72.3 | 24.7 | 3.0 | 93.4 | 3.6 | 3.0 | 76.8 | 19.0 | 4.2 | 86.4 | 10.0 | 3.6 |
| ron_Latn | 63.1 | 31.9 | 5.0 | 78.4 | 15.8 | 5.8 | 60.3 | 35.1 | 4.6 | 89.2 | 6.4 | 4.4 | 63.7 | 30.9 | 5.4 | 68.3 | 27.7 | 4.0 |
| **avg** | 64.5 | 31.4 | 4.0 | 79.1 | 17.2 | 3.8 | 57.5 | 38.4 | 4.1 | 80.3 | 15.3 | 4.5 | 59.4 | 35.7 | 5.0 | 73.1 | 22.1 | 4.8 |

Table 10: Win/Loss/Tie rates by Language for Aya-Vision-8B on m-WildVision

| Language | Aya-Vision-32B | | | | | | | | |
| | Llama-3.2-90B-Vision | | | Molmo-72B | | | Qwen-2.5-VL-72B | | |
| | Win | Loss | Tie | Win | Loss | Tie | Win | Loss | Tie |
|---|---|---|---|---|---|---|---|---|---|
| eng_Latn | 26.2 | 73.6 | 0.2 | 66.0 | 32.8 | 1.2 | 35.8 | 63.6 | 0.6 |
| fra_Latn | 39.6 | 60.4 | 0.0 | 72.2 | 27.6 | 0.2 | 46.8 | 52.8 | 0.4 |
| hin_Deva | 47.4 | 52.0 | 0.6 | 86.0 | 14.0 | 0.0 | 69.2 | 30.8 | 0.0 |
| arb_Arab | 54.2 | 45.2 | 0.6 | 81.4 | 18.6 | 0.0 | 59.6 | 40.4 | 0.0 |
| tur_Latn | 45.2 | 54.4 | 0.4 | 78.6 | 20.8 | 0.6 | 51.4 | 48.2 | 0.4 |
| jpn_Jpan | 47.2 | 52.4 | 0.4 | 84.2 | 15.8 | 0.0 | 54.8 | 44.6 | 0.6 |
| zho_Hans | 42.8 | 57.0 | 0.2 | 75.2 | 24.6 | 0.2 | 43.6 | 55.6 | 0.8 |
| vie_Latn | 41.8 | 58.0 | 0.2 | 77.0 | 22.6 | 0.4 | 55.0 | 44.8 | 0.2 |
| kor_Hang | 51.6 | 48.4 | 0.0 | 78.6 | 21.2 | 0.2 | 56.4 | 43.6 | 0.0 |
| deu_Latn | 40.4 | 59.6 | 0.0 | 78.6 | 21.0 | 0.4 | 47.4 | 51.8 | 0.8 |
| ind_Latn | 39.8 | 59.8 | 0.4 | 76.4 | 23.2 | 0.4 | 49.2 | 50.4 | 0.4 |
| ita_Latn | 41.0 | 59.0 | 0.0 | 75.2 | 24.2 | 0.6 | 38.2 | 61.2 | 0.6 |
| pol_Latn | 42.2 | 57.6 | 0.2 | 75.4 | 24.0 | 0.6 | 43.4 | 56.4 | 0.2 |
| por_Latn | 35.2 | 64.6 | 0.2 | 70.6 | 29.0 | 0.4 | 44.6 | 55.4 | 0.0 |
| rus_Cyrl | 40.0 | 60.0 | 0.0 | 66.8 | 33.0 | 0.2 | 47.6 | 52.0 | 0.4 |
| spa_Latn | 38.8 | 60.8 | 0.4 | 69.2 | 30.6 | 0.2 | 45.4 | 54.0 | 0.6 |
| ukr_Cyrl | 44.6 | 55.2 | 0.2 | 80.0 | 20.0 | 0.0 | 48.0 | 51.8 | 0.2 |
| ces_Latn | 45.6 | 54.2 | 0.2 | 75.6 | 24.4 | 0.0 | 53.0 | 47.0 | 0.0 |
| nld_Latn | 42.0 | 57.2 | 0.8 | 76.8 | 23.2 | 0.0 | 46.8 | 52.6 | 0.6 |
| ell_Grek | 46.2 | 53.6 | 0.2 | 84.2 | 15.4 | 0.4 | 62.4 | 37.2 | 0.4 |
| heb_Hebr | 51.2 | 48.6 | 0.2 | 85.8 | 14.0 | 0.2 | 63.4 | 36.6 | 0.0 |
| pes_Arab | 51.0 | 48.8 | 0.2 | 84.4 | 15.0 | 0.6 | 57.6 | 42.4 | 0.0 |
| ron_Latn | 40.4 | 59.2 | 0.4 | 78.8 | 21.0 | 0.2 | 51.6 | 48.2 | 0.2 |
| **avg** | 43.2 | 56.5 | 0.3 | 77.3 | 22.4 | 0.3 | 50.9 | 48.8 | 0.3 |

Table 11: Win/Loss/Tie rates by Language for Aya-Vision-32B on m-ArenaHard

| Language | Aya-Vision-32B | | | | | | | | |
| | Llama-3.2-90B-Vision | | | Molmo-72B | | | Qwen-2.5-VL-72B | | |
| | Win | Loss | Tie | Win | Loss | Tie | Win | Loss | Tie |
|---|---|---|---|---|---|---|---|---|---|
| eng_Latn | 49.25 | 38.81 | 11.94 | 35.82 | 54.48 | 9.70 | 62.69 | 24.63 | 12.69 |
| fra_Latn | 64.93 | 24.63 | 10.45 | 53.73 | 39.55 | 6.72 | 49.25 | 42.54 | 8.21 |
| hin_Deva | 74.63 | 23.13 | 2.24 | 72.39 | 25.37 | 2.24 | 35.82 | 61.19 | 2.99 |
| arb_Arab | 70.90 | 19.40 | 9.70 | 73.13 | 20.90 | 5.97 | 44.03 | 47.76 | 8.21 |
| tur_Latn | 63.91 | 30.08 | 6.02 | 64.66 | 30.08 | 5.26 | 52.63 | 44.36 | 3.01 |
| jpn_Jpan | 61.94 | 28.36 | 9.70 | 61.94 | 35.82 | 2.24 | 48.51 | 45.52 | 5.97 |
| zho_Hans | 65.67 | 28.36 | 5.97 | 66.42 | 26.87 | 6.72 | 44.03 | 46.27 | 9.70 |
| vie_Latn | 64.93 | 24.63 | 10.45 | 50.75 | 42.54 | 6.72 | 52.99 | 41.04 | 5.97 |
| kor_Hang | 64.93 | 28.36 | 6.72 | 58.96 | 33.58 | 7.46 | 44.78 | 44.78 | 10.45 |
| deu_Latn | 69.92 | 21.80 | 8.27 | 60.15 | 33.83 | 6.02 | 48.87 | 48.12 | 3.01 |
| ind_Latn | 68.66 | 26.87 | 4.48 | 56.72 | 37.31 | 5.97 | 47.76 | 44.78 | 7.46 |
| ita_Latn | 62.69 | 29.85 | 7.46 | 55.97 | 35.07 | 8.96 | 52.99 | 39.55 | 7.46 |
| pol_Latn | 74.63 | 20.90 | 4.48 | 65.67 | 28.36 | 5.97 | 48.51 | 45.52 | 5.97 |
| por_Latn | 52.99 | 41.79 | 5.22 | 51.49 | 42.54 | 5.97 | 54.48 | 36.57 | 8.96 |
| rus_Cyrl | 60.45 | 29.10 | 10.45 | 50.75 | 40.30 | 8.96 | 50.75 | 41.04 | 8.21 |
| spa_Latn | 61.19 | 29.85 | 8.96 | 52.99 | 37.31 | 9.70 | 50.75 | 43.28 | 5.97 |
| ukr_Cyrl | 75.37 | 20.90 | 3.73 | 61.94 | 32.84 | 5.22 | 50.75 | 43.28 | 5.97 |
| ces_Latn | 73.88 | 20.15 | 5.97 | 67.91 | 27.61 | 4.48 | 50.75 | 46.27 | 2.99 |
| nld_Latn | 64.93 | 24.63 | 10.45 | 52.24 | 42.54 | 5.22 | 50.00 | 45.52 | 4.48 |
| ell_Grek | 66.42 | 26.12 | 7.46 | 78.36 | 17.91 | 3.73 | 38.81 | 51.49 | 9.70 |
| heb_Hebr | 68.66 | 24.63 | 6.72 | 68.66 | 26.87 | 4.48 | 42.54 | 51.49 | 5.97 |
| pes_Arab | 70.90 | 23.88 | 5.22 | 78.36 | 18.66 | 2.99 | 46.27 | 50.00 | 3.73 |
| ron_Latn | 64.18 | 31.34 | 4.48 | 68.66 | 26.87 | 4.48 | 47.01 | 45.52 | 7.46 |
| **avg** | 65.91 | 26.85 | 7.24 | 61.20 | 32.92 | 5.88 | 48.48 | 44.81 | 6.72 |

Table 12: Win/Loss/Tie rates by Language for Aya-Vision-32B on AyaVisionBench

| Language | Aya-Vision-32B | | | | | | | | |
| | Qwen-2.5-VL-72B | | | Llama-3.2-90B-Vision | | | Molmo-72B | | |
| | Win | Loss | Tie | Win | Loss | Tie | Win | Loss | Tie |
|---|---|---|---|---|---|---|---|---|---|
| eng_Latn | 37.4 | 56.4 | 6.2 | 67.6 | 29.2 | 3.2 | 56.2 | 39.2 | 4.6 |
| fra_Latn | 46.2 | 50.0 | 3.8 | 69.9 | 26.4 | 3.6 | 59.0 | 37.2 | 3.8 |
| hin_Deva | 67.4 | 30.6 | 2.0 | 78.4 | 17.6 | 4.0 | 75.6 | 20.0 | 4.4 |
| arb_Arab | 57.4 | 39.2 | 3.4 | 79.0 | 17.8 | 3.2 | 79.2 | 16.8 | 4.0 |
| tur_Latn | 56.0 | 39.6 | 4.4 | 77.8 | 19.0 | 3.2 | 76.5 | 20.5 | 3.0 |
| jpn_Jpan | 49.0 | 46.4 | 4.6 | 72.2 | 25.4 | 2.4 | 76.2 | 20.2 | 3.6 |
| zho_Hans | 39.0 | 56.4 | 4.6 | 77.0 | 19.0 | 4.0 | 78.0 | 19.6 | 2.4 |
| vie_Latn | 57.4 | 38.6 | 4.0 | 76.6 | 21.4 | 2.0 | 64.2 | 31.6 | 4.2 |
| kor_Hang | 55.4 | 40.8 | 3.8 | 75.4 | 21.0 | 3.6 | 70.4 | 25.2 | 4.4 |
| deu_Latn | 49.2 | 46.4 | 4.4 | 67.0 | 28.6 | 4.4 | 68.0 | 28.0 | 4.0 |
| ind_Latn | 51.0 | 45.8 | 3.2 | 72.0 | 26.0 | 2.0 | 65.2 | 30.0 | 4.8 |
| ita_Latn | 46.2 | 49.0 | 4.8 | 69.8 | 26.2 | 4.0 | 59.0 | 33.8 | 7.2 |
| pol_Latn | 50.8 | 46.8 | 2.4 | 73.6 | 23.4 | 3.0 | 67.2 | 29.0 | 3.8 |
| por_Latn | 49.2 | 45.8 | 5.0 | 68.2 | 26.8 | 5.0 | 61.2 | 33.6 | 5.2 |
| rus_Cyrl | 50.2 | 47.2 | 2.6 | 73.2 | 23.6 | 3.2 | 60.3 | 36.3 | 3.4 |
| spa_Latn | 48.6 | 46.6 | 4.8 | 65.2 | 30.6 | 4.2 | 57.0 | 37.8 | 5.2 |
| ukr_Cyrl | 58.4 | 38.8 | 2.8 | 74.4 | 21.4 | 4.2 | 70.6 | 25.4 | 4.0 |
| ces_Latn | 54.4 | 42.2 | 3.4 | 69.6 | 27.2 | 3.2 | 67.6 | 28.8 | 3.6 |
| nld_Latn | 47.6 | 48.8 | 3.6 | 69.4 | 25.8 | 4.8 | 61.4 | 33.8 | 4.8 |
| ell_Grek | 66.6 | 30.2 | 3.2 | 75.0 | 22.0 | 3.0 | 84.2 | 11.8 | 4.0 |
| heb_Hebr | 66.0 | 30.6 | 3.4 | 74.2 | 22.8 | 3.0 | 74.0 | 22.4 | 3.6 |
| pes_Arab | 64.4 | 30.8 | 4.8 | 80.6 | 16.6 | 2.8 | 77.6 | 18.4 | 4.0 |
| ron_Latn | 58.0 | 39.2 | 2.8 | 73.6 | 24.4 | 2.0 | 74.6 | 21.8 | 3.6 |
| **avg** | 53.3 | 42.9 | 3.8 | 73.0 | 23.6 | 3.4 | 68.8 | 27.0 | 4.2 |

Table 13: Win/Loss/Tie rates by Language for Aya-Vision-32B on m-WildVision.

| | eng_Latn | fra_Latn | heb_Hebr | hin_Deva | ron_Latn | tha_Thai | zho_Hans | **avg** |
|---|---|---|---|---|---|---|---|---|
| Pangea-7B | 55.30 | 43.60 | 59.30 | 53.50 | 45.80 | 67.20 | 50.20 | 53.56 |
| Molmo-7B-D | 68.09 | 54.17 | 34.29 | 31.92 | 30.28 | 53.73 | 46.21 | 45.53 |
| Llama-3.2-11B-Vision | 56.03 | 45.08 | 31.07 | 45.00 | 38.38 | 42.16 | 20.22 | 39.71 |
| Pixtral-12B | 57.20 | 43.56 | 40.00 | 55.38 | 41.20 | 55.97 | 29.24 | 46.08 |
| Qwen-2.5-VL-7B | 57.98 | 52.65 | 54.29 | 54.62 | 44.72 | 67.16 | 51.62 | 54.72 |
| Aya-Vision-8B | 57.59 | 54.92 | 58.57 | 66.92 | 54.93 | 33.21 | 56.32 | 54.64 |
| Molmo-72B | 59.92 | 54.92 | 58.21 | 62.69 | 50.70 | 65.30 | 47.29 | 57.01 |
| Llama-3.2-90B-Vision | 75.00 | 67.05 | 59.64 | 70.38 | 59.51 | 68.66 | 53.43 | 64.81 |
| Qwen-2.5-VL-72B | 55.25 | 49.62 | 62.86 | 66.15 | 46.13 | 74.25 | 58.48 | 58.96 |
| Aya-Vision-32B | 55.64 | 60.61 | 66.43 | 71.54 | 57.75 | 43.07 | 61.73 | 59.54 |

Table 14: MaxM

| | fra_Latn | jpn_Jpan | ind_Latn | por_Latn | hin_Deva | arb_Arab | eng_Latn | **avg** |
|---|---|---|---|---|---|---|---|---|
| Pangea-7B | 45.30 | 40.50 | 46.50 | 46.10 | 41.60 | 42.30 | 45.70 | 44.00 |
| Molmo-7B-D | 38.90 | 37.10 | 38.90 | 38.10 | 34.90 | 36.70 | 40.50 | 37.87 |
| Llama-3.2-11B-Vision | 43.30 | 40.90 | 42.10 | 44.10 | 39.90 | 41.60 | 47.20 | 42.73 |
| Pixtral-12B | 47.00 | 43.90 | 40.10 | 47.80 | 32.60 | 36.20 | 48.30 | 42.27 |
| Qwen-2.5-VL-7B | 49.70 | 46.10 | 47.80 | 49.80 | 41.20 | 41.70 | 51.10 | 46.77 |
| Aya-Vision-8B | 40.20 | 41.40 | 39.50 | 38.50 | 38.10 | 40.10 | 41.80 | 39.94 |
| Molmo-72B | 52.80 | 49.00 | 52.80 | 55.40 | 48.00 | 51.20 | 51.50 | 51.53 |
| Llama-3.2-90B-Vision | 56.60 | 52.90 | 55.20 | 54.30 | 46.60 | 45.00 | 56.20 | 52.40 |
| Qwen-2.5-VL-72B | 62.40 | 60.60 | 64.00 | 62.00 | 60.80 | 59.70 | 62.70 | 61.74 |
| Aya-Vision-32B | 44.90 | 42.90 | 46.60 | 45.30 | 45.00 | 44.10 | 47.00 | 45.11 |

Table 15: xMMMU

| | arb_Arab | deu_Latn | fra_Latn | ita_Latn | jpn_Jpan | kor_Hang | rus_Cyrl | vie_Latn | tha_Thai | **avg** |
|---|---|---|---|---|---|---|---|---|---|---|
| Pangea-7B | 8.53 | 29.96 | 32.39 | 23.87 | 9.30 | 13.44 | 7.67 | 21.38 | 15.15 | 17.97 |
| Molmo-7B-D | 5.83 | 26.24 | 35.67 | 29.86 | 7.61 | 9.86 | 5.03 | 15.05 | 15.15 | 16.70 |
| Llama-3.2-11B-Vision | 7.97 | 24.24 | 27.99 | 22.85 | 10.75 | 13.08 | 7.01 | 17.31 | 16.88 | 16.45 |
| Pixtral-12B | 7.68 | 32.54 | 37.92 | 32.69 | 8.33 | 13.08 | 7.14 | 19.12 | 14.29 | 19.20 |
| Qwen-2.5-VL-7B | 19.26 | 35.31 | 42.66 | 36.76 | 21.98 | 32.80 | 10.45 | 37.33 | 22.51 | 28.78 |
| Aya-Vision-8B | 13.69 | 28.72 | 35.89 | 28.39 | 10.51 | 13.08 | 6.35 | 17.99 | 7.79 | 18.05 |
| Molmo-72B | 6.54 | 30.34 | 35.44 | 30.54 | 9.42 | 10.04 | 8.73 | 18.21 | 17.32 | 18.51 |
| Llama-3.2-90B-Vision | 19.91 | 36.35 | 40.29 | 35.29 | 17.27 | 30.11 | 10.98 | 29.30 | 25.97 | 27.28 |
| Qwen-2.5-VL-72B | 23.19 | 35.78 | 43.91 | 39.14 | 21.98 | 35.66 | 12.83 | 42.87 | 27.27 | 31.40 |
| Aya-Vision-32B | 116.33 | 34.83 | 40.52 | 32.20 | 15.03 | 14.57 | 10.28 | 23.91 | 11.45 | 22.12 |

Table 16: MTVQA

| | hin_Deva | ind_Latn | kor_Hang | spa_Latn | eng_Latn | zho_Hans | jpn_Jpan | **avg** |
|---|---|---|---|---|---|---|---|---|
| Pangea-7B | 29.00 | 36.50 | 28.50 | 34.00 | 26.50 | 36.00 | 35.00 | 32.21 |
| Molmo-7B-D | 4.00 | 24.50 | 8.50 | 42.50 | 65.50 | 2.00 | 16.50 | 23.36 |
| Llama-3.2-11B-Vision | 13.00 | 35.50 | 13.78 | 43.00 | 55.50 | 23.00 | 16.33 | 28.59 |
| Pixtral-12B | 50.50 | 66.50 | 60.00 | 72.50 | 74.00 | 64.00 | 64.00 | 64.50 |
| Qwen-2.5-VL-7B | 20.50 | 58.50 | 53.00 | 66.50 | 78.00 | 71.50 | 59.00 | 58.14 |
| Aya-Vision-8B | 56.50 | 60.50 | 56.00 | 60.00 | 60.50 | 55.50 | 61.50 | 58.64 |
| Molmo-72B | 19.5 | 53.5 | 27.0 | 64.5 | 65.5 | 42.5 | 45.5 | 45.43 |
| Llama-3.2-90B-Vision | 38.50 | 54.50 | 42.35 | 60.50 | 63.00 | 53.00 | 46.00 | 51.12 |
| Qwen-2.5-VL-72B | 44.50 | 77.00 | 71.94 | 80.50 | 82.00 | 71.00 | 71.00 | 71.13 |
| Aya-Vision-32B | 68.50 | 72.00 | 62.50 | 77.00 | 72.50 | 66.50 | 71.50 | 70.07 |

Table 17: xChatBench

| | tha_Thai | tel_Telu | ben_Beng | eng_Latn | spa_Latn | jpn_Jpan | zho_Hans | swh_Latn | deu_Latn | rus_Cyrl | fra_Latn | **avg** |
|---|---|---|---|---|---|---|---|---|---|---|---|---|
| Pangea-7B | 49.60 | 5.60 | 0.00 | 82.00 | 74.8 | 22.00 | 68.00 | 54.0 | 68.4 | 68.0 | 63.2 | 50.51 |
| Molmo-7B-D | 24.50 | 2.41 | 6.02 | 73.90 | 39.36 | 41.77 | 58.06 | 0.00 | 52.61 | 47.79 | 36.14 | 34.78 |
| Llama-3.2-11B-Vision | 64.26 | 6.88 | 18.88 | 84.74 | 71.89 | 55.24 | 73.90 | 56.63 | 76.31 | 77.11 | 70.68 | 59.68 |
| Pixtral-12B | 63.86 | 36.55 | 57.83 | 89.16 | 82.73 | 64.66 | 73.90 | 23.69 | 79.92 | 78.71 | 74.30 | 65.94 |
| Qwen-2.5-VL-7B | 58.44 | 4.42 | 37.75 | 85.14 | 43.37 | 61.85 | 72.29 | 4.09 | 74.30 | 63.27 | 26.10 | 48.27 |
| Aya-Vision-8B | 12.45 | 0.00 | 6.83 | 84.34 | 77.91 | 67.87 | 74.70 | 4.90 | 75.90 | 80.72 | 73.49 | 50.83 |
| Molmo-72B | 79.52 | 11.65 | 55.82 | 96.39 | 89.56 | 69.08 | 86.35 | 57.03 | 88.76 | 90.76 | 81.12 | 73.27 |
| Llama-3.2-90B-Vision | 84.34 | 7.63 | 26.51 | 96.39 | 81.53 | 69.08 | 77.91 | 82.73 | 89.96 | 87.95 | 6.02 | 60.72 |
| Qwen-2.5-VL-72B | 87.95 | 13.25 | 64.26 | 95.18 | 93.17 | 86.35 | 91.16 | 65.06 | 89.52 | 91.57 | 80.32 | 77.98 |
| Aya-Vision-32B | 39.36 | 0.00 | 14.46 | 87.95 | 82.33 | 75.50 | 80.32 | 23.69 | 81.53 | 76.31 | 72.29 | 57.61 |

Table 18: MGSM

| | Pangea-7B | Molmo-7B-D | Llama-3.2-11B-Vision | Pixtral-12B | Qwen-2.5-VL-7B | Aya-Vision-8B | Molmo-72B | Llama-3.2-90B-Vision | Qwen-2.5-VL-72B | Aya-Vision-32B |
|---|---|---|---|---|---|---|---|---|---|---|
| swh_Latn | 39.25 | 22.75 | 29.75 | 44.36 | 33.50 | 29.00 | 52.75 | 57.75 | 54.25 | 36.50 |
| spa_Latn | 54.25 | 44.25 | 66.75 | 69.00 | 68.75 | 65.25 | 73.50 | 80.00 | 81.00 | 62.91 |
| jpn_Jpan | 39.75 | 32.00 | 51.00 | 59.90 | 61.75 | 59.75 | 67.50 | 78.50 | 82.50 | 67.50 |
| kor_Hang | 46.75 | 33.00 | 50.75 | 64.75 | 61.25 | 56.50 | 73.25 | 75.50 | 79.75 | 64.25 |
| deu_Latn | 54.00 | 41.10 | 66.50 | 67.50 | 64.75 | 67.75 | 77.75 | 80.25 | 82.00 | 68.75 |
| por_Latn | 55.75 | 44.00 | 64.25 | 69.10 | 65.75 | 65.00 | 74.75 | 84.50 | 83.75 | 62.25 |
| zho_Hans | 53.75 | 45.00 | 63.50 | 68.09 | 66.08 | 63.75 | 63.25 | 73.75 | 80.75 | 61.50 |
| ben_Beng | 40.25 | 29.25 | 30.25 | 55.75 | 53.25 | 40.25 | 64.50 | 61.00 | 73.91 | 48.50 |
| eng_Latn | 65.00 | 49.00 | 71.25 | 74.94 | 72.43 | 71.00 | 74.25 | 83.75 | 87.75 | 69.25 |
| ind_Latn | 47.75 | 36.00 | 64.75 | 59.00 | 61.46 | 58.75 | 74.75 | 81.50 | 81.25 | 65.50 |
| hin_Deva | 39.00 | 32.50 | 48.50 | 59.55 | 54.00 | 55.00 | 66.75 | 71.50 | 75.75 | 42.25 |
| arb_Arab | 38.75 | 33.75 | 52.50 | 62.50 | 62.03 | 59.00 | 57.25 | 74.50 | 77.14 | 67.50 |
| fra_Latn | 45.00 | 44.25 | 64.50 | 68.75 | 68.17 | 63.50 | 72.50 | 62.25 | 82.96 | 68.00 |
| yor_Latn | 20.25 | 27.00 | 15.25 | 29.55 | 30.00 | 29.75 | 35.50 | 40.50 | 37.50 | 29.00 |
| ita_Latn | 52.50 | 40.75 | 64.75 | 70.03 | 71.43 | 65.00 | 76.75 | 83.50 | 83.25 | 63.25 |
| **avg** | 46.13 | 36.97 | 53.62 | 61.52 | 59.64 | 56.62 | 67.00 | 72.58 | 76.23 | 58.46 |

Table 19: global MMLU

| | Pangea-7B | Molmo-7B-D | Llama-3.2-11B-Vision | Pixtral-12B | Qwen-2.5-VL-7B | Aya-Vision-8B | Molmo-72B | Llama-3.2-90B-Vision | Qwen-2.5-VL-72B | Aya-Vision-32B |
|---|---|---|---|---|---|---|---|---|---|---|
| eng_Latn->arb_Arab | 27.50 | 11.26 | 26.62 | 21.90 | 24.79 | 38.22 | 32.05 | 36.78 | 36.17 | 38.93 |
| eng_Latn->heb_Hebr | 27.36 | 11.07 | 28.32 | 23.68 | 19.92 | 38.25 | 30.52 | 40.87 | 32.02 | 41.85 |
| eng_Latn->por_Latn | 47.01 | 31.52 | 49.24 | 50.88 | 47.69 | 51.41 | 50.69 | 54.33 | 53.93 | 52.30 |
| eng_Latn->jpn_Jpan | 22.71 | 11.50 | 22.20 | 19.08 | 22.98 | 26.95 | 25.79 | 28.23 | 29.58 | 29.10 |
| eng_Latn->hin_Deva | 20.26 | 6.20 | 27.05 | 20.88 | 13.37 | 29.13 | 23.10 | 34.45 | 24.96 | 30.39 |
| eng_Latn->fra_Latn | 46.68 | 35.49 | 48.79 | 49.68 | 45.37 | 51.42 | 49.84 | 53.81 | 52.83 | 52.17 |
| eng_Latn->ita_Latn | 28.62 | 19.36 | 31.40 | 32.01 | 28.09 | 33.60 | 32.34 | 34.90 | 33.06 | 36.19 |
| eng_Latn->rus_Cyrl | 31.08 | 22.31 | 33.58 | 36.53 | 32.28 | 37.22 | 37.43 | 39.67 | 40.48 | 38.80 |
| eng_Latn->zho_Hans | 31.53 | 21.22 | 28.82 | 24.64 | 34.01 | 33.28 | 36.57 | 35.74 | 38.41 | 34.57 |
| eng_Latn->ind_Latn | 40.46 | 20.05 | 39.66 | 35.33 | 35.76 | 43.29 | 39.94 | 45.89 | 45.37 | 44.46 |
| eng_Latn->spa_Latn | 27.87 | 21.15 | 28.33 | 29.59 | 27.41 | 31.11 | 30.32 | 30.84 | 31.03 | 32.17 |
| eng_Latn->pes_Arab | 14.60 | 8.46 | 27.43 | 20.94 | 18.37 | 30.31 | 24.73 | 33.71 | 27.57 | 31.99 |
| eng_Latn->tur_Latn | 25.82 | 8.23 | 27.36 | 21.50 | 22.16 | 30.99 | 26.95 | 37.01 | 32.10 | 34.17 |
| eng_Latn->vie_Latn | 35.55 | 20.18 | 36.98 | 32.29 | 35.20 | 40.15 | 37.29 | 41.88 | 41.17 | 40.38 |
| eng_Latn->pol_Latn | 19.89 | 11.24 | 25.59 | 22.57 | 22.67 | 28.56 | 26.03 | 30.27 | 28.39 | 30.35 |
| eng_Latn->ell_Grek | 12.27 | 5.14 | 26.23 | 22.63 | 17.28 | 34.06 | 20.68 | 33.77 | 25.70 | 36.46 |
| eng_Latn->ron_Latn | 37.39 | 15.83 | 40.17 | 33.91 | 31.78 | 43.51 | 35.85 | 47.82 | 41.55 | 47.06 |
| eng_Latn->deu_Latn | 34.45 | 21.54 | 39.59 | 41.33 | 36.73 | 40.97 | 41.49 | 45.62 | 43.48 | 44.45 |
| eng_Latn->kor_Hang | 18.76 | 8.69 | 20.56 | 19.00 | 18.01 | 25.96 | 23.37 | 25.92 | 26.23 | 27.44 |
| eng_Latn->ces_Latn | 23.85 | 12.34 | 33.97 | 29.70 | 28.59 | 36.25 | 33.26 | 40.81 | 36.71 | 38.53 |
| eng_Latn->nld_Latn | 24.02 | 15.67 | 28.94 | 26.00 | 27.17 | 31.37 | 30.59 | 33.41 | 31.63 | 33.20 |
| eng_Latn->ukr_Cyrl | 19.24 | 7.90 | 29.56 | 30.41 | 26.02 | 33.77 | 26.55 | 35.85 | 33.29 | 36.40 |
| **avg** | 28.04 | 15.74 | 31.84 | 29.29 | 27.98 | 35.90 | 32.52 | 38.25 | 35.71 | 37.79 |

Table 20: flores

| | Pangea-7B | Molmo-7B-D | Llama-3.2-11B-Vision | Pixtral-12B | Qwen-2.5-VL-7B | Aya-Vision-8B | Molmo-72B | Llama-3.2-90B-Vision | Qwen-2.5-VL-72B | Aya-Vision-32B |
|---|---|---|---|---|---|---|---|---|---|---|
| ('Irish', 'Ireland') | 56.40 | 42.33 | 53.99 | 57.67 | 76.38 | 47.24 | 57.06 | 76.99 | 57.98 | 56.13 |
| ('Swahili', 'Kenya') | 64.10 | 49.45 | 53.11 | 60.07 | 72.53 | 54.95 | 67.77 | 79.85 | 55.31 | 66.18 |
| ('Igbo', 'Nigeria') | 46.00 | 40.50 | 44.00 | 41.50 | 48.00 | 34.67 | 41.50 | 52.00 | 36.55 | 38.00 |
| ('Minangkabau', 'Indonesia') | 47.80 | 44.62 | 51.79 | 51.39 | 68.13 | 52.40 | 58.17 | 76.49 | 51.79 | 61.75 |
| ('Sundanese', 'Indonesia') | 53.00 | 41.00 | 44.00 | 49.00 | 73.50 | 46.50 | 52.00 | 72.50 | 56.50 | 52.53 |
| ('Chinese', 'China') | 74.00 | 70.10 | 63.34 | 69.45 | 89.71 | 65.16 | 75.56 | 83.60 | 85.53 | 75.24 |
| ('Spanish', 'Mexico') | 62.20 | 54.49 | 53.56 | 63.16 | 79.57 | 57.59 | 64.71 | 74.61 | 68.94 | 67.70 |
| ('Tamil', 'India') | 51.90 | 35.98 | 58.41 | 51.87 | 75.70 | 44.39 | 58.41 | 86.45 | 58.88 | 61.68 |
| ('Hindi', 'India') | | 51.74 | 68.16 | 30.85 | 84.58 | 62.69 | 78.11 | 90.05 | 75.12 | 78.11 |
| ('Spanish', 'Argentina') | 68.30 | 57.74 | 57.36 | 69.43 | 80.75 | 64.02 | 75.47 | 78.87 | 75.85 | 75.85 |
| ('Korean', 'South Korea') | 70.70 | 56.55 | 59.66 | 73.45 | 85.86 | 74.39 | 74.14 | 85.17 | 77.59 | 80.00 |
| ('Urdu', 'India') | | 50.45 | 54.55 | 39.09 | 80.00 | 47.27 | 69.55 | 83.64 | 64.09 | 63.93 |
| ('Filipino', 'Philippines') | 58.60 | 45.32 | 51.72 | 64.53 | 74.88 | 44.06 | 64.53 | 82.76 | 65.02 | 66.34 |
| ('Chinese', 'Singapore') | 65.60 | 70.67 | 62.26 | 68.40 | 87.26 | 66.82 | 83.02 | 85.38 | 76.42 | 79.72 |
| ('Spanish', 'Colombia') | 64.70 | 61.00 | 54.36 | 68.46 | 80.91 | 58.51 | 73.86 | 85.48 | 75.10 | 71.67 |
| ('Indonesian', 'Indonesia') | 62.10 | 53.64 | 56.31 | 62.86 | 78.83 | 56.69 | 63.83 | 81.07 | 66.50 | 67.88 |
| ('Spanish', 'Uruguay') | 49.80 | 44.44 | 48.25 | 58.41 | 70.16 | 43.91 | 61.27 | 69.52 | 57.78 | 61.90 |
| ('Portuguese', 'Brazil') | 72.90 | 68.31 | 57.75 | 73.59 | 84.86 | 66.78 | 77.46 | 85.56 | 76.76 | 78.01 |
| ('Norwegian', 'Norway') | 64.50 | 47.49 | 54.52 | 64.21 | 80.60 | 53.20 | 69.90 | 78.93 | 68.56 | 66.22 |
| ('Oromo', 'Ethiopia') | 35.50 | 43.93 | 34.11 | 35.51 | 43.46 | 32.71 | 42.06 | 46.73 | 35.05 | 36.45 |
| ('Bengali', 'India') | 59.10 | 47.00 | 55.59 | 48.25 | 79.72 | 49.82 | 68.88 | 84.97 | 61.27 | 64.31 |
| ('Bulgarian', 'Bulgaria') | 53.90 | 45.80 | 49.06 | 22.91 | 69.19 | 44.74 | 57.68 | 67.39 | 61.99 | 56.49 |
| ('Amharic', 'Ethiopia') | 36.30 | 33.48 | 39.32 | 32.91 | 58.37 | 29.44 | 45.30 | 62.82 | 36.48 | 29.18 |
| ('Malay', 'Malaysia') | 59.70 | 51.75 | 56.19 | 61.90 | 79.68 | 57.01 | 69.84 | 80.32 | 62.50 | 72.38 |
| ('Egyptian_Arabic', 'Egypt') | 49.30 | 43.07 | 49.26 | 43.35 | 74.38 | 51.49 | 58.62 | 71.92 | 61.08 | 68.47 |
| ('Telugu', 'India') | 54.50 | 43.50 | 55.50 | 32.50 | 73.50 | 47.50 | 57.00 | 83.50 | 58.50 | 57.79 |
| ('Spanish', 'Ecuador') | 63.50 | 56.27 | 55.52 | 70.72 | 78.73 | 57.82 | 69.89 | 78.18 | 66.02 | 71.43 |
| ('Spanish', 'Spain') | 72.60 | 66.04 | 69.81 | 82.39 | 92.14 | 74.53 | 79.56 | 90.88 | 83.33 | 87.07 |
| ('Kinyarwanda', 'Rwanda') | 35.70 | 34.63 | 35.32 | 34.47 | 43.83 | 32.76 | 40.43 | 54.89 | 38.30 | 40.43 |
| ('Javanese', 'Indonesia') | 49.50 | 46.46 | 47.81 | 51.18 | 67.34 | 48.15 | 54.88 | 76.09 | 55.22 | 55.56 |
| ('Romanian', 'Romania') | 64.60 | 51.66 | 58.94 | 67.88 | 85.10 | 62.79 | 70.20 | 87.09 | 75.83 | 74.17 |
| ('Urdu', 'Pakistan') | 66.20 | 50.00 | 57.41 | 56.94 | 80.56 | 50.93 | 69.44 | 88.43 | 65.74 | 69.44 |
| ('Japanese', 'Japan') | 48.30 | 43.78 | 50.74 | 49.26 | 69.46 | 48.28 | 57.14 | 64.04 | 58.62 | 59.11 |
| ('Breton', 'France') | 34.60 | 30.86 | 34.57 | 35.80 | 44.20 | 34.41 | 35.06 | 48.64 | 37.78 | 39.36 |
| ('Sinhala', 'Sri_Lanka') | 39.10 | 28.89 | 48.00 | 28.44 | 62.05 | 28.89 | 45.78 | 67.56 | 45.50 | 39.56 |
| ('Russian', 'Russia') | 74.00 | 64.50 | 66.50 | 37.00 | 84.00 | 66.33 | 84.00 | 85.50 | 79.00 | 80.00 |
| ('Marathi', 'India') | | 43.56 | 48.02 | 31.19 | 80.20 | 50.75 | 68.81 | 84.65 | 61.39 | 66.17 |
| ('Spanish', 'Chile') | 70.50 | 64.96 | 60.26 | 71.37 | 81.62 | 63.52 | 76.07 | 85.04 | 73.08 | 77.16 |
| ('Mongolian', 'Mongolia') | 42.30 | 33.33 | 39.42 | 39.74 | 54.81 | 28.53 | 47.76 | 55.77 | 39.10 | 36.01 |
| **avg** | 57.20 | 48.96 | 52.78 | 52.59 | 73.71 | 51.32 | 63.20 | 76.24 | 61.69 | 62.80 |

Table 21: CVQA

| | Pangea-7B | Molmo-7B-D | Llama-3.2-11B-Vision | Pixtral-12B | Qwen-2.5-VL-7B | Aya-Vision-8B | Molmo-72B | Llama-3.2-90B-Vision | Qwen-2.5-VL-72B | Aya-Vision-32B |
|---|---|---|---|---|---|---|---|---|---|---|
| eng_Latn | 24.70 | 26.90 | 41.58 | 43.30 | 36.40 | 40.99 | 53.81 | 51.60 | 53.80 | 46.07 |
| spa_Latn | 46.20 | 47.80 | 50.54 | 57.83 | 57.80 | 51.36 | 69.16 | 69.68 | 72.50 | 60.46 |
| hin_Deva | 24.30 | 30.00 | 29.75 | 17.29 | 33.80 | 33.56 | 39.08 | 39.13 | 46.20 | 34.20 |
| nld_Latn | 37.30 | 41.50 | 39.88 | 43.71 | 46.40 | 38.86 | 51.57 | 52.26 | 57.40 | 43.65 |
| ukr_Cyrl | 33.00 | 35.80 | 37.51 | 29.18 | 45.40 | 41.14 | 54.08 | 56.10 | 65.20 | 48.49 |
| por_Latn | 48.80 | 47.60 | 55.15 | 59.55 | 59.20 | 55.20 | 70.70 | 68.05 | 73.30 | 59.87 |
| arb_Arab | 20.40 | 21.50 | 38.22 | 27.23 | 36.10 | 31.22 | 40.84 | 39.79 | 36.10 | 34.48 |
| rus_Cyrl | 25.50 | 29.20 | 26.20 | 22.31 | 28.80 | 27.82 | 38.25 | 31.54 | 39.30 | 27.08 |
| fra_Latn | 25.50 | 33.10 | 25.72 | 29.92 | 35.20 | 31.18 | 43.96 | 37.80 | 46.20 | 28.65 |
| pes_Arab | 21.20 | 28.30 | 28.25 | 21.70 | 30.30 | 30.23 | 35.25 | 35.72 | 35.40 | 31.12 |
| deu_Latn | 17.20 | 19.90 | 28.67 | 44.88 | 26.60 | 43.30 | 57.06 | 50.83 | 49.30 | 44.94 |
| hrv_Latn | 17.90 | 30.90 | 30.25 | 25.31 | 25.90 | 26.71 | 36.42 | 34.57 | 33.30 | 29.50 |
| hun_Latn | 23.90 | 25.50 | 28.21 | 28.75 | 28.60 | 25.94 | 32.77 | 30.18 | 37.50 | 27.75 |
| ben_Beng | 27.80 | 26.00 | 31.25 | 23.88 | 35.00 | 28.30 | 46.50 | 47.38 | 49.50 | 34.46 |
| tel_Telu | 18.60 | 33.90 | 34.00 | 30.30 | 37.60 | 36.68 | 38.10 | 39.80 | 47.00 | 31.62 |
| npi_Deva | 17.50 | 25.40 | 28.57 | 14.29 | 22.20 | 16.67 | 23.02 | 27.78 | 23.80 | 25.60 |
| srp_Cyrl | 26.40 | 27.20 | 26.55 | 26.65 | 27.50 | 25.69 | 34.80 | 31.20 | 35.90 | 28.30 |
| lit_Latn | 32.60 | 35.30 | 46.32 | 48.68 | 50.30 | 40.18 | 65.15 | 69.56 | 69.10 | 53.09 |
| **avg** | 31.31 | 32.87 | 34.81 | 33.04 | 39.56 | 34.72 | 46.14 | 45.16 | 52.94 | 38.30 |

Table 22: kaleidoscope