# OpenReview forum: "Aya Vision: Advancing the Frontier of Multilingual Multimodality"
_ICLR.cc/2026/Conference — Submitted to ICLR 2026_

### Official Review · Reviewer_kmAj · 2025-10-30

**Soundness:** 3
**Presentation:** 3
**Contribution:** 3
**Rating:** 4
**Confidence:** 4

**Summary:**

This paper introduces Aya Vision, a family of open-weight 8B and 32B multilingual multimodal models (MLLMs) designed to address key challenges in non-English vision-language understanding. The authors identify two primary obstacles: (1) the scarcity of high-quality, diverse multilingual data, as simple machine translation often introduces errors and "translationese", and (2) the "catastrophic forgetting" of text-only capabilities that occurs when vision is introduced. To solve these problems, the paper proposes two main contributions: A Multilingual Multimodal Synthetic Annotation Framework and a Cross-Modal Model Merging. The paper also introduces new benchmarks, including Aya VisionBench, for evaluating open-ended multilingual generation

**Strengths:**

1. The paper’s most significant contribution is the cross-modal model merging technique. It provides an elegant, simple, and training-free solution to the well-documented problem of catastrophic forgetting.
2. The authors provide a thorough evaluation by not only testing multimodal performance but also rigorously assessing text-only capabilities, which is critical given the paper's motivation.

**Weaknesses:**

1. The paper criticizes machine translation for causing cultural bias and misalignment but still relies on a translation-based pipeline, i.e., machine translation + LLM post-editing. While this improves fluency, it fails to address the core cultural issue—a translated American-centric caption remains American-centric. Although Section P mentions limited image re-rendering, the method itself does not bridge the cultural-context gap it highlights.
2. The data-balance ablation (Figure 8) suggests issues with the quality of the synthesized multilingual data. Performance peaks at 35% multilingual data and drops at 67%, which the paper attributes to upsampling and the need for diverse English data. This implies the synthesized multilingual data is inherently less diverse or useful than English, undermining the claim of a “high-quality, diverse multilingual multimodal instruction data” framework.
3. One of my main concerns is that the paper claims to “set a new standard for multilingual performance,” yet its results do not consistently support this. In Table 1, Aya-Vision-8B scores 46.16 versus Qwen-2.5-VL-7B’s 50.00, and the gap widens at larger scales (Aya-Vision-32B: 52.81 vs. Qwen-2.5-VL-72B: 59.72). Despite claims of being “optimized for open-ended generation,” the model underperforms on structured tasks like VQA (CVQA and MTVQA) and reasoning.

**Questions:**

Please refer to the Weaknesses.

---

> ### Author Response · Authors · 2025-11-20
> **Authors' Response (1/n)**
>
> We thank the reviewer for their constructive feedback and positive assessment of our cross-modal merging technique as an “elegant, simple, and training-free” solution to catastrophic forgetting. We also appreciate your recognition of the importance of our rigorous evaluation of text-only capabilities. Below, we address your concerns regarding cultural bias in translation pipelines, the interpretation of our data ablations, and our positioning regarding state-of-the-art performance.
>
> > The paper criticizes machine translation for causing cultural bias and misalignment but still relies on a translation-based pipeline, i.e., machine translation + LLM post-editing. While this improves fluency, it fails to address the core cultural issue—a translated American-centric caption remains American-centric. Although Section P mentions limited image re-rendering, the method itself does not bridge the cultural-context gap it highlights.
>
> We agree that a translation-based pipeline alone cannot fully resolve cultural bias, and we do not claim that our method “solves” the cultural-context gap. In the introduction, we mention cultural biases and misalignments of naive MT pipelines to motivate the data quality side of our work; in this paper we focus on problems that are tractable at scale: linguistic artifacts (translationese, low diversity, literalness) and language coverage.
>
> That said, we fully acknowledge that an “American-centric” caption that is merely translated and stylistically improved can still carry an American-centric perspective. Addressing this deep cultural localization would require tasks like culturally adapted recaptioning, region-specific prompt design, and new source datasets that are natively non-US-centric, which is outside the scope of this work. Our aim here is to make existing resources less literal, more natural, and better multilingual, and to demonstrate the impact of those improvements on multilingual multimodal performance, not to claim that cultural bias in multimodal corpora is solved.
>
> > The data-balance ablation (Figure 8) suggests issues with the quality of the synthesized multilingual data. Performance peaks at 35% multilingual data and drops at 67%, which the paper attributes to upsampling and the need for diverse English data. This implies the synthesized multilingual data is inherently less diverse or useful than English, undermining the claim of a “high-quality, diverse multilingual multimodal instruction data” framework.
>
> In this ablation, the total number of training tokens is held constant and we only vary the ratio of multilingual versus English data. Moving from 0% to 35% multilingual data produces a large gain in multilingual multimodal performance, which is strong evidence that the multilingual data is both useful and complementary to English. The drop at 67% arises because, at that point, the multilingual pool must be heavily upsampled, while the much larger and more diverse English pool is significantly downsampled. This leads to redundancy on the multilingual side and reduced coverage of domains and tasks on the English side, not to an inherent lack of quality in the multilingual examples. We explain the reason in lines 463-465.

---

> ### Author Response · Authors · 2025-11-20
> **Authors' Response (2/n)**
>
> > One of my main concerns is that the paper claims to “set a new standard for multilingual performance,” yet its results do not consistently support this. In Table 1, Aya-Vision-8B scores 46.16 versus Qwen-2.5-VL-7B’s 50.00, and the gap widens at larger scales (Aya-Vision-32B: 52.81 vs. Qwen-2.5-VL-72B: 59.72). Despite claims of being “optimized for open-ended generation,” the model underperforms on structured tasks like VQA (CVQA and MTVQA) and reasoning.
>
> We appreciate the reviewer's scrutiny regarding our claim of setting a "new standard." We wish to clarify that this claim is grounded in generative, open-ended instruction following and efficiency, rather than solely on constrained academic VQA benchmarks.
>
> Our primary objective was to build a model for "fluent, instruction-following generation" across 23 languages. Standard academic benchmarks (Table 1) often rely on "constrained, multiple-choice formats" or exact short-answer matching, which do not fully capture the "nuanced, open-ended, and dynamic nature of real-world user interactions". To illustrate the limitations of strict answer matching in these evaluations, we present a representative example from the OKVQA benchmark (Question ID: 115115 [2]):
>
> The image shows two statues, and the question is: “Who designed the statues?”
>
> The set of acceptable answers in the annotation is:
> [artist,artist,artist,artist,guell,guell,toscano,toscano,aritect,aritect]
> (no, this is not a typo on our end)
>
> We would like to highlight that if a model outputs “an architect” to answer this question, it would be marked incorrect. This issue is prevalent across many VQA datasets. If the field exclusively prioritizes hill-climbing on such strict-matching metrics, we risk developing VLMs that are optimized for benchmarks yet detached from the goal of being useful conversational assistants. This trade-off is a primary reason why our scores on strict academic benchmarks differ from our strong performance on open-ended generative evaluations.
>
> To further validate that our results translate to real-world human preference – and are not artifacts of proprietary LLM-based judging – we worked with LMSys to list the final Aya Vision models on the LMSys Chatbot Arena. The resulting ELO scores from blind human voting strongly support our claims:
>
> Aya Vision 32B: 1097 (outperforming Molmo 72B at 1076 and Llama-3.2-90B at 1069).\
> Aya Vision 8B: 1069 (matching Llama-3.2-90B despite being 10x smaller).
>
> Finally, the comparison to Qwen-2.5-VL-72B must be contextualized by scale. Qwen-2.5-VL-72B is more than 2x larger than Aya-Vision-32B. Qwen-2.5-VL models are trained on approximately 4 trillion tokens [1], while Aya Vision models are trained on less than 40 billion tokens – a nearly 100x reduction. Consequently, Aya Vision 32B utilizes two orders of magnitude less compute than the Qwen-2.5-VL models. When these factors are taken into account, we believe Aya Vision demonstrates extreme competitiveness relative to its compute class.
>
> Aya Vision 32B uses a total of 8192 H100 GPU hours (Appendix I). On 128 GPUs, this takes 64 hours (2.66 days). Assuming the same throughput and expanding the number of concurrent GPUs to 1024, Training Qwen2.5-VL-72B would approximately take $64\times100/(8\times24)$ days = 33.33 days.
>
> [1] - Qwen2.5-VL Technical Report (https://arxiv.org/abs/2502.13923)
>
> [2] - https://huggingface.co/datasets/lmms-lab/OK-VQA/viewer/default/val2014?row=0&views%5B%5D=val2014&sql=SELECT+*%0AFROM+val2014%0AWHERE+question_id+%3D+%27115115%27%0ALIMIT+1%3B&sql_row=0

---

> ### Author Response · Authors · 2025-11-27
>
> We appreciate your insightful feedback. We have made a strong effort to address the points you have raised and will ensure all updates are reflected in the camera-ready manuscript. We believe these revisions would significantly strengthen our paper. We hope our responses resolve your reservations, and we would be grateful if you would consider raising your score. We are happy to answer any further questions.

---

### Official Review · Reviewer_55FY · 2025-10-31

**Soundness:** 2
**Presentation:** 1
**Contribution:** 2
**Rating:** 2
**Confidence:** 4

**Summary:**

The paper introduces AyaVision, multilingual multimodal LLMs (8B/32B) that handle up to 23 languages. The motivation is to address three challenges: (i) data scarcity and translationese to train multilingual MLLMs, addressed via a synthetic data engine that enriches the annotations with additional context about he images, (ii) text-only capability loss after adding vision, which is addressed via a training-free cross-modal weight-merging method, (iii) lack of open-ended multilingual evaluation, addressed via new benchmarks (AyaVisionBench, m-WildVision and xChatBench). The paper reports strong preference win rates on the proposed open-ended AyaVisionBench.

**Strengths:**

- The paper makes a meaningful data contribution by using a pipeline that combines distillation, filtering, and context-aware rephrasing to reduce translation artefacts. The data engine also increases lexical diversity and sequence length of instruction pairs, which is quantified by gains and results in an automatic quality improvement on COMET.

- The paper also makes a meaningful benchmark contribution, it proposes three benchmarks: AyaVisionBench,  m-WildVision and xChatBench offer open-ended, multimodal, multilingual evaluation. This extends evaluation beyond multiple-choice or english-only setups and should help standardize evaluation for future multilingual multimodal work.

- The training-free cross-modal merging method is simple and effective, as shown in the ablations. The reported deltas show large recovery on text-only win rates and non-trivial gains on multimodal win rates.

**Weaknesses:**

- The model shows large gains on the two proposed benchmarks, AyaVisionBench and m-WildVision, measured as open-ended pairwise win rates averaged across 23 languages. However, on standard accuracy-scored benchmarks, its performance is clearly below strong baselines. The paper does not explain why a model that improves under open-ended evaluations falls behind by 6 to 17 points on benchmarks like xMMMU, CVQA, and MTVQA. If this gap comes from answer-format constraints (for example, multiple choice or one-word responses) or a mismatch between free-form outputs and the expected scoring format, the paper should include a simple normalization step that maps free-form text to the required label, for example, an LLM-based post-processing that converts a sentence to option B or to the exact short answer. Without such a comparison, it is hard to tell whether the wins on AyaVisionBench reflect alignment with the response style used in training and evaluation, rather than improvements that transfer to widely used benchmarks.

- The paper mixes several motivations, but the evidence for each one is narrow. The paper claims translationese in training data is a major problem in data annotation methods of existing multilingual MLLM and proposes translation plus LLM rephrasing. Yet it does not show that prior datasets have this issue (quantitatively or qualitatively), nor that the proposed method fixes it beyond a COMET jump. There is no human verification/alignment and no comparison to a simple off-the-shelf LLM translation, so it is unclear whether any benefit comes from the proposed method or from the tools used. The paper also argues that text-only performance worsens when additional languages are added in multilingual multimodal performance and uses linear weight merging, but this relation is not explained, and there is no evidence that higher text-only scores lead to higher multimodal accuracy. Altogether, the motivations feel mixed.

- The citation format does not match the official ICLR citation requirement.

Overall, the main technical contribution is not supported by clear evidence, and its novelty is limited. The method is a simple linear weight mix between the text-only LLM and the multimodal model while keeping vision modules fixed. The paper does not examine basic design alternatives or stronger baselines that would test whether this is the right choice, nor does it provide clear guidance on when and why the method helps. At the same time, the performance improvements are only seen in the pairwise LLM-judge win rates on proposed open-ended benchmarks, while scores on widely used benchmarks are lower than the baselines.

**Questions:**

Please explain why the scores on CVQA, MTVQA, and xMMMU are lower. Is the main cause answer-format mismatch, short-answer matching, domain differences, or lower model accuracy? If it is mostly formatting, please say whether a small LLM post-processing step that maps free-form outputs to the exact expected label would raise the scores, and by roughly how much.

---

> ### Author Response · Authors · 2025-11-20
> **Authors' Response (1/n)**
>
> We thank Reviewer 55FY for their detailed feedback. We are encouraged that you found our synthetic annotation framework to be a “meaningful data contribution” and recognized the value of AyaVisionBench in helping to “standardize evaluation” for the multilingual community. We also appreciate your assessment of our cross-modal merging method as “simple and effective.”
>
> > The model shows large gains on the two proposed benchmarks, AyaVisionBench and m-WildVision, measured as open-ended pairwise win rates averaged across 23 languages. However, on standard accuracy-scored benchmarks, its performance is clearly below strong baselines. The paper does not explain why a model that improves under open-ended evaluations falls behind by 6 to 17 points on benchmarks like xMMMU, CVQA, and MTVQA.
>
> Aya Vision does not surpass the strongest baselines such as Qwen-2.5-VL-72B on certain academic benchmarks like CVQA, MTVQA, and xMMMU. However, the 6-17 point gaps highlighted by the reviewer arise from comparisons to models that are substantially larger or newer. First, Qwen-2.5-VL-72B has more than twice as many parameters as Aya Vision 32B. Second, performance on benchmarks like xMMMU is heavily dependent on the reasoning capabilities inherited from the base LLM; Qwen-2.5-72B is a significantly newer model compared to Aya Expanse 32B, possessing stronger inherent reasoning capabilities. Third, regarding data scale, Qwen-2.5-VL models are trained on approximately 4T tokens [1], while Aya Vision models are trained on less than 40B tokens, a nearly 100x reduction.
>
> As mentioned in Appendix I, Aya Vision 32B uses a total of 8192 H100 GPU hours. On 128 GPUs, this takes 64 hours (2.66 days). Assuming the same throughput and expanding the number of concurrent GPUs to 1024, Training Qwen2.5-VL-72B would approximately take $64\times100/(8 \times 24)$ days = $33.33$ days.
>
> When comparisons are made within a similar size and compute class, Aya Vision 8B and 32B are on par with or slightly below similarly sized open baselines such as Qwen-2.5-VL-7B on these accuracy-style benchmarks, despite using far less multilingual multimodal training data. In this sense, Aya Vision is highly competitive relative to its resource budget and size.
>
> [1] - Qwen2.5-VL Technical Report (https://arxiv.org/abs/2502.13923)
>
> > If this gap comes from answer-format constraints (for example, multiple choice or one-word responses) or a mismatch between free-form outputs and the expected scoring format, the paper should include a simple normalization step that maps free-form text to the required label, for example, an LLM-based post-processing that converts a sentence to option B or to the exact short answer.
>
> > Please explain why the scores on CVQA, MTVQA, and xMMMU are lower. Is the main cause answer-format mismatch, short-answer matching, domain differences, or lower model accuracy? If it is mostly formatting, please say whether a small LLM post-processing step that maps free-form outputs to the exact expected label would raise the scores, and by roughly how much.
>
> Aya Vision is explicitly designed for "fluent, instruction-following generation" and "open-ended interactions" prioritizing conversational quality over the strict, constrained formats required by traditional benchmarks. To illustrate the limitations of strict answer matching in these evaluations,consider a representative example from the OKVQA benchmark (Question ID: 115115) [1]:\
> The image shows two statues, and the question is: “Who designed the statues?”
>
> The set of acceptable answers in the annotation is: [artist,artist,artist,artist,guell,guell,toscano,toscano,aritect,aritect]
> (no, this is not a typo on our end)
>
> A model response such as “an architect” would be marked incorrect under the official evaluation, despite being semantically consistent with several of the annotated answers. This is just the tip of the iceberg, where various VQA datasets are filled with ambiguous questions expecting specific answers validated by regex matching, which often penalize reasonable paraphrases or more natural, assistant-style responses.
>
> These benchmarks are useful for measuring certain capabilities, but if they are treated as the primary signal for progress, models are incentivized to optimize for narrowly formatted outputs instead of informative, conversational answers. Our goal with Aya Vision is to move toward the latter. This is precisely why we complement traditional accuracy-based evaluations with open-ended, pairwise evaluations on AyaVisionBench and m-WildVision, which are designed to better capture the quality and usefulness of assistant-style multilingual vision-language generation.
>
> [1] https://huggingface.co/datasets/lmms-lab/OK-VQA/viewer/default/val2014?row=0&views%5B%5D=val2014&sql=SELECT+*%0AFROM+val2014%0AWHERE+question_id+%3D+%27115115%27%0ALIMIT+1%3B&sql_row=0)

---

> ### Author Response · Authors · 2025-11-20
> **Authors' Response (2/n)**
>
> >  it is hard to tell whether the wins on AyaVisionBench reflect alignment with the response style used in training and evaluation, rather than improvements that transfer to widely used benchmarks.
>
> We do not agree that we only measure assistant-style performance, though we intentionally prioritize it. Aya Vision is evaluated on many standard accuracy-based benchmarks as well, but AyaVisionBench and m-WildVision are designed to specifically capture assistant-style, open-ended generation.
> AyaVisionBench is curated from the existing Cauldron test set: we reuse its images and attach multilingual, long-form prompts. The images are not generated by Aya Vision, and all models are evaluated under the same VLM-as-a-judge protocol. m-WildVision is a multilingual extension of WildVision-Bench, which is already widely used to evaluate vision–language models. We preserve the original images and task structure and add carefully curated multilingual prompts.
> Both AyaVisionBench and m-WildVision are therefore objective benchmarks built on established image sets, not tailored to Aya Vision’s training data or style. The gains we observe on these two benchmarks are also consistent with improvements on a broad range of external multimodal and multilingual benchmarks, as reported in Table 1, Table 2, and Figure 8 (right), indicating that the benefits of Aya Vision are not confined to our own evaluations.
>
> > The paper mixes several motivations, but the evidence for each one is narrow. The paper claims translationese in training data is a major problem in data annotation methods of existing multilingual MLLM and proposes translation plus LLM rephrasing. Yet it does not show that prior datasets have this issue (quantitatively or qualitatively), nor that the proposed method fixes it beyond a COMET jump. There is no human verification/alignment and no comparison to a simple off-the-shelf LLM translation, so it is unclear whether any benefit comes from the proposed method or from the tools used. The paper also argues that text-only performance worsens when additional languages are added in multilingual multimodal performance and uses linear weight merging, but this relation is not explained, and there is no evidence that higher text-only scores lead to higher multimodal accuracy. Altogether, the motivations feel mixed.
>
> In the paper, we quantify the impact of our recaptioning and multilingual pipeline at the data level, not just with a single COMET number. After recaptioning, the average word count increases from 14.2 to 100.1, token count from 27.2 to 140.8, and lexical diversity (MTLD) rises from 11.0 to 61.2, approaching the variability observed in fluent human writing (Section 2.2).
>
> For multilingual data, we compare NLLB alone to NLLB + LLM rephrasing with COMET and observe consistent improvements across all 23 languages, from 0.75 to 0.83 on average, as reported in Table 7 (Appendix O). This goes beyond a single global score and documents per-language gains.
>
> At the model level, Figure 9 shows that data improvements (which include the recaptioning and multilingual pipeline) produce a 17.2 point increase in multilingual vision win rate over the unprocessed baseline, before any merging or scaling. This directly quantifies the benefit of training on processed versus unprocessed data.
>
> These three pieces: lexical statistics, COMET improvements, and system-level gains are the main evidence we provide that the pipeline improves over naive MT-only data. We agree that a human evaluation would be a valuable addition, but given the scale (23 languages) and cost, we rely on standard automatic metrics and downstream performance as a pragmatic compromise. We will clarify in the text that we do not claim a human-rated or absolute solution to translationese, but a practical, empirically supported improvement.
>
> > The citation format does not match the official ICLR citation requirement.
>
> We acknowledge this and will correct the citation format to comply with ICLR requirements in the final manuscript.

---

> ### Author Response · Authors · 2025-11-20
> **Authors' Response (3/n)**
>
> > Overall, the main technical contribution is not supported by clear evidence, and its novelty is limited. The method is a simple linear weight mix between the text-only LLM and the multimodal model while keeping vision modules fixed. The paper does not examine basic design alternatives or stronger baselines that would test whether this is the right choice, nor does it provide clear guidance on when and why the method helps. At the same time, the performance improvements are only seen in the pairwise LLM-judge win rates on proposed open-ended benchmarks, while scores on widely used benchmarks are lower than the baselines.
>
> We respectfully disagree with the assessment that the main technical contribution is unclear or only weakly supported. As outlined in the Introduction, our goal is to build a competitive multilingual multimodal assistant under realistic compute constraints, and the paper makes three concrete contributions toward this:
> 1) A multilingual multimodal synthetic annotation framework
> 2) A simple but effective cross-modal merging strategy
> 3) A family of open-weight models and benchmarks release.
>
> These components are evaluated on a broad suite of external multimodal and text-only benchmarks, not only on our proposed pairwise win-rate evaluations, and together they form a coherent approach.
>
> As for merging, our goal is not to introduce a new generic merging algorithm, but to show that a simple, training-free cross-modal interpolation between a multilingual text-only backbone and its multilingual multimodal continuation is an effective way to mitigate catastrophic forgetting in a realistic low-compute setting.

---

> ### Author Response · Authors · 2025-11-27
>
> We appreciate your insightful feedback. We have made a strong effort to address the points you have raised and will ensure all updates are reflected in the camera-ready manuscript. We believe these revisions would significantly strengthen our paper. We hope our responses resolve your reservations, and we would be grateful if you would consider raising your score. We are happy to answer any further questions.

---

### Official Review · Reviewer_6eZf · 2025-10-31

**Soundness:** 1
**Presentation:** 2
**Contribution:** 2
**Rating:** 2
**Confidence:** 4

**Summary:**

The paper introduces Aya Vision, a family of multilingual multimodal large language models (8B and 32B parameters) designed to handle both image and text inputs across 23 languages. The authors propose two main ideas: **(1)** a synthetic multilingual data generation and translation pipeline, and **(2)** a cross-modal model merging technique to preserve text-only performance after multimodal training. The paper also presents AyaVisionBench, a new benchmark for evaluating multilingual multimodal models. Experimental results show Aya Vision performing reasonably against several existing open-weight models and stronger on its own proposed benchmark.

**Strengths:**

The following are the strengths of the paper:

1. The paper tackles an important problem of extending multimodal models beyond English, making progress toward inclusive and global AI systems.

2. The effort to maintain text-only capability while introducing visual understanding is valuable, as it tries to address the catastrophic forgetting issue common in multimodal training. The cross-model merging approach is interesting and if works in general, could be very useful for MLLM post training.

**Weaknesses:**

The following are the weaknesses of the paper:

1. Many important implementation details are missing or unclear. For example, how dataset categories were selected and sampled to reach 2.29M samples (lines 126-128) is not explained. The process of “regularization” and sampling remains undefined.

2. The keyword-based filtering step is unclear. The “curated list of keywords” is never provided, making it unclear how such filtering can reliably detect errors in generated descriptions (refer to lines 156-160).

3. In Stage 2: LLM-based semantic filtering, the LLM-based semantic filtering is a weak proxy since the LLM cannot see images. As stated, complex samples are often discarded, which limits the model’s ability to learn complex reasoning tasks.

4. The hybrid translation pipeline lacks justification. Simply chaining NLLB and command-r-plus does not guarantee better quality, and the paper does not explain why this choice was made over stronger multilingual multimodal translation models such as LLaMA-4 or Qwen3. The subset of data chosen for translation (lines 182–183) is also not described.

5. There seems to be data size inconsistencies.  Synthetic re-annotation (3.5M of 2.29M original) and downsampled original data (3.7M from 6M) do not match the claimed final training set size of 2.75M samples. How is this 2.75M number exactly calculated?

6. The cross-model merging method is interesting but conceptually unclear. Theoretically, the text-only and multimodal models occupy different weight spaces, and the paper does not justify how linear interpolation of weights can work reliably.

7. AyaVisionBench cannot be considered a major contribution. All its details are buried in the appendix, and it is small, only 135 samples per language (around 3K total). The comparison with other multilingual multimodal benchmarks is missing. In fact, stronger public benchmarks like “All Languages Matter (ALM-Bench) [1]” already exist, covering 100 languages and 22K samples.

8. Aya Vision performs poorly on academic and structured benchmarks (Table 1), with models such as Qwen2.5-VL outperforming it consistently. Its strong results appear only on its own benchmark, which weakens the claim of “best-in-class performance.”

9. Many components depend on proprietary LLMs (command-r-plus, GPT-based judges) and claim “human-preferred” outputs without any real human evaluation. For example, as show in Appendix K, simply asking LLM/MLLM to generate "human-preferred" does not mean that the resulting text is "human-preferred".

Overall, the paper lacks clarity, key methodological details, and fair validation. The improvements shown are narrow and not convincingly demonstrated beyond the proposed benchmark.

---

[1] Vayani, A., Dissanayake, D., Watawana, H., Ahsan, N., Sasikumar, N., Thawakar, O., Ademtew, H. B., Hmaiti, Y., Kumar, A., Kuckreja, K., et al. (2024). All languages matter: Evaluating LMMs on culturally diverse 100 languages. CVPR 2025.

**Questions:**

Please refer to the Weaknesses section for detailed points requiring clarification.

---

> ### Author Response · Authors · 2025-11-20
> **Authors' Response (1/n)**
>
> We thank Reviewer 6eZf for their review and for recognizing that our work tackles the “important problem” of extending multimodal models beyond English, representing a step toward “inclusive and global AI systems.” We are also glad that you found the cross-modal merging approach “interesting” and potentially useful for the field. We appreciate your detailed questions regarding implementation specifics and validity, which we address below to provide the necessary clarity.
>
> >Many important implementation details are missing or unclear. For example, how dataset categories were selected and sampled to reach 2.29M samples (lines 126-128) is not explained. The process of “regularization” and sampling remains undefined.
>
> Due to page limits, the main text focuses on the core methodology, while implementation details, including data construction, are provided in the Appendix.
>
> Concretely, we first aggregate multiple vision-language datasets and map all examples into a set of high-level task categories (VQA, captioning, document/OCR, charts/tables, reasoning, etc.). For each category, we define a target size on the number of English samples. We then randomly shuffle the data within each category and uniformly downsample categories that exceed their target size, while keeping all samples for categories below the target size; the sum across them gives the 2.29M curated English examples.
>
> In this context, regularization refers to balancing the sample counts across categories via these per-category targets and uniform sampling, so that no single overrepresented task dominates the training distribution. The full per-category statistics and sampling procedure are documented in the appendix D, where we systematically collect all details (Figure 10, Table 3).
>
>
> >The keyword-based filtering step is unclear. The “curated list of keywords” is never provided, making it unclear how such filtering can reliably detect errors in generated descriptions (refer to lines 156-160).
>
> We appreciate the reviewer’s comment. We acknowledge that keyword-based filtering alone is not sufficient for reliably identifying errors in generated descriptions. This is precisely why our pipeline does not rely solely on keywords for error detection. Instead, keyword filtering serves only as an initial screening step. The curated keyword list targets a small set of easily identifiable, undesirable response types. These include:
> - responses in which the model explicitly states it is performing a “recaptioning” or related task, or otherwise restates the instruction instead of completing it (e.g., “my task is to…”).
> - responses that reveal the presence of a reference answer, such as mentions of “ground truth”,  “reference answer”, or “gold answer” in the output
> - responses that trigger safety-related refusals, for example, those beginning with “I’m unable to…” or “I’m sorry…”.
> - responses that do not follow formatting instructions, such as those omitting the required \<answer\> … \</answer\> tags.
>
> These surface-level cues are easily detectable with lightweight keyword rules. The actual assessment of correctness is performed by the following semantic filtering stage, which compares the model’s output with the ground-truth description to ensure that the final decision is as accurate as possible.

---

> ### Author Response · Authors · 2025-11-20
> **Authors' Response (2/n)**
>
> >In Stage 2: LLM-based semantic filtering, the LLM-based semantic filtering is a weak proxy since the LLM cannot see images. As stated, complex samples are often discarded, which limits the model’s ability to learn complex reasoning tasks.
>
> Stage 2 is not intended to judge image correctness directly, but to verify that the rephrased caption preserves the semantics of the original, image-grounded caption and ground-truth answer. In this stage, command-r-plus-08-2024 only sees the **original caption + ground-truth answer** and the **rephrased caption**, and acts as a semantic judge: it checks whether the answer to the original caption remains valid given the rephrased version.
>
> Because the original captions are produced by an image-aware teacher (and aligned with a ground-truth answer), Stage 2 is a high-precision consistency check to prevent the text-only rephrasing step from introducing semantic drift. **The overall rejection rate is 3.2% of recaptioned data (62,370 samples)**, with reasoning tasks showing a 4.6% error rate versus 2.5% for simpler VQA, reflecting that complex tasks are more fragile, not that complexity itself is penalized. The vast majority of complex reasoning examples are retained, and in practice the resulting models perform strongly on document, chart/table, and multi-step reasoning benchmarks, suggesting that coverage of complex tasks remains substantial despite this conservative filtering.
>
> In summary, Stage 2 is a text-only semantic safeguard layered on top of an image-aware teacher, trading a small fraction of noisy or inconsistent complex samples for cleaner supervision, rather than systematically removing complex reasoning data.

---

> ### Author Response · Authors · 2025-11-20
> **Authors' Response (3/n)**
>
> >The hybrid translation pipeline lacks justification. Simply chaining NLLB and command-r-plus does not guarantee better quality, and the paper does not explain why this choice was made over stronger multilingual multimodal translation models such as LLaMA-4 or Qwen3. The subset of data chosen for translation (lines 182–183) is also not described.
>
> We thank the reviewer for this insightful question and would like to clarify the design choice behind our multilingual data pipeline. Although using large language models directly for machine translation (MT) is a straightforward option, prior studies have shown that LLMs only perform competitively on high-resource languages but still face challenges with low-resource languages [3, 4]. For instance, as reported in a related multilingual multimodal study [1], issues such as incorrect punctuation spacing, omission or misapplication of Arabic grammatical features (e.g., nunation), and untranslated English words are common when using LLMs alone. We observed similar issues during our own experiments.
>
> On the other hand, MT models like NLLB demonstrate more consistent performance across languages but are not without limitations. Notably, translation artifacts such as translationese frequently occur. Our pipeline therefore adopts a **hybrid approach**: NLLB first produces a high-coverage, semantically faithful translation, and a multilingual LLM then rephrases it in context to improve fluency and naturalness while preserving meaning. Recent work [4, 5] suggests that combining MT systems with LLMs, using MT outputs as in-context references for rephrasing, can indeed enhance translation quality, especially for low-resource languages, which motivates our design.
>
> Moreover, given the known tendency for models to favor their own generations when evaluated on data they produced [6,7], we deliberately avoided using an LLM for translation in our evaluation data to prevent bias.
>
> **The effectiveness of this pipeline is quantified both at the model and data levels.** Figure 9 shows that our data improvements (including the translation + rephrasing pipeline) yield a **17.2% increase** in multilingual vision win rate compared to training on unprocessed data, directly quantifying the benefit of the processed multilingual corpus. At the data level, the pipeline expands average caption length from 27.2 to 140.8 tokens and increases lexical diversity from 11.0 to 61.2 (lines 143–148, p.3), creating richer supervision. Translation quality, measured with COMET, improves from an average of 0.75 (NLLB alone) to 0.83 after post-editing. Table 7 in Appendix O reports reference-free COMET scores by language and shows consistent gains across all 23 languages, with relative improvements ranging from **1.8% for Hindi to 80.3% for Traditional Chinese**. We believe these systematic improvements in translation and caption quality are an important driver of Aya Vision’s stronger multilingual performance.
>
> Finally, we would like to bring to the reviewer’s notice that this work was released on 4 March 2025, whereas Llama 4 was released on 5 April 2025 and Qwen3 on 29 April 2025. Thus, it was not possible to use these models in our pipeline. In addition, Llama models are released under a license that restricts using Llama generated data to train or improve other open models, which further limits their suitability as a general-purpose data generator for open-weight systems like Aya Vision.
>
> **Regarding the subset of data chosen for translation**: to avoid excessive duplication and overfitting, we do not translate all English examples into every language. Instead, for each target language we set a per-language translation budget and **uniformly sample** from the pool of English image-text pairs, prioritizing tasks and datasets covered by our multilingual evaluation suite. This yields a balanced multilingual dataset that avoids repeatedly translating near-duplicate English samples and keeps the overall training cost tractable. (These are specified between lines 182-188.)
>
> [1] PALO: A Polyglot Large Multimodal Model for 5B People\
> [2] Aya Model: An Instruction Finetuned Open-Access Multilingual Language Model\
> [3] How Good Are GPT Models at Machine Translation? A Comprehensive Evaluation\
> [4] Multilingual Machine Translation with Large Language Models: Empirical Results and Analysis\
> [5] Leveraging GPT-4 for Automatic Translation Post-Editing\
> [6] LLM Evaluators Recognize and Favor Their Own Generations, 2024.\
> [7] LLM see, LLM do: Leveraging active inheritance to target non-differentiable objectives, 2024.

---

> ### Author Response · Authors · 2025-11-20
> **Authors' Response (4/n)**
>
> >There seems to be data size inconsistencies. Synthetic re-annotation (3.5M of 2.29M original) and downsampled original data (3.7M from 6M) do not match the claimed final training set size of 2.75M samples. How is this 2.75M number exactly calculated?
>
> The final training set size is the number of sequence-packed samples (as specified in line 217). Sequence packing is an established method to reduce the number of padding tokens during training by concatenating various samples into a single longer sequence (with additional book keeping to prevent attention leakage) [1] - increasing the signal density and computational efficiency. Since, multiple samples are now packed into a single sequence, the number of sequences is fewer than the number of samples. Our implementation uses a heap-based approach to reduce the number of padding tokens.
>
> [1] https://docs.nvidia.com/nemo-framework/user-guide/24.09/nemotoolkit/features/optimizations/sequence_packing.html
>
> >The cross-model merging method is interesting but conceptually unclear. Theoretically, the text-only and multimodal models occupy different weight spaces, and the paper does not justify how linear interpolation of weights can work reliably.
>
> While we show the benefits of cross-modal merging through an empirical lens, a deep dive into the theoretical mechanism enabling this behaviour is out of the scope of this work. The theory behind the surprising effectiveness of model merging in a single modality itself is still an active area of research. Some recent works such as "Model Soups" (Wortsman et al., 2022) and "Linear Mode Connectivity and the Lottery Ticket Hypothesis" (Frankle et al., 2020), has shown that one can often find a linear path between two such solutions (i.e., W_text-LLM and W_mm-LLM) where the loss remains consistently low. Our linear interpolation (α * W_mm-LLM + (1-α) * W_text-LLM) is explicitly traversing this low-loss path. Finding an optimal α is equivalent to finding the "sweet spot" on this path that offers the best trade-off of specialized capabilities. The existence of these paths is a key enabler for successful model merging. We argue that the text-only and multi-modal models are not completely disjoint in the weight space – since the Multimodal model is initialized from the text model.

---

> ### Author Response · Authors · 2025-11-20
> **Authors' Response (5/n)**
>
> >AyaVisionBench cannot be considered a major contribution. All its details are buried in the appendix, and it is small, only 135 samples per language (around 3K total). The comparison with other multilingual multimodal benchmarks is missing. In fact, stronger public benchmarks like “All Languages Matter (ALM-Bench) [1]” already exist, covering 100 languages and 22K samples.
>
> We thank the reviewer for the opportunity to clarify the role and design of AyaVisionBench, and to position it relative to existing multilingual multimodal benchmarks.
>
> AyaVisionBench is not intended as “just another” generic VQA benchmark. Its goal is to evaluate multilingual, long-form, assistant-style generation rather than short, rigid formats. Many existing multilingual multimodal benchmarks primarily focus on formats such as multiple choice, single-word or short-phrase VQA, or yes/no questions. In contrast, AyaVisionBench pairs images with open-ended prompts that require models to produce fluent, multi-sentence responses, allowing us to directly measure diversity, coherence, and instruction-following behavior in a vision–language assistant setting. This aligns with the broader shift in NLP from optimizing for rigid automatic metrics to evaluating how well systems behave as conversational assistants.
>
> Regarding the comparison to ALM-Bench: the closest component to AyaVisionBench is the long-form subset of ALM-Bench highlighted by the reviewer. However, this subset contains 8716 samples across 100 languages (~87 examples per language), while AyaVisionBench provides 135 examples per language, roughly 1.6x more long-form prompts per language. AyaVisionBench is thus complementary: it trades breadth in the number of languages for deeper coverage per language, with carefully curated prompts and images that required substantial expert effort. Given the annotation cost of multilingual long-form, image-grounded evaluation, we prioritize per-language depth and quality over scale alone. We will make sure to cite All Languages Matter (ALM-Bench) and incorporate this discussion in the manuscript.
>
> On the question of “buried in the appendix”, due to page limits we describe AyaVisionBench at a high level in the main text and place the full specification (prompt design, domains, and per-language statistics) in the appendix, where we systematically collect implementation and dataset details. We will make this linkage clearer by explicitly pointing readers from the main text to the corresponding appendix section when discussing AyaVisionBench.
>
> Overall, we view AyaVisionBench as a targeted, high-quality evaluation resource focused on multilingual, long-form, assistant-style vision-language generation, designed to complement broader multilingual multimodal benchmarks like ALM-Bench rather than replace them
>
> >Aya Vision performs poorly on academic and structured benchmarks (Table 1), with models such as Qwen2.5-VL outperforming it consistently. Its strong results appear only on its own benchmark, which weakens the claim of “best-in-class performance.”
>
> We worked with LMSys to list the final Aya Vision models on the LMSys Arena to verify whether our results of Aya Vision models generating human preferred responses held true.
> At the time of testing, the following were the ELO scores obtained from LMSys:
> | Model             | Score |
> |-------------------|-------|
> | Aya Vision 32B    | 1097  |
> | Molmo 72B         | 1076  |
> | Claude 3 Opus     | 1076  |
> | Gemini Flash 1.5  | 1072  |
> | Pixtral 12B       | 1072  |
> | Llama3.2-90B      | 1069  |
> | Aya Vision 8B     | 1069  |
> | InternVL2 26B     | 1067  |
>
> Since Qwen2.5VL wasn’t listed on LMSys Arena, it is not possible to get a head-to-head comparison; however, Llama3.2-90B helps ground our discussion. Aya Vision 8B gets the same average ELO score as Llama3.2-90B even though Llama3.2-90B is 10x larger and scores much higher on “traditional VQA type benchmarks”. Aya Vision’s performance on LMSys Arena strengthens our argument that our training process indeed primes the model towards human-preferred responses.
>
> Additionally, while the multimodal results take center stage in the discourse, we would like to reiterate that Aya Vision also retains excellent text-only performance due to our cross-modal merging technique. Aya Vision 8B comprehensively beats other multimodal models in both academic and preference evaluations (Table 2 and Figure 5).

---

> ### Author Response · Authors · 2025-11-20
> **Authors' Response (6/n)**
>
> >Many components depend on proprietary LLMs (command-r-plus, GPT-based judges) and claim “human-preferred” outputs without any real human evaluation. For example, as show in Appendix K, simply asking LLM/MLLM to generate "human-preferred" does not mean that the resulting text is "human-preferred".
>
> Thank you for giving us a chance to clarify these points. First of all, we’d like to clarify the usage of proprietary models. For recaptioning, we use an open-weight multimodal teacher model, prompted with task-specific templates. For translation, we rely on NLLB-3.3B, an open-source model, to translate English data into 22 target languages. For post-editing and semantic filtering, we use command-r-plus-08-2024, which is released as an open-weights model under a non-commercial research license. Thus, the core components of our synthetic annotation pipeline are open-weight models rather than black-box proprietary APIs. We will clarify this more explicitly in the data section and appendices, and emphasize that the pipeline is model-agnostic: practitioners can swap our choices for other open models depending on their licensing and cost constraints.
>
> We whole-heartedly agree with the reviewer that simply asking an LLM to generate “human-preferred” samples does not imply the resulting text is human-preferred. As the goal with Aya Vision models is to actually build a multilingual multimodal assistant that generates human-preferred responses, we rely on LLM-as-judge style evaluations as a cost-efficient proxy.
> We worked with LMSys to list the final Aya Vision models on the LMSys Arena to verify whether our results of Aya Vision models generating human preferred responses held true.
> At the time of testing, the following were the ELO scores obtained from LMSys:
> | Model             | Score |
> |-------------------|-------|
> | Aya Vision 32B    | 1097  |
> | Molmo 72B         | 1076  |
> | Claude 3 Opus     | 1076  |
> | Gemini Flash 1.5  | 1072  |
> | Pixtral 12B       | 1072  |
> | Llama3.2-90B      | 1069  |
> | Aya Vision 8B     | 1069  |
> | InternVL2 26B     | 1067  |
>
> Since Qwen2.5VL wasn’t listed on LMSys Arena, it is not possible to get a head-to-head comparison; however, Llama3.2-90B helps ground our discussion. Aya Vision 8B gets the same average ELO score as Llama3.2-90B even though Llama3.2-90B is 10x larger and scores much higher on “traditional VQA type benchmarks”. Aya Vision’s performance on LMSys Arena strengthens our argument that our training process indeed primes the model towards human-preferred responses.

---

> ### Author Response · Authors · 2025-11-20
> **Authors' Response (7/n)**
>
> >Overall, the paper lacks clarity, key methodological details, and fair validation. The improvements shown are narrow and not convincingly demonstrated beyond the proposed benchmark.
>
> We respectfully disagree with the assessment that the paper lacks clarity, key methodological details, and fair validation, and we clarify these points below.
>
> Given the strict page limits, the main text is intentionally focused on the overall framework (multilingual data pipeline, multimodal training, and cross-modal merging), while implementation details are systematically collected in the appendix. Concretely:
>
> - Dataset construction and sampling strategy are described in Appendix D.
> - Evaluation and benchmark details in Appendix E, Q.
> - Aya Vision's architecture and training details in Appendix F, hyperparameter details in Appendix G.
> - The cross-modal merging procedure and its ablations appear in Section 7 / Appendix H.
> - Recaptioning, keyword filtering, and LLM-based semantic filtering are detailed in Appendix K, L, M, N.
> - The multilingual translation + rephrasing pipeline, including COMET evaluation by language, is covered in Appendix O, P.
>
> Together with our released code, weights and data artifacts, these sections provide the information needed to reproduce the work.
>
> The improvements we report are not limited to AyaVisionBench. Aya Vision is evaluated on a diverse suite of benchmarks, including:
>
> Multimodal benchmarks: xMMMU, MaXM, CVQA, MTVQA, Kaleidoscope, document/table understanding, and others, covering a range of visual reasoning skills and domains.
> Multilingual text-only benchmarks: MGSM, Global MMLU-Lite, FLORES, and IFEval, which assess reasoning, knowledge, and instruction-following across languages.
>
> Across these benchmarks, Aya Vision consistently matches or outperforms strong open-weight and commercial baselines under standardized prompting and decoding settings. We also calibrate and justify our VLM-as-judge setup and cross-check trends against reference-based metrics, as described in the paper and in our other responses.
>
> AyaVisionBench is designed as a complementary, long-form, assistant-style evaluation to emphasize conversational usefulness, but it is not the sole evidence of improvement. The multilingual data pipeline (recaptioning + translation + rephrasing) yields systematic gains over unprocessed data across multiple multimodal and text benchmarks, and the cross-modal merging strategy improves both multilingual text performance and multilingual multimodal win rates compared to no merging or simply adding more text-only SFT. These trends hold across several independent benchmarks, not just our proposed one.
>
> In summary, while the main text necessarily abstracts some implementation details for space, the full paper (main sections + appendix) provides a complete methodological description and broad empirical validation that extends well beyond AyaVisionBench.

---

> ### Author Response · Authors · 2025-11-27
>
> We appreciate your insightful feedback. We have made a strong effort to address the points you have raised and will ensure all updates are reflected in the camera-ready manuscript. We believe these revisions would significantly strengthen our paper. We hope our responses resolve your reservations, and we would be grateful if you would consider raising your score. We are happy to answer any further questions.

---

### Official Review · Reviewer_ibxe · 2025-11-01

**Soundness:** 4
**Presentation:** 4
**Contribution:** 4
**Rating:** 8
**Confidence:** 5

**Summary:**

In this paper, the authors propose Aya Vision, a model family of multilingual multimodal open models. The authors also propose a new framework for synthetic annotation of multilingual multimodal instruction tuning data. The authors also introduce a new benchmark AyaVisionBench, a multilingual multimodal benchmark.

**Strengths:**

- The authors introduce a strong set of models Aya Vision, achieving state of the art performance on multilingual multimodal benchmarks.
- The authors performed comprehensive experiments, comparing various different models against Aya Vision, demonstrating the effectiveness of their models.
- The proposed new framework of generating synthetic annotation could help scale up multilingual multimodal training by increasing the amount of data that could be gathered for large scale training.
- The authors constructed a new benchmark AyaVisionBench, which has a broad coverage of language and topics that could comprehensively evaluate multilingual multimodal models on generation quality.

All these resources are very helpful to the multilingual community.

**Weaknesses:**

- The authors employed LLM as a judge approach for evaluation of win rate, which could be very unreliable. It would be good if they could have alternative evaluation format or perform a small set of manual evaluation to verify the results.
- The authors claim that they proposed a novel multimodal merging technique, yet multimodal merging is commonly discussed in literature. It would be good if the authors could compare their work against existing work that discuss multimodal merging.
- The synthetic data relies on expensive proprietary models. It would be great if the authors could compare with existing open models and demonstrate on par performance.

**Questions:**

- What are the computational resources required to train the Aya Vision model family?

---

> ### Author Response · Authors · 2025-11-20
> **Authors' Response (1/n)**
>
> We sincerely thank the reviewer for the very positive and detailed assessment, and for highlighting the contributions of Aya Vision, our synthetic annotation framework, and AyaVisionBench to the multilingual multimodal community. We are glad that you view these resources as broadly useful.
>
> >Reliability of LLM-as-a-judge evaluations
>
> We agree that LLM/VLM as-a-judge evaluations should be treated with care. Our intention was to mitigate this risk in three ways, which we will clarify more explicitly in the revised version:
>
> Calibration of the judge model. For multimodal preference evaluation, we selected Claude-3.7-Sonnet as the VLM judge only after a comparative study on a translated version of Multimodal RewardBench across 8 languages, where it outperformed GPT-4o and Gemini-2.0-Flash in preference-ranking accuracy. We will highlight this calibration more clearly in the main text rather than only in Appendix E.1.1.
>
> Triangulation with reference-based benchmarks. In addition to win-rates on AyaVisionBench and m-WildVision, we report results on a broad set of reference-based academic benchmarks (xMMMU, MaXM, CVQA, MTVQA, Kaleidoscope for multimodal; MGSM, Global MMLU-Lite, FLORES, IFEval for text-only), which use automatic metrics such as accuracy and BLEU-style scores rather than a judge model. The performance trends on these benchmarks are consistent with the LLM-judge win-rates, providing an independent check on the conclusions.
>
> We already note in the Limitations section that VLM-as-a-judge is only a proxy for human preference and may deviate, especially when generations are close, and we provide detailed judge prompts in Appendix Q to reduce this gap.
> We worked with LMSys to list the final Aya Vision models on the LMSys Arena to verify whether our results of Aya Vision models generating human preferred responses held true.
>
> At the time of testing, the following were the ELO scores obtained from LMSys:
> | Model             | Score |
> |-------------------|-------|
> | Aya Vision 32B    | 1097  |
> | Molmo 72B         | 1076  |
> | Claude 3 Opus     | 1076  |
> | Gemini Flash 1.5  | 1072  |
> | Pixtral 12B       | 1072  |
> | Llama3.2-90B      | 1069  |
> | Aya Vision 8B     | 1069  |
> | InternVL2 26B     | 1067  |
>
> Since Qwen2.5VL wasn’t listed on LMSys Arena, it is not possible to get a head-to-head comparison; however, Llama3.2-90B helps ground our discussion. Aya Vision 8B gets the same average ELO score as Llama3.2-90B even though Llama3.2-90B is 10x larger and scores much higher on “traditional VQA type benchmarks”. Aya Vision’s performance on LMSys Arena strengthens our argument that our training process indeed primes the model towards human-preferred responses.

---

> ### Author Response · Authors · 2025-11-20
> **Authors' Response (2/n)**
>
> >Novelty and positioning of the multimodal merging technique
>
> Thank you for pointing out the need to better situate our merging approach in the context of existing work on multimodal model merging. Our contribution is not a new generic merging algorithm, but a simple, training-free cross-modal merge tailored to the multimodal setting.
>
> Specifically, we merge a multilingually post-trained text-only LLM with its multimodal continuation along a shared optimization trajectory, interpolating the two checkpoints while preserving the visual modules. Our ablations in Section 7 show that this cross-modal merging both restores text-only performance and improves multilingual multimodal win rates, and that it is more effective than simply adding extra text-only data during SFT under the same compute budget.
>
> We will revise the paper to tone down the “novelty” language and more precisely state that we propose a cross-modal merging strategy, rather than a fundamentally new merging paradigm. We already include multimodal merging methods such as REMEDY in the Related Work section, but we will expand this discussion to reflect recent releases and clarify how our setting, merging a multilingual text-only backbone with a multimodal model to counter catastrophic forgetting, complements and differs from these works.
>
> >Use of proprietary models for synthetic data and comparison with open models
>
> Thank you for giving us a chance to clarify this point.
>
> - For recaptioning, we use an open-weight multimodal teacher model, prompted with task-specific templates.
>
> - For translation, we rely on NLLB-3.3B, an open-source model, to translate English data into 22 target languages.
>
> - For post-editing and semantic filtering, we use command-r-plus-08-2024, which is released as an open-weights model under a non-commercial research license.
>
> Thus, the core components of our synthetic annotation pipeline are open-weight models rather than black-box proprietary APIs. We will clarify this more explicitly in the data section and appendices, and emphasize that the pipeline is model-agnostic: practitioners can swap our choices for other open models depending on their licensing and cost constraints. Systematically studying the trade-off between using smaller, cheaper open models versus larger open-weight models like Command R+ for data generation could be an interesting direction, and we will consider this as a future work.
>
> >What are the computational resources required to train the Aya Vision model family?
>
> We would like to thank the reviewer for drawing attention to this important aspect of model development. We discuss the computation footprint of training (in H100 hours) Aya Vision 8B and 32B in section I of the Appendix. To ensure accessibility for the reviewers and AC, We present the computational footprint for both the models in the table below.
>
> Training compute requirements in H100 GPU-hours
>
> | Model            | Alignment | SFT   |
> |------------------|-----------|-------|
> | Aya Vision-8B    | 384       | 2176  |
> | Aya Vision-32B   | 3072      | 5120  |

---

> > ### Comment · Reviewer_ibxe · 2025-11-24
> >
> > Thank you for your comprehensive responses. I will maintain my positive scores.

---

> > > ### Author Response · Authors · 2025-11-27
> > >
> > > We thank you for all the support you've shown for our paper! We are glad you find these resources very helpful to the multilingual community.

---

### Official Review · Reviewer_KQV9 · 2025-11-01

**Soundness:** 3
**Presentation:** 3
**Contribution:** 2
**Rating:** 4
**Confidence:** 3

**Summary:**

This paper addresses the important challenge of building multilingual multimodal language models, focusing on data scarcity, translation distortion, and catastrophic forgetting when integrating visual understanding into multilingual LLMs. To tackle these issues, the authors introduce (1) a synthetic annotation framework for high-quality multilingual multimodal data and (2) a cross-modal model merging technique to preserve text-only capabilities while enhancing multimodal generation. The resulting Aya Vision models (8B and 32B) demonstrate strong performance on both text-only and multimodal benchmarks, surpassing larger counterparts.

**Strengths:**

1. The paper tackles a highly relevant and timely problem. Building multilingual multimodal systems remains a core challenge for the next generation of foundation models, and the proposed dataset construction framework provides valuable insights for the community.

2. The writing is clear, well-structured, and easy to follow. The motivation and contributions are concisely articulated, making the work accessible to a broad audience.

**Weaknesses:**

Major Concerns

1. Clarity on Key Concepts (Lines 62–63): The terms context-aware and human-preferred are mentioned as central ideas but are not elaborated in Section 2. The paper should explicitly explain how these notions are implemented or reflected in the proposed framework, particularly the human-preferred aspect.

2. Justification for Translation Choice (Lines 171–187): It remains unclear why the NLLB-3.3B model is preferred over GPT-based translation. The paper lacks empirical or qualitative evidence supporting this decision. A brief comparative evaluation or rationale would strengthen the argument.

3. Missing Experimental Analyses:

- The paper notes that Aya-Vision is built upon multilingual post-trained base models. The authors should discuss why this multilingual prior is essential and whether similar results can be achieved by directly fine-tuning strong visual-language models such as Qwen-VL.

- Figure 9’s ablation fails to demonstrate the effectiveness of the proposed context-aware rephrasing step. The authors should include a comparison with models trained on unprocessed data to quantify its benefit.

- There is little discussion on alternative model-merging strategies. Incorporating or at least discussing comparisons with TIES-Merging [1] and DARE [2] would provide deeper insight into the design choices.

Minor Issues

1. The Related Works section should not be confined entirely to the Appendix; key discussions should appear in the main text to improve context and readability.

2. Appendix C merely lists supported languages without justifying their selection. The authors should briefly explain the inclusion criteria or regional balance.

3. Appendix J introduces SafeGuards, but the main text does not reference or evaluate safety performance. A clear link to this appendix and a short discussion in the results section would help readers understand its relevance.

[1] TIES-Merging: Resolving interference when merging models. NeurIPS 2023.

[2] Language Models are Super Mario: Absorbing Abilities from Homologous Models as a Free Lunch. ICML 2024.

**Questions:**

Please see Weakness.

---

> ### Author Response · Authors · 2025-11-20
> **Authors' Response (1/n)**
>
> We would like to thank Reviewer KQV9 for their thorough and insightful review. We are greatly encouraged that you found our work to be a "highly relevant and timely" contribution to a core challenge in foundation models, and we are especially grateful that you found the paper "clear, well-structured, and easy to follow."
>
> > Clarity on Key Concepts (Lines 62–63): The terms context-aware and human-preferred are mentioned as central ideas but are not elaborated in Section 2. The paper should explicitly explain how these notions are implemented or reflected in the proposed framework, particularly the human-preferred aspect.
>
> We thank you for requesting more clarity on these core concepts. We are happy to elaborate on how they are implemented in our framework.
>
> - **Context-Aware Rephrasing**: This term refers to our specialized translation post-editing step. It is "context-aware" because, as shown in the template in Appendix K, the rephrasing model is prompted with both the original English source text and the initial machine-translated output. This full context allows the model to correct fluency issues and translationese while ensuring the final output remains semantically faithful to the original prompt.
>
> - **Human-Preferred**: This describes the primary objective of our synthetic data generation and evaluation process. Data Generation: Our recaptioning pipeline (Section 2.2) is explicitly designed to transform short, simplistic academic captions into responses that are "more detailed, natural, and diverse"  and stylistically aligned with what users prefer in a conversational assistant.
>
> We will make changes to the manuscript to incorporate this discussion.
>
>
> > Justification for Translation Choice (Lines 171–187): It remains unclear why the NLLB-3.3B model is preferred over GPT-based translation. The paper lacks empirical or qualitative evidence supporting this decision. A brief comparative evaluation or rationale would strengthen the argument.
>
> We thank the reviewer for allowing us to explain the rationale behind our choice of translation model. Although using large language models directly for machine translation is a straightforward option, prior studies have shown that LLMs only perform competitively on high-resource languages but still face challenges with low-resource languages [3, 4]. For instance, as reported in a related multilingual multimodal study [1], issues such as incorrect punctuation spacing, omission or misapplication of Arabic grammatical features (e.g., nunation), and untranslated English words are common when using LLMs alone. We observed similar issues during our own experiments.
>
> On the other hand, machine translation models like NLLB demonstrate more consistent performance across languages but are not without limitations. Notably, translation artifacts such as translationese frequently occur. Therefore, in this paper, we propose a hybrid approach: using NLLB to generate initial translations as in-context references, followed by rephrasing with a multilingual LLM. We believe this yields more natural and accurate translations. This design is supported by recent studies [4, 5], which suggest that combining LLM with traditional translation systems and leveraging examples in in-context learning can enhance translation quality, especially for low resources languages.
>
> Moreover, given the known tendency for models to favor their own generations when evaluated on data they produced [6,7], we deliberately avoided using an LLM for translation in our evaluation data to prevent bias.
>
> [1] PALO: A Polyglot Large Multimodal Model for 5B People\
> [2] Aya Model: An Instruction Finetuned Open-Access Multilingual Language Model\
> [3] How Good Are GPT Models at Machine Translation? A Comprehensive Evaluation\
> [4] Multilingual Machine Translation with Large Language Models: Empirical Results and Analysis\
> [5] Leveraging GPT-4 for Automatic Translation Post-Editing\
> [6] LLM Evaluators Recognize and Favor Their Own Generations, 2024.\
> [7] LLM see, LLM do: Leveraging active inheritance to target non-differentiable objectives, 2024.

---

> ### Author Response · Authors · 2025-11-20
> **Authors' Response (2/n)**
>
> > The paper notes that Aya-Vision is built upon multilingual post-trained base models. The authors should discuss why this multilingual prior is essential and whether similar results can be achieved by directly fine-tuning strong visual-language models such as Qwen-VL.
>
> We thank the reviewer for this comment. Our goal is a multilingual multimodal model that remains strong on both vision language tasks and text-only usage across 23 languages. In this setting, high quality multilingual multimodal data is scarce, so we rely heavily on cross lingual transfer from a multilingual post trained backbone. Our ablations on the multilingual data ratio show that naively oversampling translated multimodal data can even hurt performance, while starting from a strong multilingual text model substantially improves downstream multilingual multimodal quality.
>
> The multilingual prior is also critical for our cross modal merging strategy. This technique assumes that the multimodal model and its text only counterpart follow a similar optimization path, so that simple weight interpolation can recover text skills without additional training. This works well only if the backbone already has strong multilingual instruction following ability. Empirically, Aya Vision 8B loses only 5.9% text only on mArenaHard relative to its backbone, while other VLMs such as Qwen-2.5-VL 7B and Molmo-7B lose 16-44% relative to their text bases.
>
> Regarding Qwen-VL, our contributions are largely model agnostic. In principle, the same multilingual synthetic data pipeline and cross modal merging could be applied to a Qwen text and Qwen-VL pair. In this work we instead treat Qwen-2.5-VL 7B and 72B as strong baselines, and Aya Vision 8B and 32B outperform them on multilingual multimodal preference evaluations while remaining competitive on academic benchmarks at similar or smaller scale.
>
> We will clarify these points in the revised version by explicitly stating why a multilingual post-trained backbone is needed for our data-scarce multilingual setting and for cross-modal merging and adding a short discussion that our methods are in principle applicable to other VLM families such as Qwen-VL.

---

> ### Author Response · Authors · 2025-11-20
> **Authors' Response (3/n)**
>
> > Figure 9’s ablation fails to demonstrate the effectiveness of the proposed context-aware rephrasing step. The authors should include a comparison with models trained on unprocessed data to quantify its benefit.
>
> We thank the reviewer for this comment. Figure 9 is designed to show the effect of our data framework on Aya Vision 8B, rather than to isolate the context aware rephrasing step. The 17.2 point increase directly quantifies the benefit of training on processed versus unprocessed data. We also quantify the effect of this pipeline, including rephrasing, at the data level. It expands average caption length from 27.2 to 140.8 tokens, increases lexical diversity from 11.0 to 61.2 (presented in line 143-148, p3), and improves translation quality as measured by COMET. Table 7 in Appendix O reports the reference free COMET score by language, and we observe consistent improvements for all 23 languages, with relative gains ranging from 1.8% for Hindi to 80.3% for Traditional Chinese. We believe these systematic improvements in translation quality are an important driver of the stronger multilingual performance of Aya Vision.
>
> Training an additional full Aya Vision 8B variant that removes only the rephrasing step while keeping all other components fixed would be computationally expensive, and the three stages of the pipeline interact in practice. In the revised version we will make the comparison to unprocessed data more explicit in the caption and text of Figure 9, clarify that the “data improvements” bar corresponds to the combined recaptioning, translation, and context aware rephrasing pipeline, and highlight the language specific COMET gains from Table 7.
>
> > There is little discussion on alternative model-merging strategies. Incorporating or at least discussing comparisons with TIES-Merging [1] and DARE [2] would provide deeper insight into the design choices.
>
> We agree that model-merging is an active and relevant line of work. Our focus in this paper, however, is a specific cross-modal setting: we merge a multilingual text-only LLM with its multimodal continuation to recover text capabilities that are partially lost during vision-language fine-tuning, while simultaneously improving multimodal performance. Concretely, we perform a simple weight-space interpolation between two checkpoints that share an optimization trajectory, while keeping the vision encoder and connector compatible with the merged model.
> In contrast, TIES-merging and DARE are primarily designed for merging multiple text-only models or sparse task-specific deltas over a shared backbone in order to combine heterogeneous task specializations. Applying these methods in our setting would require additional design choices, such as how to handle vision-specific parameters and the vision-language connector, and would address somewhat different questions than the ones we study here: multilingual multimodal data design and cross-modal capability preservation.
>
> We therefore view TIES and DARE as complementary but orthogonal to our contribution: **our aim is to show that a simple cross-modal interpolation is already sufficient to substantially restore text performance and improve multilingual multimodal performance, without additional training** – a stark departure from how text-only performance is preserved in prior approaches. For these reasons, we prefer to keep the scope of this work focused on our cross-modal merging setup, and leave a systematic comparison with advanced merging schemes such as TIES and DARE to future work.

---

> ### Author Response · Authors · 2025-11-20
> **Authors' Response (4/n)**
>
> > The Related Works section should not be confined entirely to the Appendix; key discussions should appear in the main text to improve context and readability.
>
> We thank the reviewer for this suggestion. We will use the extra page in the camera ready version to move the related works section to the main text.
>
> > Appendix C merely lists supported languages without justifying their selection. The authors should briefly explain the inclusion criteria or regional balance.
>
> The 23 languages in Appendix C are exactly those natively supported and explicitly optimized in our multilingual backbone LLM and cover approximately half the world’s population. We follow this set to stay within the backbone's strongest coverage while ensuring a diverse mix of scripts and regions. In addition, all these languages are supported with reasonable quality by our translation backbone (NLLB-3.3B) and post-editing pipeline, they have sufficient downstream evaluation coverage in our multilingual benchmark suite so that multilingual gains can be meaningfully measured, and taken together, they provide a diverse set of scripts and regions, while keeping the total number of translation directions computationally manageable.
>
> In the revised version, we will update Appendix C and briefly cross-reference it in Section 2 to explicitly state these inclusion criteria and to clarify that our goal is to balance global coverage and script diversity under realistic data and compute constraints, rather than to claim exhaustive language coverage.
>
>
> > Appendix J introduces SafeGuards, but the main text does not reference or evaluate safety performance. A clear link to this appendix and a short discussion in the results section would help readers understand its relevance.
>
> We thank the reviewer for bringing this to our attention, we’ll use the extra page in the camera ready version to incorporate this.

---

> ### Author Response · Authors · 2025-11-27
>
> We appreciate your insightful feedback. We have made a strong effort to address the points you have raised and will ensure all updates are reflected in the camera-ready manuscript. We believe these revisions would significantly strengthen our paper. We hope our responses resolve your reservations, and we would be grateful if you would consider raising your score. We are happy to answer any further questions.

---

### Author Response · Authors · 2025-11-20
**Global Response by Authors**

We would like to express our sincere gratitude to all the reviewers for their thoughtful and constructive feedback on our work. We are greatly encouraged by the positive reception and are pleased that the reviewers recognized our work as a step toward "inclusive and global AI systems" [6eZf], acknowledging our "excellent" [ibxe] and "meaningful" [55FY] contributions to the community.

We are particularly heartened that reviewers recognized the core strengths of our work, including:
- The effectiveness of our cross-modal model merging technique as a "simple and effective" [55FY], "elegant" [kmAj], and "interesting" [6eZf] solution to catastrophic forgetting.
- Our synthetic annotation framework for generating high-quality data, which provides "valuable insights" [KQV9] and represents a "meaningful data contribution" [55FY].
- The "strong results" [KQV9] and "state of the art performance" [ibxe] of our models, often "surpassing larger counterparts" [KQV9].
- The contribution of our new resources, including AyaVisionBench, which are "very helpful to the multilingual community" [ibxe] and help "standardize evaluation" [55FY].
- The overall quality of the manuscript, described as "clear, well-structured, and easy to follow" [KQV9].

Below, we address the major themes raised across reviews regarding benchmark discrepancies, data pipeline design, and the merging technique.

### Academic Benchmarks vs. Human Preference & New LMSys Results

Reviewers noted a gap between Aya Vision’s strong performance on open-ended evaluations (AyaVisionBench, m-WildVision) and its lower scores on strict-format academic benchmarks (xMMMU, CVQA) compared to larger models. We clarify that this difference reflects Aya Vision’s optimization objective: the model is trained to produce fluent, helpful, instruction-following responses rather than the short, exact-match answers emphasized in traditional VQA benchmarks. As discussed in our individual responses, such benchmarks often penalize valid, natural-language answers that do not exactly match a reference string, even when they are semantically equivalent.

To further validate that this design choice leads to better real-world utility, we submitted Aya Vision to the LMSys Chatbot Arena for blind human evaluation. The results strongly support our approach:\
**Aya Vision 32B (ELO: 1097): Outperforms Molmo 72B (1076) and Llama-3.2-90B (1069).**\
**Aya Vision 8B (ELO: 1069): Matches Llama-3.2-90B, despite being ~10x smaller.**

These results demonstrate that Aya Vision sets a new standard for efficiency and user preference within the open-weight model category. Moreover, comparisons to large-scale baselines such as Qwen-2.5-VL-72B should be interpreted in light of compute efficiency. Qwen models are trained on approximately 4 trillion tokens, whereas Aya Vision uses fewer than 40 billion, about 100 times less data, making its performance highly competitive relative to its compute scale.

As detailed in Appendix I, training Aya Vision 32B requires a total of 8192 H100 GPU hours. On 128 GPUs, this corresponds to 64 hours (2.66 days). Assuming identical throughput and scaling to 1024 GPUs, training Qwen-2.5-VL-72B would take approximately $64\times100/(8\times24)$ = ~33.3 days.

### Rationale for the Hybrid Data Pipeline

Reviewers requested clarification on our decision to use a hybrid translation pipeline (NLLB + LLM rephrasing) instead of relying solely on either machine translation (MT) or large language model (LLM) translation. We adopt the hybrid approach to balance faithfulness and fluency: pure LLM translation often introduces hallucinations or grammatical inconsistencies in low-resource languages, while pure MT systems tend to produce rigid, unnatural “translationese”. By first generating faithful translations with NLLB and then applying LLM-based rephrasing, we combine the semantic accuracy of MT with the naturalness of LLM outputs.

Empirically, this hybrid design yields clear benefits.\
**Data Level**: We observe consistent COMET score improvements across all 23 languages (0.75 $\to$ 0.83 avg) and a massive increase in lexical diversity (MTLD 11.0 $\to$ 61.2).\
**Model Level**: Figure 9 demonstrates that training on this processed data yields a 17.2% increase in multilingual vision win rates compared to unprocessed data.

### Novelty and Positioning of Cross-Modal Merging
Reviewers asked for comparisons to general merging methods (e.g., TIES) and clarification on novelty.
We clarify that our contribution is not a generic merging algorithm, but a specific cross-modal strategy tailored to mitigate catastrophic forgetting. We demonstrate that simple interpolation between a multilingual text backbone and its multimodal continuation restores text capabilities and improves multimodal generation without additional training. We view this as orthogonal to complex weight-merging schemes like TIES, offering a lightweight solution specifically for the modality gap.

---

### Meta-Review · Area_Chair_xvUi · 2026-01-06

**Summary:**

Reviewers raised concerns about limited methodological clarity, the conceptually unclear and weakly justified cross-modal merging strategy, and the evaluation solidness. In particular, the strongest gains are demonstrated on author-introduced benchmarks and LLM-as-judge evaluations, while performance on standard accuracy-based benchmarks lags largely behind, leaving the “human-preferred” claim insufficiently supported. Given the high standard of ICLR as a top-tier conference, I partially agree with the reviewers that the submission is not yet ready for acceptance and therefore recommend rejection.

**Reviewer Concerns:**

After carefully reviewing the discussion, I believe that several key concerns remain unresolved: the lack of solid theoretical justification for cross-modal merging, the gap between strong results on open-ended or LLM-judged benchmarks and weaker performance on standard accuracy-based benchmarks, and the "human-preferred” claim is not solidly validated by the experiments.

**Reviewer Scores:**

The initial scores are 4/4/8/2/2, with four negative reviews and one positive review, and the concerns raised by the reviewers are substantial. While the authors attempted to address these issues in the rebuttal, I believe it would be difficult to overturn the overall negative assessment.

---

### Decision · Program_Chairs · 2026-01-26

Reject